# Dependence of diffusion in *Escherichia coli* cytoplasm on protein size, environmental conditions, and cell growth

**Nicola Bellotto[1], Jaime Agudo-Canalejo[2†], Remy Colin[1†], Ramin Golestanian[2,3]\*, Gabriele Malengo[1]\*, Victor Sourjik[1]\***

[1]Max Planck Institute for Terrestrial Microbiology and Center for Synthetic Microbiology (SYNMIKRO), Marburg, Germany; [2]Max Planck Institute for Dynamics and Self-Organization, Göttingen, Germany; [3]Rudolf Peierls Centre for Theoretical Physics, University of Oxford, Oxford, United Kingdom

**Abstract** Inside prokaryotic cells, passive translational diffusion typically limits the rates with which cytoplasmic proteins can reach their locations. Diffusion is thus fundamental to most cellular processes, but the understanding of protein mobility in the highly crowded and non-homogeneous environment of a bacterial cell is still limited. Here, we investigated the mobility of a large set of proteins in the cytoplasm of *Escherichia coli*, by employing fluorescence correlation spectroscopy (FCS) combined with simulations and theoretical modeling. We conclude that cytoplasmic protein mobility could be well described by Brownian diffusion in the confined geometry of the bacterial cell and at the high viscosity imposed by macromolecular crowding. We observed similar size dependence of protein diffusion for the majority of tested proteins, whether native or foreign to *E. coli*. For the faster-diffusing proteins, this size dependence is well consistent with the Stokes-Einstein relation once taking into account the specific dumbbell shape of protein fusions. Pronounced subdiffusion and hindered mobility are only observed for proteins with extensive interactions within the cytoplasm. Finally, while protein diffusion becomes markedly faster in actively growing cells, at high temperature, or upon treatment with rifampicin, and slower at high osmolarity, all of these perturbations affect proteins of different sizes in the same proportions, which could thus be described as changes of a well-defined cytoplasmic viscosity.

**\*For correspondence:**
Ramin.Golestanian@ds.mpg.de (RG);
gabriele.malengo@synmikro.
mpi-marburg.mpg.de (GM);
victor.sourjik@synmikro.mpi-
marburg.mpg.de (VS)

†These authors contributed
equally to this work

**Competing interest:** The authors
declare that no competing
interests exist.

**Reviewing Editor:** Ariel Amir,
Harvard University, United States

## Editor's evaluation

The work of Bellotto et al. provides a comprehensive and compelling study of the diffusion of proteins in the cytoplasm of the bacterium *Escherichia coli*, using multiple measurement methods, notably Fluorescence Correlation Spectroscopy. It is found that fast diffusing proteins roughly follow the Stokes-Einstein relation, while proteins that strongly interact with the cytoplasm manifest subdiffusion. This study will be a valuable resource for scientists seeking to understand the temporal dynamics of proteins within cells.

## Introduction

Diffusion of molecules is important for the function of any cellular system, setting the upper limit for the mobility of proteins and other (macro)molecules and for the rates of many biochemical reactions that rely on random encounters between molecules (*Schavemaker et al., 2018*). Although the fundamental physics of diffusion in dilute aqueous solutions is well understood and mathematically described (*Einstein, 1906*; *Langevin, 1908*; *Perrin, 1910*), diffusion in a cellular environment may be

quite different (*Schavemaker et al., 2018*; *Mika and Poolman, 2011*). The concentration of macro-molecules in the bacterial cytoplasm, primarily proteins but also ribonucleic acids (RNAs), a phenom-enon known as macromolecular crowding, is extremely high. For *Escherichia coli*, it is around 300 mg/ml, which corresponds to a volume fraction of 25–30% (*Cayley et al., 1991*; *Zimmerman and Trach, 1991*). Such macromolecular crowding could hinder free diffusion and influence kinetics of protein association and of gene expression (*Klumpp et al., 2013*; *Tabaka et al., 2014*; *van den Berg et al., 2017*). The effects of crowding on protein diffusion have been demonstrated both in vitro and in vivo (*Dix and Verkman, 2008*; *Rivas and Minton, 2016*). Compared to water, the diffusion of a free green fluorescent protein (GFP) was reported to be 3–4 times slower in the eukaryotic cytoplasm (*Swaminathan et al., 1997*) and up to 10 times slower in the bacterial cytoplasm (*Elowitz et al., 1999*; *Nenninger et al., 2010*; *Mika et al., 2010*; *Kumar et al., 2010*).

In addition to the high density of macromolecules, the diversity in the size and chemical proper-ties of the solutes makes the cytoplasmic environment highly inhomogeneous (*Luby-Phelps, 1999*; *Spitzer and Poolman, 2013*). How much the diffusion of a particular molecule is affected by macro-molecular crowding might thus depend on the size (*Muramatsu and Minton, 1988*) and the shape of the molecule (*Balbo et al., 2013*) as well as on the nature of the crowders (*Banks and Fradin, 2005*; *Goins et al., 2008*). The effects of crowding observed in living cells appear to be even more complex, varying not only with the properties of the diffusing particle but also with the physiological state of the cell (*Parry et al., 2014*; *Joyner et al., 2016*) and the local cellular environment (*Konopka et al., 2006*; *Persson et al., 2020*). Moreover, non-trivial effects on diffusion arise due to reversible assembly and disassembly of the diffusing protein complexes (*Agudo-Canalejo et al., 2020*), and possibly also due to the active enhancement of enzyme diffusion by catalytic reactions (*Golestanian, 2015*; *Agudo-Canalejo et al., 2018*; *Zhang and Hess, 2019*).

The dependence of the diffusion coefficient ($D$) of a protein in the cytoplasm on its size might thus not necessarily follow the Stokes-Einstein (also called Stokes-Einstein-Sutherland-Smoluchowski) rela-tion that is valid in dilute solutions, $D \propto T/(\eta R)$ (*Einstein, 1906*), where $T$ is the absolute tempera-ture in Kelvin, $\eta$ is the viscosity of the medium, and $R$ is the hydrodynamic radius of the particle. For globular proteins, $R$ is given by the radius of gyration (*Tyn and Gusek, 1990*) and depends on the molecular mass (MM) as $R \propto \mathrm{MM}^{\beta}$, where the exponent $\beta$ would be 1/3 for perfectly compact and globular proteins but is in practice within the range of 0.35–0.43 for typical proteins, reflecting the fractal nature of the spatial distribution of protein mass (*Smilgies and Folta-Stogniew, 2015*; *Enright and Leitner, 2005*). Several studies of protein diffusion in the cytoplasm of *E. coli* have yielded different dependencies on the molecular mass, from ~0.33 (*Nenninger et al., 2010*) to ~2 (*Kumar et al., 2010*), with an average $\beta \sim 0.7$ estimated based on the data pooled from multiple studies (*Mika and Poolman, 2011*; *Kalwarczyk et al., 2012*), and thus substantially steeper than predicted by the Stokes-Einstein relation. Similar exponent of ~0.7 was observed for limited sets of differently sized proteins (*Mika et al., 2010*; *Stracy et al., 2021*). However, neither of these studies took explicitly into account the non-globularity of the used fluorescent constructs, where two or more proteins are typically connected by flexible linkers. For such multidomain proteins, shape fluc-tuations and hydrodynamic interactions between the different domains can have a sizeable effect on the effective diffusion coefficient of the whole protein (*Agudo-Canalejo and Golestanian, 2020*), and they might thus be important to consider when interpreting deviations from the Stokes-Einstein relation.

Besides macromolecular crowding, the translational diffusion of cytoplasmic proteins is also influ-enced by intracellular structures, such as cytoskeletal filaments (*Sabri et al., 2020*), and by (transient) binding to other macromolecules (*Saxton, 2007*; *Guigas and Weiss, 2008*; *von Bülow et al., 2019*). Both these factors can not only reduce protein mobility but also lead to the anomalous subdiffusive behavior, where the mean square displacement (MSD) of diffusing particles does not scale linearly with time, as for Brownian diffusion in dilute solutions, but rather follows MSD $\alpha \; t^{\alpha}$ with the anomalous diffusion exponent $\alpha$ being <1 (*Saxton, 1996*; *Etoc et al., 2018*). Subdiffusion is commonly observed in eukaryotes, particularly at longer spatial scales, primarily due to the obstruction by the cytoskeletal filaments to the diffusion of proteins and larger particles (*Di Rienzo et al., 2014*; *Sabri et al., 2020*). The mobility of larger nucleoprotein (*Golding and Cox, 2004*; *Lampo et al., 2017*) and multiprotein particles (*Yu et al., 2018*) in the bacterial cytoplasm is also subdiffusive, while the diffusion of several tested small proteins was apparently Brownian (*Bakshi et al., 2011*; *English et al., 2011*).

Even for the same protein, for example, GFP or its spectral variants, estimates of the diffusion coefficient in the cytoplasm obtained in different studies vary widely (*Schavemaker et al., 2018*), which could be in part due to differences in methodologies. Most early studies in bacteria relied on fluorescence recovery after photobleaching (FRAP), where diffusion is quantified from the recovery of fluorescence in a region of the cell bleached by a high-intensity laser (*Lorén et al., 2015*). These measurements provided values of diffusion coefficient for GFP ranging from 3 to 14 µm² s⁻¹ (*Elowitz et al., 1999*; *Mullineaux et al., 2006*; *Konopka et al., 2009*; *Kumar et al., 2010*; *Mika et al., 2010*; *Nenninger et al., 2010*; *Schavemaker et al., 2017*). More recently, single-particle tracking (SPT), where diffusion is measured by following the trajectories of single fluorescent molecules over time (*Kapanidis et al., 2018*), became increasingly used. Finally, diffusion can also be studied in vivo using fluorescence correlation spectroscopy (FCS) (*Cluzel et al., 2000*), which measures the time required by a fluorescent molecule to cross the observation volume of a confocal microscope (*Elson, 2011*). SPT and FCS measure protein mobility locally within the cell, with FCS having also a significantly better temporal resolution than FRAP and SPT. Both methods provided higher but still varying values of $D_{GFP}$, from 8 µm² s⁻¹ up to 18 µm² s⁻¹ (*Meacci et al., 2006*; *English et al., 2011*; *Sanamrad et al., 2014*; *Diepold et al., 2017*; *Rocha et al., 2019*).

Protein mobility also depends on the environmental and cellular conditions that affect the structure of the bacterial cytoplasm (*Schavemaker et al., 2018*). Diffusion of large cytoplasmic particles, measured by SPT, was shown to be sensitive to the antibiotics-induced changes in the cytoplasmic crowding (*Wlodarski et al., 2020*) and to the energy-dependent fluidization of the cytoplasm (*Parry et al., 2014*). Protein diffusion is also affected by high osmolarity that increases macromolecular crowding and might create barriers to diffusion (*Konopka et al., 2006*; *Konopka et al., 2009*; *Liu et al., 2019*). Furthermore, the surface charge of cytoplasmic proteins has been shown to have a dramatic effect on their mobility (*Schavemaker et al., 2017*).

Variations between values of diffusion coefficients observed even for the same model organism in different studies, each investigating only a limited number of protein probes, using different strains, growth conditions, and measurement techniques, hampered drawing general conclusions about the effective viscosity of bacterial cytoplasm and its dependence on the protein size. Furthermore, while the impact of several physiological perturbations on protein diffusion has been established, most of these previous studies used either large particles or free GFP, and how these perturbations affect the properties of the cytoplasm over the entire physiological range of protein sizes remained unknown.

Here, we address these limitations by systematically analyzing the mobility of a large number of differently sized cytoplasmic fluorescent protein constructs under standardized conditions by FCS. We further combined experiments with Brownian dynamics simulations and theoretical modeling of diffusion to correct for effects of confined cell geometry. Our work establishes general methodology to analyze FCS measurements of protein mobility in a confined space, which could be broadly applicable to cellular systems.

For the majority of studied constructs, we observe consistent dependence of the diffusion coefficient on the protein size, with a pronounced upper limit on diffusion at a given molecular mass. When corrected for the confinement due to the bacterial cell geometry, the diffusion of these constructs was nearly Brownian. Moreover, part of the deviation of the mass-dependence of their diffusion coefficients from the Stokes-Einstein relation might be explained by the specific shape of the fusion proteins. The slower and more anomalous diffusion of several protein constructs was apparently due to their strong interactions with other cellular proteins and protein complexes, and disruption of these interactions restored a Brownian diffusion close to the upper limit expected for their mass. Proteins that are not native to *E. coli* were observed to diffuse very similarly to their *E. coli* counterparts, except for their motion being slightly subdiffusive. Under the same experimental conditions FCS and FRAP measurements yield similar values of diffusion coefficients, suggesting that no pronounced dependence of protein mobility on spatial scale could be observed in the bacterial cytoplasm. Finally, we investigated the effects of environmental osmolarity and temperature, of exposure to antibiotics and of cell growth on the mobility of proteins of different size, demonstrating that the effects of all these perturbations, including cell growth, on protein diffusion could be simply explained by changes in a unique cytoplasmic viscosity.

## Results

## Dependence of cytoplasmic protein mobility on molecular mass measured by FCS

For our analysis of cytoplasmic protein mobility, we generated a plasmid-encoded library of 31 cytoplasmic proteins (*Table 1*) of *E. coli* fused to superfolder GFP (sfGFP) (*Pédelacq et al., 2006*). We selected proteins that belong to different cellular pathways and, according to the available information, are not known to bind DNA or to form homomultimers, although we did not exclude a priori proteins that interact with other proteins. The structure of all selected proteins is known and roughly globular, avoiding effects of the irregular protein shape on mobility. The expected size and stability of each construct were verified by gel electrophoresis and immunoblotting (*Figure 1—figure supplement 1*). Only one of the constructs, ThpR-sfGFP, showed >20% degradation to free sfGFP, and it was therefore excluded from further analyses. This was also the sole construct with an atypically high isoelectric point (pI), and all remaining constructs have pI ranging from 5.1 to 6.2, as common for cytoplasmic proteins (*Schwartz et al., 2001*). We further imaged the distribution of fusion proteins in the cytoplasm. Except for RihA-sfGFP and NagD-sfGFP that were subsequently excluded, all other constructs showed uniform localization (*Figure 1A*). Expression of most fusion proteins used for the measurements of diffusion had little effect on *E. coli* growth (*Figure 1—figure supplement 2*), and even for several proteins where expression delayed the onset of the exponential growth, the growth rate around the mid-log phase when cultures were harvested for the analysis was similar. The mobility of the remaining 28 fusion constructs and of free sfGFP was investigated in living *E. coli* cells by FCS (see Materials and methods and Appendix 2). In order to reduce the impact of photobleaching on FCS measurements, cell length was moderately (approximately twofold) increased by treatment with the cell-division inhibitor cephalexin for 45 min, yielding an average cell length of ~5 μm (*Figure 1A*). The resulting larger cell volume indeed reduces the rate of photobleaching. During each FCS measurement, the laser focus was positioned close to the polar region in the cell cytoplasm, in order to keep the confocal volume possibly away from both the cell membrane and the nucleoid, and the fluorescence intensity in the confocal volume was measured over time (*Figure 1—figure supplement 3*). For each individual cell, six subsequent acquisitions of 20 s each were performed at the same position. The autocorrelation function (ACF) of the fluorescence intensity fluctuations was independently calculated for each time interval and fitted to extract the mobility parameters of the fluorescent proteins. Although we initially considered both the Brownian diffusion and the anomalous diffusion models, the latter model proved to be considerably better in fitting the experimental data (*Figure 1—figure supplement 4*). The anomalous diffusion model was therefore used to determine the diffusion (or residence) time ($\tau_D$) of a fluorescent molecule in the confocal volume and the anomalous diffusion exponent $\alpha$ for all ACFs (*Figure 1B*, *Table 1*, and *Figure 1—figure supplement 3*). The averaged values of $\tau_D$ and $\alpha$ for each individual cell were then calculated from these six individual acquisitions (*Figure 1C* and *Figure 1—figure supplement 5*). Although, as mentioned above, all finally used protein constructs showed no or little degradation, we tested a possible impact of the fraction of free sfGFP for the construct that displayed the strongest (~15%) degradation, DsdA-sfGFP. To this end, we fitted the FCS data using a model of two-components anomalous diffusion, where the weight of the fast component was fixed to 15% and its values of $\tau_D$ and $\alpha$ to the average values obtained for sfGFP (*Figure 1—figure supplement 6*). The average value of $\tau_D$ for the slow component was only ~7% lower compared to our regular fit using the one-component model, and the value of $\alpha$ remained unchanged, suggesting that the impact of an even smaller fraction of free GFP for other constructs could also be neglected. As another control, we observed no significant correlation between the values of $1/\tau_D$ or $\alpha$ and the length or the width of individual cells, although a weak trend of $\alpha$ increasing with cell width might exist (*Figure 1—figure supplement 7*). Finally, when individual cephalexin-treated and untreated cells of similar length were compared, we observed no effect of the treatment on the value of $\alpha$ and only marginal (*p*=0.08) increase in the mobility of sfGFP (*Figure 1—figure supplement 8*).

Despite their substantial intercellular variability, the obtained mean values of the diffusion time were clearly different between protein constructs (*Figure 1C* and *Table 1*). We next plotted the mean values of $1/\tau_D$, which reflect protein mobility, against the molecular mass of protein constructs (*Figure 1D*). This dependence revealed a clear trend, where mobility of more than half of the constructs decreased uniformly with their molecular mass, while some exhibited much lower mobility than the other constructs of similar mass. In contrast, the anomalous diffusion exponent $\alpha$ showed no apparent

**Table 1.** Molecular mass, biological function, and measured parameters for all studied sfGFP fusion constructs. The concentration of expression inducer and the number of cells measured with each technique is also indicated.

| Protein name | Molecular mass of sfGFP fusion construct | Biological function in *E. coli* | IPTG concentration used for FCS (FRAP) | Number of cells analyzed by FCS | $\tau_D$ (µs; mean ± SEM) | $\alpha$ (mean ± SEM) | Diffusion coefficient, FCS (µm²/s, mean ± SEM) | Number of cells analyzed by FRAP | Diffusion coefficient, FRAP (µm²/s, mean ± SEM) |
|---|---|---|---|---|---|---|---|---|---|
| sfGFP | 26.9 | – | 5 µM (15 µM) | 52 | 561±14 | 0.86±0.01 | 14.7±0.3 | 11 | 11.3±1.3 |
| YggX | 39.2 | Probable Fe (2+)-trafficking protein | 5 µM (5 µM) | 8 | 611±19 | 0.85±0.01 | 12.9±0.4 | 10 | 9.4±1.6 |
| ClpS | 39.2 | ATP-dependent Clp protease adapter protein | 0 µM | 11 | 1054±33 | 0.75±0.01 | | | |
| FolK | 45.1 | 2-amino-4-hydroxy-6-hydroxymethyldihydropteridine pyrophosphokinase | 0 µM | 8 | 734±24 | 0.87±0.01 | 11.6±0.4 | | |
| Crr | 45.2 | Component of glucose-specific phosphotransferase enzyme IIA | 0 µM | 14 | 1065±36 | 0.87±0.01 | | | |
| UbiC | 45.7 | Chorismate pyruvate-lyase | 15 µM | 14 | 1140±58 | 0.87±0.01 | | | |
| ThpR | 46.9 | RNA 2',3'-cyclic phosphodiesterase | Discarded due to instability of sfGFP fusion construct | | | | | | |
| CoaE | 49.6 | Dephospho-CoA kinase | 0 µM | 11 | 854±47 | 0.87±0.01 | 9.8±0.6 | | |
| Adk | 50.6 | Adenylate kinase | 5 µM (15 µM) | 23 | 802±26 | 0.88±0.00 | 10.6±0.4 | 16 | 9.8±1.5 |
| Cmk | 51.7 | Cytidylate kinase | 5 µM | 16 | 1163±58 | 0.87±0.01 | | | |
| NagD | 54.1 | Ribonucleotide monophosphatase | Discarded due to non-uniform protein localization | | | | | | |
| KdsB | 54.6 | 3-deoxy-manno-octulosonate cytidylyltransferase | 0 µM | 11 | 1659±70 | 0.84±0.01 | | | |
| Map | 56.3 | Methionine aminopeptidase | 0 µM | 20 | 1830±78 | 0.81±0.01 | | | |

*Table 1 continued on next page*

*Table 1 continued*

| Protein name | Molecular mass of sfGFP fusion construct | Biological function in *E. coli* | IPTG concentration used for FCS (FRAP) | Number of cells analyzed by FCS | $\tau_D$ (µs; mean ± SEM) | $\alpha$ (mean ± SEM) | Diffusion coefficient, FCS (µm²/s, mean ± SEM) | Number of cells analyzed by FRAP | Diffusion coefficient, FRAP (µm²/s, mean ± SEM) |
|---|---|---|---|---|---|---|---|---|---|
| MmuM | 60.4 | Homocysteine S-methyltransferase | 5 µM | 14 | 2241±138 | 0.73±0.01 | | | |
| RihA | 60.8 | Pyrimidine-specific ribonucleoside hydrolase | Discarded due to non-uniform protein localization | | | | | | |
| PanE | 60.8 | 2-dehydropantoate 2-reductase | 0 µM (5 µM) | 18 | 1059±26 | 0.85±0.01 | 7.8±0.2 | 11 | 5.2±0.6 |
| SolA | 67.9 | N-methyl-L-tryptophan oxidase | 0 µM | 7 | 795±31 | 0.82±0.01 | 9.9±0.5 | | |
| Pgk | 68.1 | Phosphoglycerate kinase | 0 µM | 16 | 991±41 | 0.90±0.01 | 8.6±0.3 | | |
| EntC | 69.9 | Isochorismate synthase | 15 µM | 15 | 1777±119 | 0.82±0.01 | | | |
| AroA | 73.1 | 3-phosphoshikimate 1-carboxyvinyltransferase | 5 µM | 9 | 995±69 | 0.86±0.01 | 8.7±0.7 | | |
| ThrC | 74.1 | Threonine synthase | 0 µM | 14 | 908±28 | 0.87±0.01 | 9.1±0.3 | | |
| MurF | 74.4 | UDP-N-acetylmuramoyl-tripeptide--D-alanyl-D-alanine ligase | 0 µM | 7 | 1008±76 | 0.85±0.02 | 8.3±0.7 | | |
| DsdA | 74.9 | D-serine dehydratase | 0 µM | 14 | 1017±53 | 0.89±0.01 | 8.4±0.4 | 10 | 7.8±0.7 |
| HemN | 79.7 | Oxygen-independent coproporphyrinogen III oxidase | 0 µM | 13 | 1262±54 | 0.86±0.01 | 6.7±0.4 | | |
| PrpD | 80.9 | 2-methylcitrate dehydratase | 0 µM | 12 | 1866±140 | 0.84±0.01 | | | |

*Table 1 continued*

| Protein name | Molecular mass of sfGFP fusion construct | Biological function in *E. coli* | IPTG concentration used for FCS (FRAP) | Number of cells analyzed by FCS | $\tau_D$ (µs; mean ± SEM) | $\alpha$ (mean ± SEM) | Diffusion coefficient, FCS (µm²/s, mean ± SEM) | Number of cells analyzed by FRAP | Diffusion coefficient, FRAP (µm²/s, mean ± SEM) |
|---|---|---|---|---|---|---|---|---|---|
| DnaK | 96.0 | Molecular chaperone | 5 µM | 10 | 2296±78 | 0.76±0.01 | | | |
| MalZ | 96.0 | Maltodextrin glucosidase | 0 µM | 9 | 3725±229 | 0.77±0.01 | | | |
| GlcB | 107.5 | Malate synthase G | 5 µM (15 µM) | 16 | 1315±45 | 0.86±0.01 | 6.4±0.2 | 10 | 6.7±1.1 |
| MetE | 111.7 | 5-methyltetrahydropteroyltriglutamate--homocysteine methyltransferase | 5 µM | 8 | 1137±53 | 0.87±0.01 | 7.4±0.3 | | |
| LeuS | 124.2 | Leucine--tRNA ligase | 0 µM | 14 | 1637±75 | 0.86±0.01 | 5.1±0.2 | | |
| AcnA | 124.7 | Aconitate hydratase A | 5 µM (15 µM) | 19 | 1415±56 | 0.86±0.01 | 6.1±0.2 | 10 | 4.3±0.4 |
| MetH | 163.0 | Methionine synthase | 0 µM (5 µM) | 9 | 1402±45 | 0.81±0.01 | 5.8±0.1 | 15 | 4.0±0.5 |

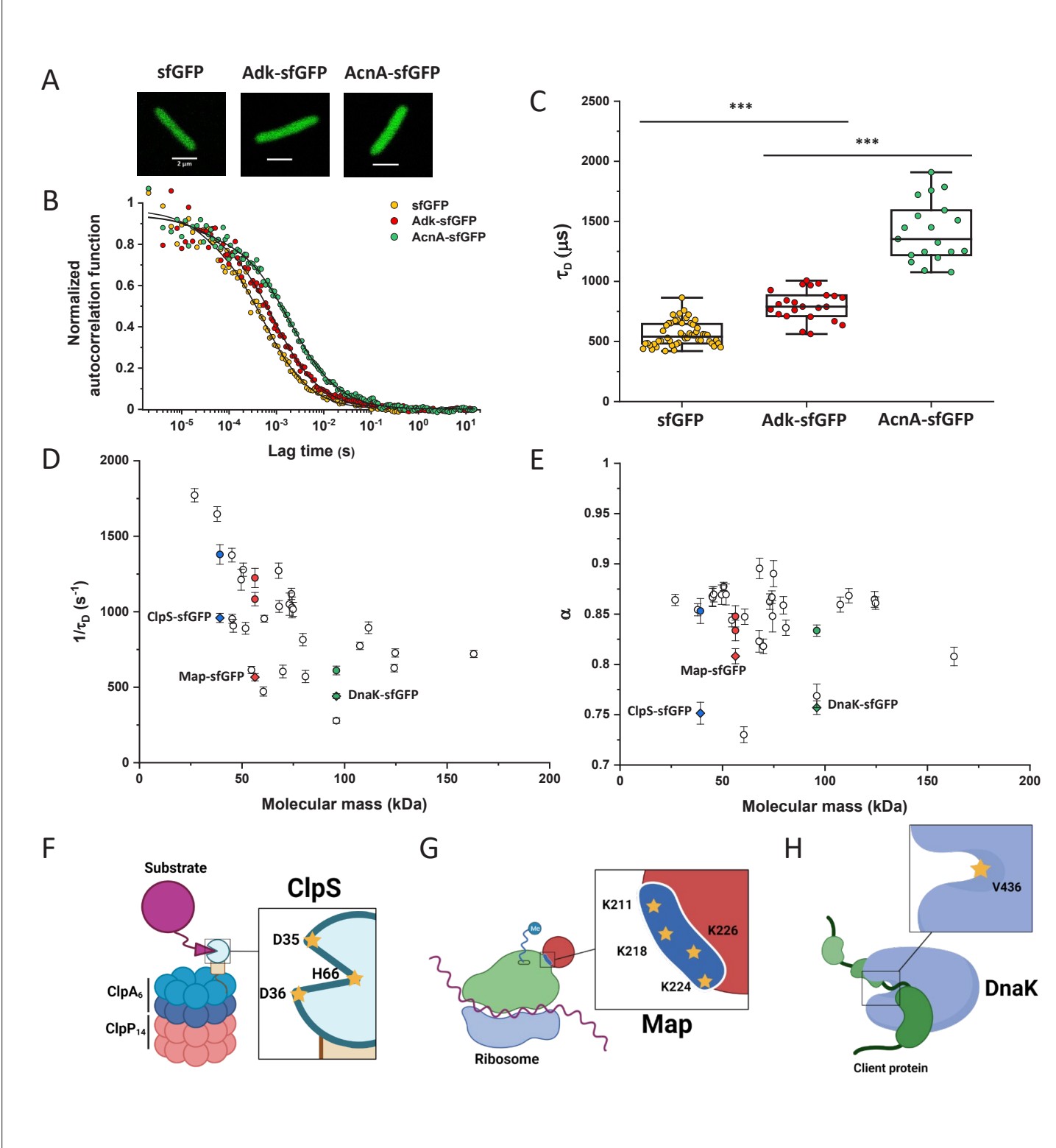

**Figure 1.** Dependence of protein mobility in bacterial cytoplasm on molecular mass and cellular interactions. (**A**) Examples of fluorescence microscopy images of *Escherichia coli* cells expressing either sfGFP or the indicated sfGFP-tagged cytoplasmic proteins. Scale bars are 2 µm. (**B**) Representative autocorrelation functions (ACFs) measured by FCS for the indicated protein constructs. Data were fitted using the anomalous diffusion model (solid lines). All ACF curves were normalized to their respective maximal values to facilitate comparison. (**C**) Diffusion times ($\tau_D$) measured for the indicated protein constructs. Each dot in the box plot represents the value for one individual cell, averaged over six consecutive acquisitions (*Figure 1—figure*

*Figure 1 continued on next page*

*Figure 1 continued*

*supplement 3*). The numbers of cells measured for each construct are shown in Appendix 6. ***$p<0.0001$ in a two-tailed heteroscedastistic *t*-test. Exact *p*-valuescan be found in Appendix 5. (**D, E**). Dependence of protein mobility ($1/\tau_D$; **D**) and apparent anomaly of diffusion ($\alpha$; **E**) on molecular mass. Each symbol represents the average value for all individual cells that have been measured for that particular construct and the error bars represent the standard error of the mean. Individual values are shown in *Figure 1—figure supplement 5* and the numbers of measured cells for each construct are shown in Appendix 6. Protein constructs with low mobility for which effects of specific interactions were further investigated are highlighted in color and labeled. The values of $1/\tau_D$ and $\alpha$ for both the original constructs (diamonds) and the constructs where mutations were introduced to disrupt interactions (circles) are shown. For Map, two alternative amino acid substitutions that disrupt its interaction with the ribosome are shown (see *Figure 1—figure supplement 10*). (**F–H**) Cartoons illustrating the cellular interactions that could affect mobility of ClpS (**F**), Map (**G**), and DnaK (**H**). ClpS engages with the ClpAP protease and with substrates, cartoon adapted from Figure 1A from *Román-Hernández et al., 2011*. Map interacts with the actively translating ribosomes, cartoon adapted from Figure 3A from *Sandikci et al., 2013*. DnaK interacts with unfolded client protein. Amino acidic residues that were mutated to disrupt interactions are highlighted (see text for details). FCS, fluorescence correlation spectroscopy.

The online version of this article includes the following source data and figure supplement(s) for figure 1:

**Source data 1.** Individual $\tau_D$ measurements from *Figure 1C*.

**Figure supplement 1.** Expression analysis for all *Escherichia coli* protein constructs made in this study.

**Figure supplement 1—source data 1.** Uncropped western blot images for *Figure 1—figure supplement 1*.

**Figure supplement 2.** Growth curves of *Escherichia coli* strains expressing tested sfGFP-tagged proteins.

**Figure supplement 3.** Workflow of a typical FCS experiment.

**Figure supplement 4.** Comparison between fits of the experimental data with Brownian and anomalous diffusion models.

**Figure supplement 5.** Individual measurements of $\tau_D$ and $\alpha$ for all *Escherichia coli* protein constructs included in the analysis of mass dependence.

**Figure supplement 5—source data 1.** Individual values of $\tau_D$ from *Figure 1—figure supplement 5A*.

**Figure supplement 6.** Comparison between one-component and two-components anomalous diffusion fit for DsdA-sfGFP.

**Figure supplement 6—source data 1.** Individual values used to calculate mean and standard error of the mean values of $\tau_D$ and $\alpha$ values from *Figure 1—figure supplement 6B*.

**Figure supplement 7.** Mobility ($1/\tau_D$) and anomaly of diffusion ($\alpha$) of sfGFP in individual cells with different width and length.

**Figure supplement 7—source data 1.** Individual values of $1/\tau_D$ and measurements of cell length from *Figure 1—figure supplement 7A*.

**Figure supplement 8.** Comparison of protein mobility in cephalexin-treated and untreated cells.

**Figure supplement 8—source data 1.** Individual values of $1/\tau_D$ and measurements of cell length from *Figure 1—figure supplement 8A*.

**Figure supplement 9.** Expression analysis for the mutants with impaired interactions.

**Figure supplement 9—source data 1.** Uncropped western blot image for *Figure 1—figure supplement 9*.

**Figure supplement 10.** Mobility ($1/\tau_D$) and anomaly of diffusion ($\alpha$) of ClpS, Map and DnaK and of indicated mutants with disrupted protein interactions.

**Figure supplement 10—source data 1.** Individual values of $1/\tau_D$ from *Figure 1—figure supplement 10A*.

dependence on the protein size, ranging from 0.8 to 0.86 for most of the constructs (*Figure 1E*). Notably, the few protein constructs with $\alpha$ of ~0.8 or lower were also among the ones with low mobility for their molecular mass (*Figure 1D and E*, colored symbols).

## Macromolecular interactions reduce protein mobility

We reasoned that the main group of constructs that exhibit mobility close to the apparent mass-dependent upper limit represents proteins whose diffusion is only limited by macromolecular crowding, and that the lower $1/\tau_D$ and $\alpha$ of other constructs might be due to their specific interactions with other cellular proteins or protein complexes. Indeed, for three of these proteins (ClpS, Map, and DnaK) such interactions are well characterized and can be specifically disrupted. ClpS is the adaptor protein that delivers degradation substrates to the protease ClpAP (*Román-Hernández et al., 2011*). The substrate-binding site of ClpS is constituted by three amino acid residues (D35, D36, and H66) that interact with the N-terminal degron of target proteins (*Figure 1F*). If these residues are mutated into alanine, substrate binding in vitro is substantially reduced (*Román-Hernández et al., 2011*; *Humbard et al., 2013*). Additionally, ClpS directly docks to the hexameric ClpA. Consistently, we observed that while the stability of the mutant construct ClpS$^{D35A\_D36A\_H66A}$-sfGFP was not affected (*Figure 1—figure supplement 9*), its mobility in a $\Delta clpA$ strain became significantly higher and less anomalous, with both $1/\tau_D$ and $\alpha$ reaching levels similar to those of other proteins of similar mass (*Figure 1D and E* and *Figure 1—figure supplement 10*).

Similar results were obtained for the other two constructs. Map is the methionine aminopeptidase that cleaves the N-terminal methionine from nascent polypeptide chains (*Solbiati et al., 1999*). Map interacts with the negatively charged backbone of ribosomes through four positively charged lysine residues (K211, 218, 224, and 226) located in a loop (*Figure 1G*). If these residues are mutated into alanine, the in vitro affinity of Map for the ribosomes is reduced (*Sandikci et al., 2013*). The mobility of Map-sfGFP was indeed much increased by alanine substitutions at all four lysine sites (*Figure 1D and E* and *Figure 1—figure supplement 9* and *Figure 1—figure supplement 10*). Interestingly, charge inversion of lysines to glutamic acid did not further increase Map-sfGFP mobility as was expected based on in vitro experiments (*Sandikci et al., 2013*).

DnaK is the major bacterial chaperone that binds to short hydrophobic polypeptide sequences, which become exposed during protein synthesis, membrane translocation, or protein unfolding (*Genevaux et al., 2007*). DnaK accommodates its substrate peptides inside a hydrophobic pocket (*Figure 1H*). The substitution of the valine residue 436 with bulkier phenylalanine creates steric hindrance that markedly decreases substrate binding to DnaK in vitro (*Mayer et al., 2000*), and both the $1/\tau_D$ and $\alpha$ of DnaK$^{V436F}$-sfGFP were significantly higher than for the correspondent wild-type construct (*Figure 1D and E* and *Figure 1—figure supplement 9* and *Figure 1—figure supplement 10*). Nevertheless, in this case, the $1/\tau_D$ did not reach the levels of other proteins of similar molecular mass, which is likely explained by multiple interactions of DnaK with other components of the cellular protein quality control machinery besides its binding to substrates (*Kumar and Sourjik, 2012*).

## Apparent anomaly of diffusion could be largely explained by confinement

When FCS measurements are performed in a confined space with dimensions comparable to those of the observation volume, such confinement may affect the apparent mobility of fluorescent molecules (*Gennerich and Schild, 2000*; *Jiang et al., 2020*). To investigate the effect of confinement on our FCS measurements, we performed Brownian dynamics simulations of FCS experiments with particles undergoing three-dimensional, purely Brownian diffusion inside a bacterial cell-like volume (*Figure 2A Inset*; see Materials and methods). For the values of cell diameter commonly observed under our growth conditions, 0.8–0.9 µm, and over a wide range of particle diffusion coefficients, simulated ACFs could be indeed successfully fitted with the anomalous diffusion model, yielding an anomalous diffusion exponent of around 0.8–0.9 (*Figure 2A and B*). This made us hypothesize that the relatively small apparent deviation from Brownian diffusion in the fit, with $\alpha$ between 0.82 and 0.9 common to most constructs, may primarily reflect a confinement-induced effect rather than proper subdiffusion.

In order to estimate what deviation from Brownian diffusion could still be compatible with our experimental data, we performed additional simulations where particles undergo fractional Brownian motion, a particular type of subdiffusion, under cell confinement and for different degrees of ansatz anomaly (*Figure 2—figure supplement 1A*). As in the case of Brownian diffusion under confinement, fitting these ACFs using the anomalous diffusion model yielded values of $\alpha$ that were consistently lower than the ansatz used for simulations (*Figure 2—figure supplement 1B*). The range of fit values observed for experimental data, 0.82–0.9, corresponded to the ansatz values of 0.95–1.0, hence very close to Brownian diffusion.

In apparent agreement with these simulation results, when *E. coli* cell width was increased by treatment with the inhibitor of bacterial cell wall biosynthesis A22 (*Ouzounov et al., 2016*; *Figure 2C*), in addition to the standard cephalexin-induced elongation, the anomalous diffusion exponent of sfGFP (*Figure 2D*) also significantly increased. A small, but significant increase in protein mobility was also observed (*Figure 2—figure supplement 2*). Since it was previously reported that treatment with A22 can reduce dry-mass density of *E. coli* cells (*Oldewurtel et al., 2021*), we further performed a cell sedimentation assay (*Figure 2—figure supplement 3A–C*). The treatment with cephalexin slightly, by 1 g/L, that is <0.1% of *E. coli* volumetric mass density 1.11 kg/L (*Martínez-Salas et al., 1981*), decreased the density of *E. coli* cells in this assay. The additional treatment with A22, in our growth conditions, had only minor and not significant impact, once the effect of the A22-induced cell volume increase on sedimentation was accounted for (*Figure 2—figure supplement 3H–J*). We thus conclude that the influence of A22 on the anomaly of protein diffusion is most likely due to its effect on cell width and not on the cytoplasmic density.

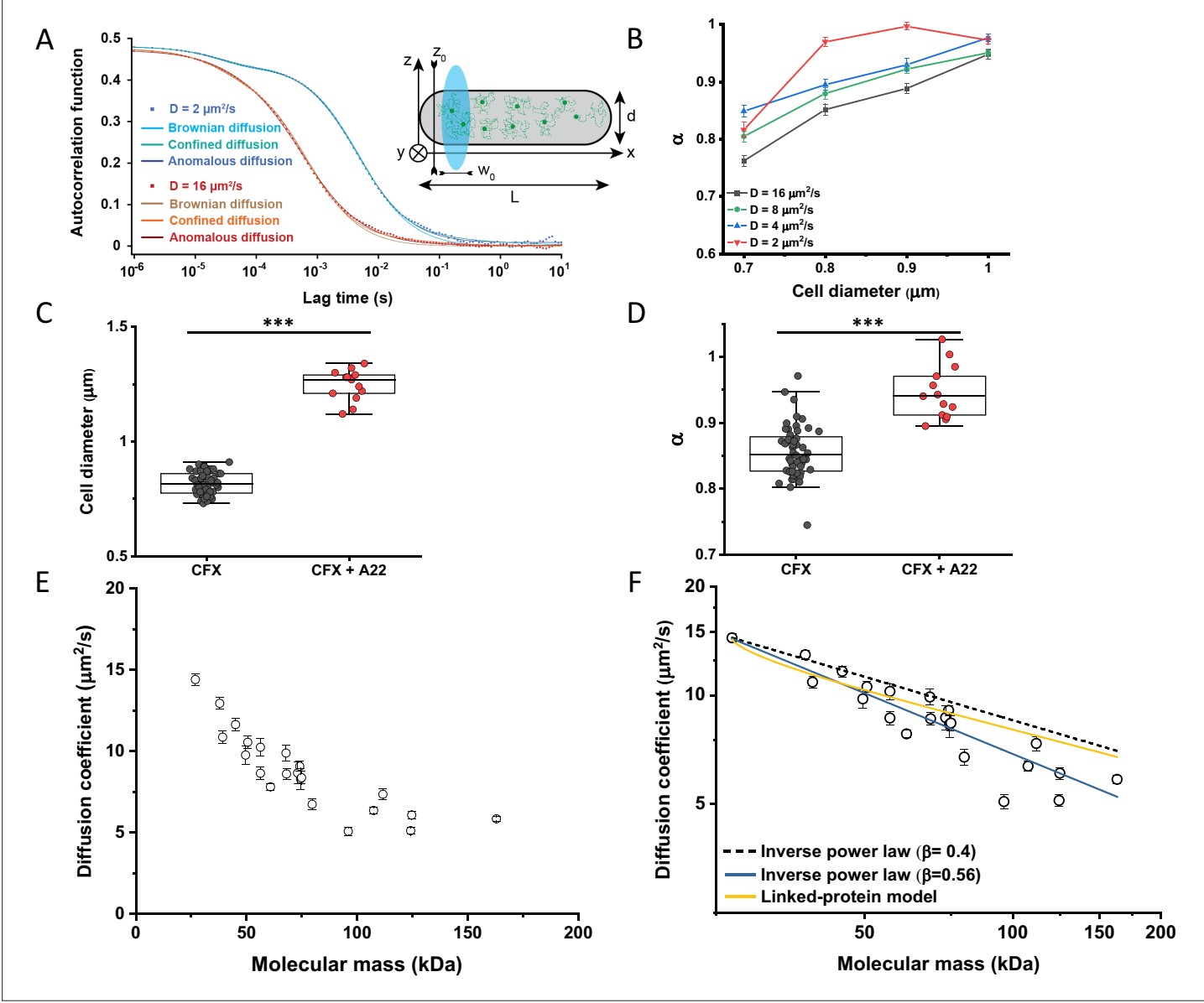

**Figure 2.** Protein diffusion in bacterial cytoplasm corrected for confinement. (**A**) Representative ACFs of simulated fluorescence intensity fluctuations. Simulations were performed in a confined geometry of a cell with indicated length $L$ and diameter $d$, and dimensions of the measurement volume $\omega_0$ and $z_0$, representing an experimental FCS measurement (*Inset*; see Materials and methods) for two different values of the ansatz diffusion coefficient. Solid lines are fits by the models of unconfined Brownian diffusion, anomalous diffusion and by the Ornstein-Uhlenbeck (OU) model of Brownian diffusion under confinement, as indicated. (**B**) The exponent $\alpha$ extracted from the fit of the anomalous diffusion model to the ACFs data that were simulated at different values of the cell diameter. Corresponding values of the diffusion coefficient are shown in *Figure 2—figure supplement 7*. (**C, D**) *Escherichia coli* cells treated with cephalexin alone or with cephalexin and 1 µg/ml of A22 (see Materials and methods), show A22-dependent increase in the measured cell diameter (**C**) and higher values of the exponent $\alpha$ extracted from the fit to the ACF measurements (**D**). The numbers of cells measured for each condition are shown in Appendix 6. ***$p<0.0001$ in a two-tailed heteroscedastistic *t*-test. Exact *p*-values can be found in Appendix 5. (**E**) Dependence of the diffusion coefficient calculated from fitting the experimental ACFs with the OU model of confined diffusion. Only the subset of apparently freely diffusing constructs from *Figure 1D* has been analyzed with the OU model (see also *Table 1*). Each circle represents the average value for all individual cells that have been measured for that particular construct (Appendix 6), and the error bars represent the standard error of the mean. Error bars that are not visible are smaller than the symbol size. (**F**) Fit of the mass dependence with an inverse power law (solid blue line, exponent $\beta=0.56\pm0.05$), and predictions of the Stokes-Einstein relation (black dashed line) and of the model describing diffusion of two linked globular proteins (solid yellow line), both with exponent $\beta=0.4$. ACF, autocorrelation function; FCS, fluorescence correlation spectroscopy.

The online version of this article includes the following source data and figure supplement(s) for figure 2:

**Source data 1.** Average and error from each simulation in *Figure 2B*.

*Figure 2 continued*

**Figure supplement 1.** Simulations of particles undergoing fractional Brownian motion.

**Figure supplement 1—source data 1.** Mean values and standard errors of the mean from *Figure 2—figure supplement 1B*.

**Figure supplement 2.** Mobility of sfGFP in cells treated with cephalexin (CFX) or the combination of cephalexin and A22.

**Figure supplement 2—source data 1.** Individual values of $1/\tau_D$ from *Figure 2—figure supplement 2*.

**Figure supplement 3.** Sedimentation assay of cellular density for indicated treatments.

**Figure supplement 3—source data 1.** Mean values and standard errors of the mean from *Figure 2—figure supplement 3H*.

**Figure supplement 4.** Apparent anomaly of diffusion and residence time for different pinhole sizes.

**Figure supplement 4—source data 1.** Individual values of $\alpha$ from *Figure 2—figure supplement 4A*.

**Figure supplement 5.** Apparent anomaly of diffusion (α) extracted from analysis of ACFs at shorter time scales.

**Figure supplement 5—source data 1.** Average values and standard errors of the mean of $\alpha$ from *Figure 2—figure supplement 5*.

**Figure supplement 6.** Residuals of fitting the simulated ACFs with different models.

**Figure supplement 7.** Diffusion coefficients fitted from simulation data.

**Figure supplement 7—source data 1.** Average and error from simulation in *Figure 2—figure supplement 7A*.

**Figure supplement 8.** Comparison between fits of the experimental data with confined diffusion and anomalous diffusion models.

To additionally test our conclusion that the reduced value of $\alpha$ is due to confinement by the cell width, we performed FCS measurements for sfGFP, DnaK-sfGFP, and AcnA-sfGFP on a smaller confocal volume, thus limiting the analysis to fluorophores diffusing at a distance from the cell boundary, by reducing the pinhole size to a less optimal but smaller value of 0.66 Airy units. Consistent with our expectation, the value of $\alpha$ derived from these measurements was significantly higher, >0.9, for sfGFP and AcnA-sfGFP (*Figure 2—figure supplement 4A*). The residence time ($\tau_D$) of proteins in a smaller confocal volume was slightly reduced, too (*Figure 2—figure supplement 4B*). In contrast, the anomalous diffusion exponent of DnaK-sfGFP remained low even when measured away from the cell boundary, confirming that its motion is truly subdiffusive due to interactions with other proteins. Similar conclusions could be drawn when the FCS data obtained with the regular pinhole size were fitted only for short lag times, which also reduces the impact of confinement, although such analysis is not common for FCS experiments. The apparent anomaly of diffusion showed clear increase for shorter lag times for all constructs, remaining below 0.9 only for DnaK-sfGFP but not for its non-interacting variant (*Figure 2—figure supplement 5*).

We next derived an Ornstein-Uhlenbeck (OU) model for fitting FCS data, where the confinement of Brownian diffusing fluorescent particles within the width of the cell is approximated by trapping in a harmonic potential of the same width (Appendix 3). The anomalous diffusion and OU models fit the ACF of the Brownian dynamic simulations comparably well and better than the model of unconfined Brownian diffusion (*Figure 2A* and *Figure 2—figure supplement 6*), with the OU model having one less free parameter than the anomalous diffusion model. The OU model directly estimates the ansatz diffusion coefficient with ±5% accuracy for the typical cell widths observed in our experiments (*Figure 2—figure supplement 7*).

Since the OU model proved accurate in fitting the experimental data, comparably to the anomalous diffusion model (*Figure 2—figure supplement 8*), we used it to re-fit the ACF data for all faster-diffusing constructs and to estimate their Brownian diffusion coefficients (*Figure 2E* and *Table 1*). The dependence of $D$ on molecular mass for this set of constructs was scaling as $(MM)^{-\beta}$ with $\beta=0.56\pm0.05$ (*Figure 2F*, solid blue line), less steep compared to the previous estimates (*Kumar et al., 2010*; *Mika et al., 2010*; *Stracy et al., 2021*) but still steeper than expected from the Stokes-Einstein relation, even when assuming $\beta=0.4$ for not perfectly globular proteins (*Figure 2F*, black dashed line) (*Enright and Leitner, 2005*; *Smilgies and Folta-Stogniew, 2015*). In order to elucidate whether part of this residual deviation may be accounted for by the specific shape of fusion constructs, where sfGFP is fused to the differently sized target proteins by a short flexible linker, we further applied a previously derived model describing diffusion of such linked proteins (Appendix 4) (*Agudo-Canalejo and Golestanian, 2020*). The dependence of $D$ on molecular mass predicted by this linked-protein model seems indeed to better recapitulate our experimental data, particularly for smaller protein fusions (*Figure 2F*, solid yellow line), although it moderately overestimates $D$ for several of the largest protein fusions (>100 kDa). Thus, we conclude that the size dependence of diffusion for the majority

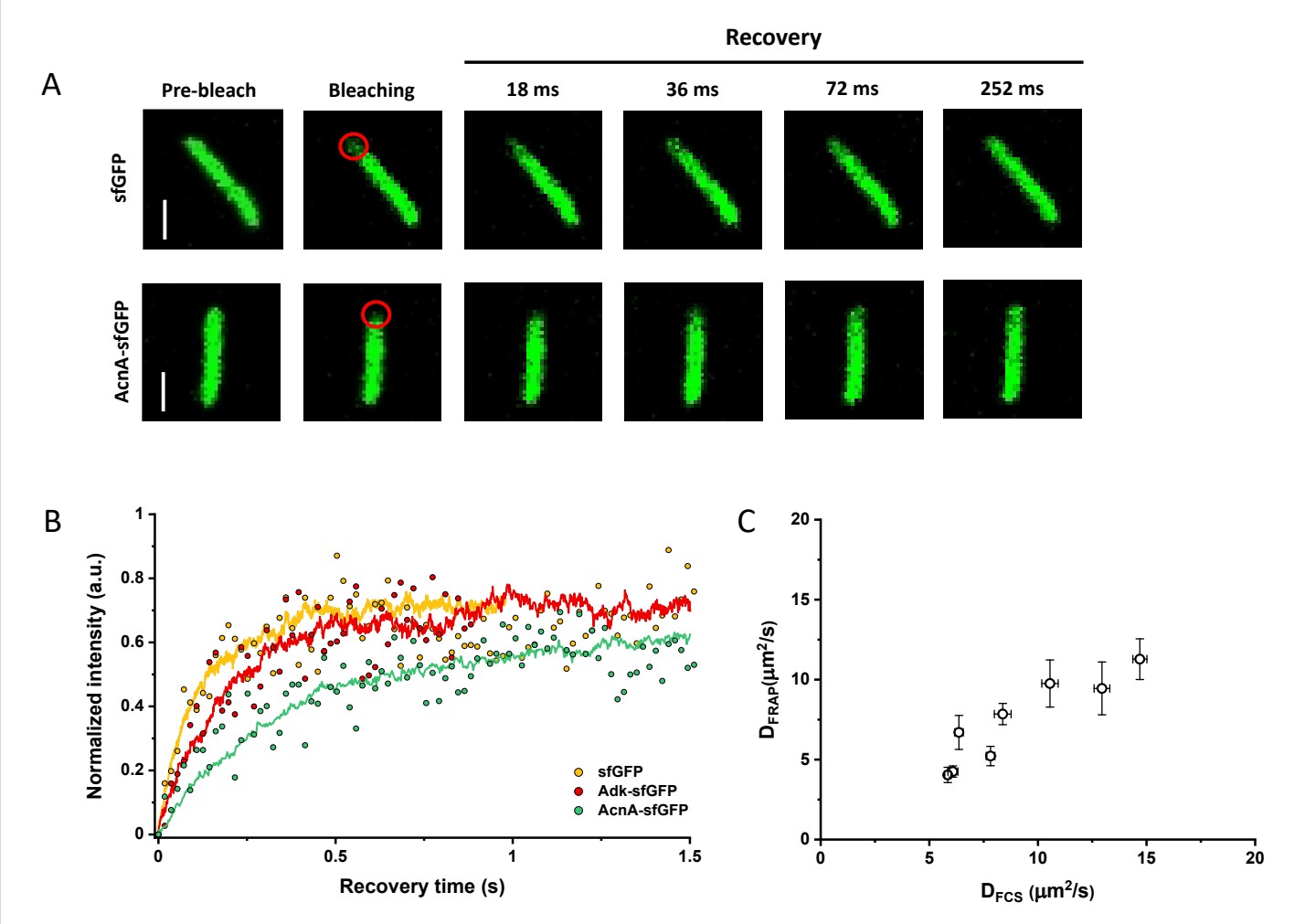

**Figure 3.** Comparison between protein diffusion coefficients measured by FCS and FRAP. (**A**) Examples of FRAP measurements for two different constructs, sfGFP and AcnA-sfGFP. A 3×3 pixels area close to one cell pole (red circle) was photobleached with a high-intensity laser illumination for 48 ms and the recovery of fluorescence in the bleached area was monitored for 11 s with the time resolution of 18 ms. The scales bars are 2 µm. (**B**) Representative curves of fluorescence recovery in FRAP experiments and their fitting using simFRAP. The experimental data (colored dots) are used by the simFRAP algorithm to simulate the underlying diffusional process (colored lines). The simulation is then used to compute the diffusion coefficient. The simulation proceeds until the recovery curve reaches a plateau, therefore it is interrupted at a different time for each curve. (**C**) Correlation between the diffusion coefficients measured in FCS experiments ($D_{FCS}$, fitting with the OU model; data from *Figure 2E*) and in FRAP experiment ($D_{FRAP}$, fitting with simFRAP). The numbers of cells measured for each construct with each technique are shown in Appendix 6. Error bars represent the standard error of the mean. Error bars that are not visible are smaller than the symbol size. FCS, fluorescence correlation spectroscopy; FRAP, fluorescence recovery after photobleaching; OU, Ornstein-Uhlenbeck.

The online version of this article includes the following source data for figure 3:

**Source data 1.** Individual mean and standard error of the mean of diffusion coefficient values from *Figure 3C*.

of cytoplasmic proteins follows the Stokes-Einstein relation, once the shape of the sfGFP-tagged protein constructs is taken into account.

## Protein diffusion coefficients measured using FRAP or FCS are consistent

Since many previous measurements of protein diffusion in bacteria were performed using FRAP, we aimed to directly compare the results of FRAP and FCS measurements for a set of constructs of different mass. Importantly, we used the same growth conditions and microscopy sample preparation protocols as for the FCS experiments. The cells were photobleached in a region close to the pole, similar to the position that was used for the FCS experiment. The recovery of fluorescence was

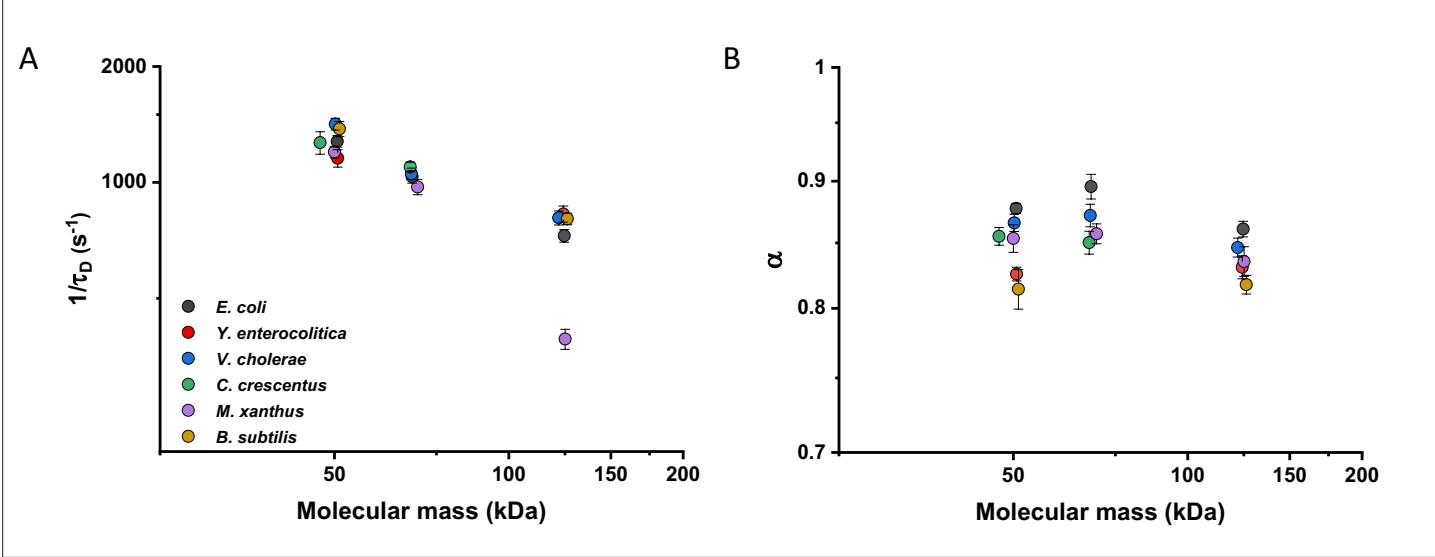

**Figure 4.** Mobility of homologous proteins from other bacterial species in *Escherichia coli*. Mass dependence of protein mobility ($1/\tau_D$; **A**) and anomaly of diffusion ($\alpha$; **B**) of sfGFP fusions to homologues of Adk, Pgk, and AcnA from indicated bacterial species (*E.c.* = *Escherichia coli*; *Y.e.* = *Yersinia enterocolitica*; *V.c.* = *Vibrio cholerae*; *C.c.* = *Caulobacter crescentus*; *M.x.* = *Myxococcus xanthus*; *B.s.* = *Bacillus subtilis*) compared with that of their counterpart from *E. coli*. Each symbol represents the average value for all individual cells that have been measured for that construct and the error bars represent the standard error of the mean. Error bars that are not visible are smaller than the symbol size. The numbers of cells measured for each construct are shown in Appendix 6.

The online version of this article includes the following source data and figure supplement(s) for figure 4:

**Source data 1.** Individual mean and standard error of the mean of $1/\tau_D$ values from *Figure 4A*.

**Figure supplement 1.** Mobility of homologous proteins from other bacterial species in *Escherichia coli*.

**Figure supplement 1—source data 1.** Individual values of $1/\tau_D$ from *Figure 4—figure supplement 1A*.

then followed for 11 s with the time resolution of 18 ms (*Figure 3A*). The diffusion coefficients were computed from the time course of recovery with the plugin for ImageJ, simFRAP (*Blumenthal et al., 2015*), which utilizes a simulation-based approach (*Figure 3B*). We observed very good correlation between both values of diffusion coefficients, although for most constructs the diffusion coefficients determined by FRAP were 5–30% lower than those obtained from the FCS data (*Figure 3C* and *Table 1*).

## Diffusive properties of cytoplasmic proteins are largely conserved between bacterial species

We then investigated whether sfGFP fusions to non-native proteins, originating from other bacteria, may show different diffusive properties in *E. coli* cytoplasm than their native counterparts. The existence of an organism-dependent 'quinary' code of unspecific, short living interactions have been recently proposed in order to explain the reduced mobility of heterologous human proteins in *E. coli* cytoplasm (*Mu et al., 2017*). Thus, we investigated the mobility of proteins from other Gram-negative proteobacteria *Yersinia enterocolitica*, *Vibrio cholerae*, *Caulobacter crescentus*, and *Myxococcus xanthus* and from the Gram-positive bacterium *Bacillus subtilis* that are homologous to several analyzed freely diffusing *E. coli* protein constructs. Within this set of constructs, we observed no significant differences of their $1/\tau_D$ values from *E. coli* homologues. An exception was AcnA from *M. xanthus* (*Figure 4A* and *Figure 4—figure supplement 1A*), whose lower mobility might be a sign of its multimerization, although cellular distribution of this construct was uniform. In contrast, all constructs showed slight but mostly significantly increased anomaly of diffusion compared to *E. coli* proteins (*Figure 4B* and *Figure 4—figure supplement 1B*), which might reflect the weakly increased propensity of non-native proteins to engage in unspecific interactions in *E. coli* cytoplasm.

## Effects of osmolarity, temperature, antibiotics, and cell growth on mobility of differently sized proteins

We further characterized the impact of several environmental and cellular perturbations of the bacterial cytoplasm on protein mobility, using apparently freely diffusing protein fusions of different sizes as probes. We started by confirming the previously characterized decrease in mobility of GFP and large protein complexes or aggregates upon osmotic upshift (*Konopka et al., 2006*; *Konopka et al., 2009*; *Mika et al., 2010*; *Liu et al., 2019*; *Wlodarski et al., 2020*). *E. coli* cells exposed to increased ionic strength by the addition of 100 mM NaCl showed decrease in cell length and width (*Figure 5—figure supplement 1A,B*) and an increase in cell density in the sedimentation assay (*Figure 2—figure supplement 3D*), consistent with a previous report (*Wlodarski et al., 2020*). Higher ionic strength also significantly decreased the mobility of sfGFP (*Figure 5A* and *Figure 5—figure supplement 2A*), comparably to previously measured values (*Konopka et al., 2009*; *Mika et al., 2010*). Importantly, the mobility of all other tested constructs decreased proportionally (*Figure 5A*), meaning that—in this range of molecular sizes—the effect of a moderate osmotic upshift can be interpreted as a simple increase in cytoplasmic viscosity due to higher molecular crowding, which is in contrast to the different effects of high osmolarity on small molecules and on GFP (*Mika et al., 2010*). No effect was observed on the anomaly of diffusion for any protein construct (*Figure 5—figure supplement 3A*).

Next, we studied the effect of environmental temperature on cytoplasmic protein mobility. According to the Stokes-Einstein equation, the diffusion of a particle directly depends on the system's temperature in Kelvin and on the viscosity of the fluid, which itself changes with temperature. In the biologically relevant range, the temperature sensitivity of diffusion is primarily determined by the temperature dependence of water viscosity. The measured increase in mobility of sfGFP and two other constructs, by approximately 20–25% between 25°C and 35°C (*Figure 5B* and *Figure 5—figure supplement 2B*), agrees well with the temperature-dependent decrease in water viscosity over 10 °C (*Huber, 2009*). Expectedly, the effect of imaging temperature was not linked to any changes of the cell size (*Figure 5—figure supplement 1C, D*). Of note, a weak, but consistent, increase in the anomaly of protein diffusion was also observed at higher environmental temperature (*Figure 5—figure supplement 3*). Surprisingly, the growth temperature of the *E. coli* culture had no apparent effect on protein mobility (*Figure 5—figure supplement 4*), suggesting that—at least in the tested temperature range—*E. coli* lacks the growth-temperature dependent regulation of cytoplasmic viscosity that has been recently reported in the budding yeast (*Persson et al., 2020*).

Antibiotics that inhibit transcription (e.g., rifampicin) or translation (e.g., chloramphenicol) are known to affect the spatial organization of bacterial chromosomes (*Bakshi et al., 2014*). The mobility of chromosomal loci and of large cytoplasmic aggregates was also shown to be affected by several antibiotics, in apparent correlation with changes in the cytoplasmic density (*Wlodarski et al., 2020*). We observed that chloramphenicol treatment caused a minor increase in cell width (*Figure 5—figure supplement 1F*) and a decrease in cell density (*Figure 2—figure supplement 3G*). However, protein mobility rather decreased in chloramphenicol-treated cells, opposite to what could be expected based alone on the chloramphenicol-induced reduction of cell density (*Figure 5C* and *Figure 5—figure supplement 2C*). The reduced protein mobility could neither be simply explained by compaction of the nucleoid in cells treated with chloramphenicol, since it was only marginally lower inside than outside of the nucleoid (*Figure 5—figure supplement 5A, B*). It should be noted that no significant difference in the anomaly of diffusion (*Figure 5—figure supplement 5C*) was observed inside or outside of the nucleoid.

In contrast, inhibition of RNA transcription by rifampicin treatment led to a marked increase in protein mobility (*Figure 5C* and *Figure 5—figure supplement 2C*). Such higher protein mobility is consistent with the previously reported rifampicin-induced reduction of macromolecular crowding in bacterial cytoplasm (*Wlodarski et al., 2020*), although only a minor decrease in cell density was observed in our sedimentation assay (*Figure 2—figure supplement 3F*) beyond the effect of DMSO that was used as a solvent for rifampicin (*Figure 2—figure supplement 3E*). Similar to the effects of osmolarity and temperature, the increase in protein mobility caused by the rifampicin treatment, and its decrease induced by chloramphenicol were similar for all tested proteins (*Figure 5C*), except for the AcnA-sfGFP construct that was disproportionally affected by chloramphenicol in both mobility and anomaly of diffusion (*Figure 5—figure supplement 3C*).

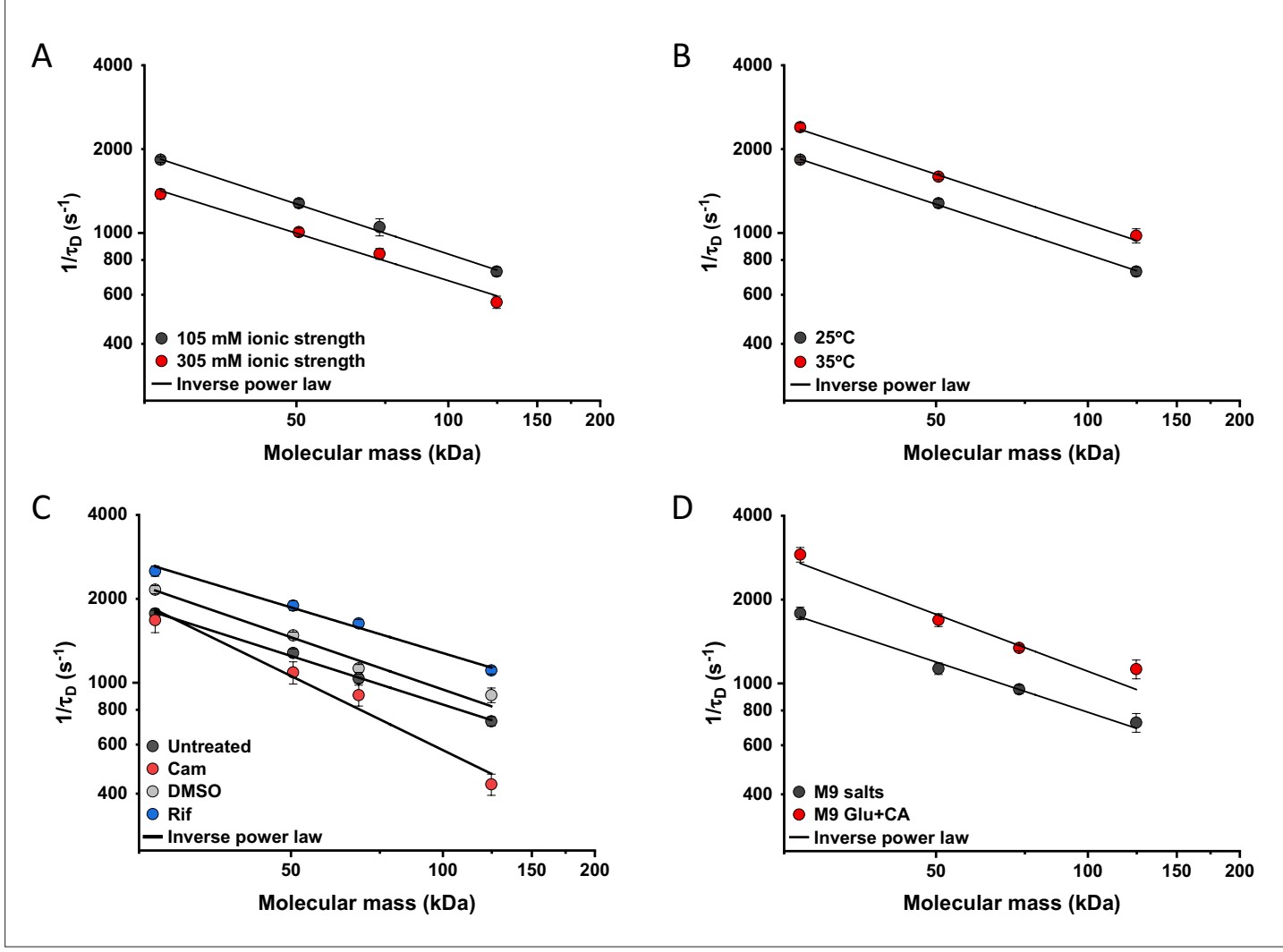

**Figure 5.** Effects of physicochemical perturbations and cell growth on mobility of differently sized proteins. Each dot represents the average value of protein mobility ($1/\tau_D$) of all the cells measured for the construct of the indicated molecular mass . The numbers of cells measured for each construct in each condition are shown in Appendix 6. Error bars represent the standard error. Error bars that are not visible are smaller than the symbol size. The solid black lines are the fit with an inverse power law to extract the size dependence of protein mobility ($\beta$) in that condition. (**A**) Protein mobility measured in cells that were resuspended in either tethering buffer (ionic strength of 105 mM; $\beta=0.60\pm0.01$) or in the same buffer but supplemented with additional 100 mM NaCl (total ionic strength of 305 mM; $\beta=0.57\pm0.05$). The measurements were performed in agarose pads prepared at the same ionic strength. (**B**) Protein mobility at different environmental temperatures. As for the other experiments, *Escherichia coli* cultures were grown at 37°C and bacterial cells during the measurements were incubated at 25°C ($\beta=0.60\pm0.01$) or at 35°C ($\beta=0.60\pm0.05$), as indicated. (**C**) Protein mobility in control cells ($\beta=0.58\pm0.02$) and after treatment with chloramphenicol (Cam; 200 µg/ml; $\beta=0.88\pm0.11$), rifampicin (Rif; 200 µg/ml, in 0.1% v/v DMSO; $\beta=0.54\pm0.04$), or DMSO control (0.1% v/v; $\beta=0.62\pm0.07$) as indicated. Antibiotics were added to growing *E. coli* culture 60 min prior to harvesting. (**D**) Protein mobility in non-growing cells incubated at 35°C on agarose pads containing only M9 salts ($\beta=0.60\pm0.05$) in comparison with growing cell incubated on pads with M9 salts supplemented with 20 mM glucose and 0.2% casamino acids (Glu+CA; $\beta=0.68\pm0.10$).

The online version of this article includes the following source data and figure supplement(s) for figure 5:

**Source data 1.** Individual mean and standard error of the mean of $1/\tau_D$ values from **Figure 5A**.

**Figure supplement 1.** Mobility of sfGFP as a function of length and width of individual cells upon indicated perturbations.

**Figure supplement 1—source data 1.** Individual measurements of cell length from **Figure 5—figure supplement 1A**.

**Figure supplement 2.** Effect of different perturbations on protein mobility ($1/\tau_D$) in individual cells.

**Figure supplement 2—source data 1.** Individual values of $1/\tau_D$ from **Figure 5—figure supplement 2A**.

**Figure supplement 3.** Effect of different perturbations on the anomaly of protein diffusion ($\alpha$) in individual cells.

**Figure supplement 3—source data 1.** Individual values of $\alpha$ from **Figure 5—figure supplement 3A**.

*Figure 5 continued on next page*

*Figure 5 continued*

**Figure supplement 4.** Effect of growth and measurement temperature on protein diffusion.

**Figure supplement 4—source data 1.** Individual values of $1/\tau_D$ from *Figure 5—figure supplement 4A*.

**Figure supplement 5.** Influence of nucleoid on protein mobility.

**Figure supplement 5—source data 1.** Individual values of $1/\tau_D$ from *Figure 5—figure supplement 5B*.

**Figure supplement 6.** Effect of nutrient availability and growth on protein mobility.

**Figure supplement 6—source data 1.** Individual values of $1/\tau_D$ from *Figure 5—figure supplement 6A*.

**Figure supplement 7.** Effect of DNP on protein mobility.

**Figure supplement 7—source data 1.** Individual values of $1/\tau_D$ from *Figure 5—figure supplement 7A*.

Finally, we investigated whether protein mobility might be influenced by cell growth, comparing FCS measurements in cells incubated at 35°C on agarose pads containing either only M9 salts or M9 salts plus glucose and casamino acids. These conditions had only minor impact on the cell shape (*Figure 5—figure supplement 1G, H*). Although at this high-temperature residual growth was also observed for cells on M9 salt pads, cell growth in presence of nutrients was expectedly much more pronounced. The observed protein mobility was also much higher in the presence of nutrients, and this increase was again similar for the four tested differently sized constructs (*Figure 5D* and *Figure 5—figure supplement 2D*), while no consistent trend was observed in the anomaly of protein diffusion across these conditions (*Figure 5—figure supplement 3D*). To further distinguish the respective contributions of metabolic activity and of biosynthesis and resulting cell growth, we incubated cells in presence of both nutrients and chloramphenicol on the agarose pad. Similar to our previous experiments where chloramphenicol was added to the batch culture, its addition had no or little effect on the mobility of sfGFP or the AcnA-sfGFP construct in absence of nutrients (*Figure 5—figure supplement 6*). In contrast, protein mobility in presence of nutrients was strongly affected by chloramphenicol treatment. Thus, the enhanced protein mobility in presence of nutrients appears to be primarily due to active protein production and cell growth. Nevertheless, even chloramphenicol-treated cells exhibited a moderate increase in protein mobility in presence of nutrients, indicating that the metabolic activity contributes to the overall effect of growth on diffusion. It is possible that the contribution of the metabolic activity might be even larger, since the inhibition of protein translation might in turn reduce metabolic activity. In any case, the impact of growth on diffusion of individual proteins cannot be simply explained by the energy state of the cell, since lowering it by the inhibition of respiration-dependent ATP synthesis using treatment with dinitrophenol (DNP) did not reduce protein mobility, at either 25°C or 35°C. This is contrary to the effect of the DNP treatment on large cytoplasmic particles (*Parry et al., 2014*; *Figure 5—figure supplement 7*). An interesting exception was the mobility of Adk-sfGFP, which was indeed reduced by the DNP treatment at high temperature. This, however, might be a specific effect related to the enzymatic activity or conformation of Adk that binds ATP as a substrate.

## Discussion

Bacteria rely on translational diffusion to deliver proteins and other macromolecules to their cellular destinations, including their reaction partners, and the diffusional properties of bacterial cytoplasm are therefore fundamental to the understanding of bacterial cell biology. Consequently, a number of studies have investigated protein mobility in bacteria, all showing strong effects of macromolecular crowding in the bacterial cytoplasm on diffusion (*Konopka et al., 2006*; *Mullineaux et al., 2006*; *Konopka et al., 2009*; *Kumar et al., 2010*; *Mika et al., 2010*; *Nenninger et al., 2010*). Nevertheless, the relatively small number of proteins investigated in each of these previous studies, and the differences between strains, growth conditions and between methodologies, limited general conclusions about protein mobility, even in the most-studied environment of *E. coli* cytoplasm. For example, combining data from different studies to determine the relation between the size of a protein and its cytoplasmic diffusion coefficient yielded only uncertain estimates (*Mika and Poolman, 2011*; *Schavemaker et al., 2018*). Such variability between different studies might be further compounded by potentially profound effects on diffusion of size-independent protein properties such as surface charge (*Schavemaker et al., 2017*) or weak interactions with other proteins and other cellular components

(*von Bülow et al., 2019*). Similarly, it remains unclear whether a typical protein in the cytoplasm exhibits Brownian diffusion, as has been shown in few examples (*Bakshi et al., 2011*; *English et al., 2011*), or rather a subdiffusive behavior as common in eukaryotic cells (*Di Rienzo et al., 2014*; *Sabri et al., 2020*) and for large proteins and nucleoprotein particles in bacteria (*Golding and Cox, 2004*; *Lampo et al., 2017*; *Yu et al., 2018*). Additionally, while in eukaryotic cells, anomalous diffusion is primarily associated with hindrance by intracellular structures, the possible causes of anomalous diffusion in bacteria are still unclear.

Here, we addressed these questions by systematically investigating the diffusive behavior of a large set of fluorescent protein fusions to differently sized cytoplasmic proteins of *E. coli*. We demonstrate that the majority of studied proteins exhibit a rather uniform relation between their molecular mass and cytoplasmic mobility, with a clear upper bound on protein mobility at a given molecular mass. This bound likely reflects the fundamental size-specific physical limit on protein diffusion in *E. coli* cytoplasm, with lower mobility of individual proteins being due to their interactions with other cellular components.

Furthermore, our simulations suggest that the apparent weak anomaly of diffusion observed in the FCS data analysis could be largely accounted for by confinement of the otherwise purely Brownian diffusing particles. In the small volume of a bacterial cell, the anomalous diffusion exponent $\alpha \sim 0.82$–$0.9$, as experimentally observed for most proteins, is expected to correspond to $\alpha \sim 0.95$–$1.0$ of the unconfined diffusion, and hence very close to Brownian. This explanation is further supported by our measurements of diffusion in A22-treated *E. coli* with an increased cell width, and thus reduced confinement, which yielded significantly higher values of $\alpha$. Although the interpretation of these experiments might be complicated by the reduced cytoplasmic density of A22-treated bacteria (*Oldewurtel et al., 2021*), under our conditions the effect of A22 on cell density seems to be negligible. Higher values of $\alpha$ were also observed when the FCS measurements were performed using smaller confocal volume, as could be expected from protein diffusion away from the cell boundary. Thus, we conclude that the diffusion of most proteins in the bacterial cytoplasm shows little if any deviation from Brownian within the precision of our experiments, although some residual anomaly cannot be excluded. Notably, similar conclusions have been drawn by previous SPT studies for several proteins (*Bakshi et al., 2011*; *English et al., 2011*).

We therefore used a model of purely Brownian diffusion under confinement (OU model) to determine diffusion coefficients by directly fitting the ACFs of our FCS measurements. The obtained overall dependence of diffusion coefficients on the molecular mass of the fusion protein showed the exponent $\beta=0.56$, steeper than predicted by the Stokes-Einstein relation, with $\beta=0.33$ for fully compact proteins or $\beta=0.4$ for the more realistic case where proteins are assumed to be not entirely compact (*Enright and Leitner, 2005*; *Smilgies and Folta-Stogniew, 2015*). Nevertheless, at least for smaller constructs, the observed dependence of the diffusion coefficient on the molecular mass could be well reproduced once the specific shape of fusion constructs, where two roughly globular proteins are fused by a short linker, was taken into account along with their imperfect globularity (*Agudo-Canalejo and Golestanian, 2020*). Only largest proteins in our set (above 100 kDa) showed mobility that was slower than predicted by this model, possibly because diffusion of larger proteins is more strongly impacted by weak interactions with other macromolecules (*von Bülow et al., 2019*).

Our analysis thus suggests that, despite the high crowdedness of the bacterial cytoplasm, the diffusion of typical cytoplasmic proteins in bacteria is mostly Brownian and can be well described by treating the cytoplasm as a viscous fluid, with only a moderate dependence of the effective viscosity on the size of diffusing proteins. Given the diffusion coefficient determined in our study for free sfGFP, $\sim 14$ $\mu m^2$ $s^{-1}$, for small proteins this effective viscosity of bacterial cytoplasm is only approximately six times higher than in dilute solution (*Potma et al., 2001*). This diffusion coefficient for GFP is substantially larger than the values reported in the early studies that used FRAP (*Elowitz et al., 1999*; *Konopka et al., 2006*; *Mullineaux et al., 2006*; *Konopka et al., 2009*; *Kumar et al., 2010*; *Mika et al., 2010*; *Nenninger et al., 2010*), although it is consistent with other FCS and SPT studies (*Meacci et al., 2006*; *English et al., 2011*; *Sanamrad et al., 2014*; *Diepold et al., 2017*; *Rocha et al., 2019*). These differences are apparently due to the limitations of early FRAP analyses that generally underestimated protein mobility, rather than due to different spatial and temporal scales assessed by the two techniques, since our direct comparison between FCS and FRAP measurements yielded

similar values of diffusion coefficients. Indeed, a more recent FRAP study also reported higher diffusion coefficients for GFP (*Schavemaker et al., 2017*).

Several proteins in our set showed much lower mobility than expected from their size, and in some cases also clearly subdiffusive behavior. For three selected examples, this deviation could be explained by specific association with other proteins or multiprotein complexes, since disrupting these interactions both increased protein mobility and reduced subdiffusion. This is consistent with theoretical studies suggesting that binding of diffusing molecules to crowders can lead to subdiffusion (*Saxton, 2007*; *Guigas and Weiss, 2008*). Thus, protein-protein interactions may be the main cause of protein subdiffusion in bacterial cytoplasm, although other explanations might hold for subdiffusion of large cytoplasmic particles (*Golding and Cox, 2004*; *Lampo et al., 2017*; *Yu et al., 2018*).

Unspecific transient interactions might also explain the slightly subdiffusive behavior of sfGFP fusions to proteins from other bacteria in *E. coli* cytoplasm. However, this anomaly was weak and there was overall only little difference between the mobility of these non-native proteins and their similarly sized *E. coli* homologues, which is in contrast to pronounced differences observed between bacterial and mammalian proteins (*Mu et al., 2017*). Thus, there is apparently little organism-specific adaptation of freely diffusing proteins to their 'bacterial host,' with a possible exception of bacteria with extreme pH or ionic strength of the cytoplasm (*Schavemaker et al., 2017*). This might facilitate horizontal gene transfer among bacteria, by ensuring that their surface properties do not hinder accommodation of proteins in a new host.

We further probed how the effective viscous properties of bacterial cytoplasm changed under different physicochemical perturbations, using a subset of proteins that showed highest mobility for their molecular mass as reporters of unhindered diffusion. Consistent with the importance of macromolecular crowding and in agreement with previous results (*Konopka et al., 2009*), protein mobility decreased upon osmotic upshift as cytoplasmic crowding increases. In contrast, the effective cytoplasmic viscosity decreases significantly (~20%) upon treatment with rifampicin that inhibits transcription and thereby reduces the overall macromolecular crowding. This observation is consistent with recent SPT measurements on large cytoplasmic particles (*Wlodarski et al., 2020*; *Rotter et al., 2021*), and it agrees well with the relative contribution of RNA to the macromolecular composition of an *E. coli* cell (*Cayley et al., 1991*) and with the reduction of molecular crowding in rifampicin-treated cells (*Wlodarski et al., 2020*).

Despite multiple effects of environmental temperature on cellular processes, such as the active (nonthermal) stirring of the cytoplasm at higher temperature (*Weber et al., 2012*), the temperature dependence of the cytoplasmic viscosity in the tested range was similar to that of water and consistent with the Stokes-Einstein relation, decreasing by 20–30% for a temperature increase of 10°C (*Huber, 2009*). Furthermore, the same temperature dependence of protein mobility was observed upon treatment with the protonophore DNP that de-energizes cells by dissipating proton gradient, arguing against general active stirring of cytoplasm in *E. coli* under our experimental conditions. We further observed no dependence of the effective cytoplasmic viscosity on growth temperature, in contrast to the homeostatic adaptation of bacterial membrane fluidity (*Sinensky, 1974*) and of bacterial signaling (*Oleksiuk et al., 2011*; *Almblad et al., 2021*) to the growth temperature. Since growth-temperature-dependent adaptation of the cytosolic viscosity was recently reported for budding yeast (*Persson et al., 2020*), it is surprising that such compensation apparently does not exist in *E. coli*. One possible explanation for this difference might be a broader range of growth temperatures for budding yeast *Saccharomyces cerevisiae* compared to *E. coli*, and a stronger temperature effect on protein diffusion in the yeast cytosol. Of note, here we did not explore protein diffusion in thermally stressed *E. coli* cells, which might have more profound effects on the properties of bacterial cytoplasm as recently shown for *Listeria monocytogenes* (*Tran et al., 2021*).

Finally, we observed that protein mobility was significantly higher in rapidly growing cells. This 'fluidizing' effect of growth seems to be primarily due to the biosynthetic processes, likely protein translation, as evidenced by the reduced mobility upon chloramphenicol treatment, or to cell growth itself. The contribution of metabolic activity in presence of nutrients was also significant but weaker, although it might be underestimated since inhibition of protein biosynthesis by chloramphenicol could possibly indirectly reduce metabolic activity. Thus, the observed phenomenon may be different from previously characterized ATP-dependent fluidization of the bacterial cytoplasm that enables mobility of large multiprotein complexes but apparently does not affect free GFP (*Montero Llopis et al.,*

2012; Parry et al., 2014), as also observed for sfGFP and other constructs in our experiments. The interplay between these energy-, metabolism-, and growth-dependent effects on diffusional properties of bacterial cytoplasm remains to be investigated.

Importantly, we observed that these perturbations to the cytoplasmic protein mobility, including cell growth and changes to the macromolecular crowding and temperature, have proportional effects on differently sized proteins. These results suggest that—within the tested size range—protein diffusion in E. coli cytoplasm remains Brownian under all tested conditions, including growing cells, and effects of these perturbations on protein mobility can be simply accounted for by changes in the cytoplasmic viscosity. We hypothesize that such proportional changes in diffusion of differently sized proteins might be important to maintain balanced rates of diffusion-limited cellular processes under various environmental conditions.

## Materials and methods

### Bacterial strains, plasmids, and media

All experiments were performed in the E. coli strain W3110 (Serra et al., 2013). Genes of interest were amplified by PCR using Q5 polymerase (New England Biosciences) and cloned in frame with sfGFP using Gibson assembly (Gibson et al., 2009) into pTrc99A vector (Amann et al., 1988), under control of the trc promoter inducible by isopropyl ß-D-1-thiogalactopyranoside (IPTG). All primers used in this study are listed in Appendix 1—table 1. In all cases sfGFP was fused at the C-terminus of the protein of interest with a GGGGS linker. The stability of the fusion constructs was verified by gel electrophoresis and immunoblotting using an anti-GFP primary antibody (JL-8 monoclonal, Takara). All plasmids used in this study are listed in Appendix 1—table 2. Point mutations were introduced by site-directed mutagenesis (New England Biosciences). The ΔclpA strain was generated by transferring the kanamycin resistance cassette from the corresponding mutant in the Keio collection (Baba et al., 2006) by P1 transduction. The cassette was further removed by FLP recombinase carried on the temperature-sensitive plasmid pCP20 (Cherepanov and Wackernagel, 1995).

E. coli cultures were grown in M9 minimal medium (48 mM $Na_2HPO_4$, 22 mM $KH_2PO_4$, 8.4 mM NaCl, 18.6 mM $NH_4Cl$, 2 mM $MgSO_4$, and 0.1 mM $CaCl_2$) supplemented with 0.2% casamino acids, 20 mM glucose, and 100 µg/ml ampicillin for selection. The overnight cultures were diluted to $OD_{600}$=0.035 and grown for 3.5 hr at 37°C and 200 rpm shaking. Cultures were treated for additional 45 min, under the same temperature and shaking conditions, with 100 µg/ml cephalexin and with 0–15 µM IPTG (Table 1), to induce expression of the fluorescent protein constructs. Where indicated, cultures were further incubated with 200 µg/ml rifampicin, DMSO as a mock treatment, 200 µg/ml chloramphenicol or 2 mM DNP for 1 hr or with 1 µg/ml A22 for 4 hr under the same temperature and shaking conditions.

### Growth curves

Measurements of bacterial growth were performed using 96-well plates (Cellstar transparent flat-bottom, Greiner). Overnight cultures were inoculated at an initial $OD_{600}$ of 0.01 in the same medium as used for growth in other experiments. Each well contained 150 µl of culture and the plate was covered with the plastic cover provided by the producer and further sealed with parafilm that prevents evaporation but allows air exchange. Plates were incubated at 37°C with continuous shaking, alternating between 150 s orbital and 150 s linear, in a Tecan Infinite 200 PRO plate reader.

### FCS data acquisition

Cells were harvested by centrifugation at 7000×g for 3 min and washed three times in tethering buffer (10 mM $K_2HPO_4$, 10 mM $KH_2PO_4$, 1 µM methionine, 10 mM sodium lactate, buffered with NaOH to pH 7). When indicated, 1 ml of chloramphenicol-treated cells were stained for 15 min with 300 nM SYTOX Orange Nucleic Acid Stain (Invitrogen). The excess of SYTOX Orange was washed in tethering buffer before proceeding with FCS experiments. 2.5 µl of bacterial cells were then spread on a small 1% agarose pad prepared in tethering buffer salts (10 mM $K_2HPO_4$, 10 mM $KH_2PO_4$ buffered with NaOH to pH 7), unless differently stated. Imaging was performed on Ibidi two-well µ-Slides (#1.5H, 170±5 µm). After the 45 min treatment with cephalexin, length of most bacterial cells was in a range of 4–8 µm.

FCS measurements were performed on an LSM 880 confocal laser scanning microscope (Carl Zeiss Microscopy) using a C-Apochromat 40×/1.2 water immersion objective selected for FCS. sfGFP was excited with a 488 nm Argon laser (25 mW) and fluorescence emission was collected from 490 to 580 nm. SYTOX Orange was excited with a 543 nm laser and fluorescence emission was collected from 553 to 615 nm. In order to avoid partial spectral overlap between the emission spectra of sfGFP and SYTOX Orange, fluorescence emission of sfGFP in the co-staining experiments was collected from 490 to 535 nm. Each sample was equilibrated for at least 20 min at 25°C (or 35°C when specified), on the stage of the microscope and measurements were taken at the same temperature. FCS measurements were acquired within 60 min from the sample preparation. The pinhole was aligned on a daily basis, by maximizing the fluorescence intensity count rate of an Alexa488 (Invitrogen) solution (35 nM) in phosphate-buffered saline (PBS; 137 mM NaCl, 2.7 mM KCl, 8 mM $Na_2HPO_4$, 1.8 mM $KH_2PO_4$, and pH 7.4). Unless differently stated, all measurements were performed with a pinhole size correspondent to 1 Airy unit, to ensure the optimal gathering of fluorescence signal. The coverslip collar adjustment ring of the water immersion objective was also adjusted daily, maximizing the fluorescence intensity signal and the brightness of Alexa 488. The laser power was adjusted in order to obtain molecular brightness (i.e., photon counts per second per molecule, cpsm) of 10 kcpsm for Alexa 488, using the ZEN software (Carl Zeiss Microscopy). The brightness of Alexa 488 was used as a daily reference to ensure constant laser power and adjusting it using the software-provided laser power percentage whenever necessary (range over the entire set of measurements was 0.11–0.18%). Before each measurement session, we acquired three sequential FCS measurements of Alexa488 in PBS, to verify the reproducibility of the confocal volume shape and size. The ratio between axial and lateral beam waist $S = \frac{z_0}{\omega_0}$ = 8.0±0.2 (Avg.±SEM) was obtained from a Brownian fit of the Alexa 488 autocorrelation curves using the ZEN software. For the lateral beam waist, we obtained $\omega_0$=0.186±0.001 μm (Avg.±SEM), calculated from the diffusion time $\tau_D$ = 20.9±0.11 μs (Avg.±SEM) obtained from the Brownian fit, being

$$D = \frac{\omega_0^2}{4\tau_D} \tag{1}$$

and being $D_{Alexa488}$=414 μm$^2$/s at 25°C (***Petrov et al., 2006***).

For the FCS measurements in vivo, the laser was positioned at the center of the short length axis and typically 0.8–1 μm from one of the cell poles along the long axis. For each cell, six sequential fluorescence intensity acquisitions of 20 s each were performed on the same spot (***Figure 1—figure supplement 3***). The laser power used for measurements in vivo was fixed to a value about seven times lower than for Alexa488 in PBS, in order to reduce photobleaching. Confocal images of the selected cell were routinely acquired before and after the FCS measurement to verify focal (*z*) and positioning (*xy*) stability (see Appendix 2 for additional information on the FCS measurements).

## FCS data analysis

Due to the small size of bacterial cells, fluorescence intensity traces are affected by photobleaching (Appendix 2). The effect of photobleaching on autocorrelation curves was corrected by detrending the long-time fluorescence decrease of each of the six fluorescence intensity traces using an ImageJ plugin (Jay Unruh, https://research.stowers.org/imagejplugins/index.html, Stowers Institute for Medical Research, USA). The plugin calculates the ACF from each fluorescence intensity trace, correcting it for the photobleaching effect by approximating the decreasing fluorescence intensity trend with a multi-segment line (the number of segments was fixed to 2). We obtained almost identical ACFs correcting for photobleaching effects by local averaging (***Appendix 2—figure 3***) using the FCS-dedicated software package Fluctuation Analyzer (***Wachsmuth et al., 2015***). In both cases, ACFs were calculated starting at 2 μs, since at times shorter than 2 μs, ACFs can be significantly affected by the GaAsp photomultipliers afterpulsing.

For each FCS measurement, we fitted all the six ACFs, calculated using the multi-segment detrending method, with a three-dimensional anomalous diffusion model that includes one diffusive component and one blinking component due to the protonation-deprotonation of the chromophore of sfGFP, according to the ***Equation (2)***:

$$G\left(\tau\right) = G_\infty + \frac{1}{N}\left(\frac{1 - F_P + F_P e^{-\frac{\tau}{\tau_P}}}{1 - F_P}\right)\frac{1}{1 + \left(\frac{\tau}{\tau_D}\right)^\alpha \sqrt{1 + \frac{1}{S^2}\left(\frac{\tau}{\tau_D}\right)^\alpha}} \tag{2}$$

where $N$ is the average number of particles in the confocal volume, $F_P$ is the fraction of particles in the non-fluorescent state, $\tau_P$ is the protonation-deprotonation lifetime at pH 7.5, $S = \frac{z_0}{\omega_0}$, the aspect ratio of the confocal volume with $z_0$ and $\omega_0$ being the axial and lateral beam waists, $\tau_D$ is the diffusion time in the confocal volume, $\alpha$ is the anomalous diffusion exponent, and $G_\infty$ is the offset of the ACF. The protonation-deprotonation lifetime ($\tau_P$) for sfGFP was fixed to 25 µs according to FCS measurements for sfGFP in PBS at pH 7.5 (**Cotlet et al., 2006**). The aspect ratio of the confocal volume was fixed to $S=8$ in the fittings to be consistent with the experimental calibration (see above). All other parameters were left free. For each FCS measurement, we calculated the average diffusion time $\tau_D$ and the average anomalous diffusion exponent $\alpha$ based on the autocorrelation curves of the six sequential fluorescence intensity traces. Importantly, no significant trend in $\tau_D$ or $\alpha$ was apparent when comparing the six sequential ACFs acquired for a given bacterial cell (**Appendix 2—figure 4**). Fitting to the anomalous diffusion model was performed using the Levenberg-Marquardt algorithm in the FCS analysis-dedicated software QuickFit 3.0 developed by Jan Wolfgang Krieger and Jörg Langowski (Deutsches Krebsforschungszentrum, Heidelberg, https://github.com/jkriege2/QuickFit3; **Krieger, 2018**). Identical results were obtained when fitting the data with OriginPro.

Alternatively, the ACFs were fitted by the OU model (Appendix 3) according to **Equation (3)**:

$$G\left(\tau\right) = G_\infty + \frac{1}{N}\left(\frac{1 - F_P + F_P e^{-\frac{\tau}{\tau_P}}}{1 - F_P}\right)\left(1 + 2\frac{\sigma^2}{\omega_0^2}\frac{1 - e^{-\frac{1}{2}\frac{\omega_0^2}{\sigma^2}\frac{\tau}{\tau_D}}}{1 + \frac{1}{8}\frac{\omega_0^2}{\sigma^2}}\right)^{-\frac{1}{2}}\left(1 + \frac{\tau}{\tau_D}\right)^{-\frac{1}{2}}\left(1 + 2\frac{\sigma^2}{S^2\omega_0^2}\frac{1 - e^{-\frac{1}{2}\frac{\omega_0^2}{\sigma^2}\frac{\tau}{\tau_D}}}{1 + \frac{1}{8}\frac{S^2\omega_0^2}{\sigma^2}}\right)^{-\frac{1}{2}} \tag{3}$$

where $S$ and $\tau_P$ were fixed to the same values mentioned for **Equation 2**, $\omega_0$ was fixed to 0.19 and $\sigma$ was fixed to $d/2=0.42$ µm, being $d$ the typical diameter of an *E. coli* cell (see OU model validation paragraph). Fitting to the OU model was performed with OriginPro.

## FRAP data acquisition and analysis

Cells for FRAP experiments were grown and prepared for imaging following the same protocol as for the FCS measurements. Due to the higher sensitivity of FCS at low fluorophore concentrations, several fusion constructs required higher induction by IPTG (**Table 1**) to obtain fluorescence intensity suitable for FRAP. The same LSM 880 confocal microscope, including objective and light path was used for FRAP as for the FCS measurements. The bacterial cell was imaged at 40×40 pixels with 30× zoom (pixel size 0.177 µm) with a pixel dwell time of 3.15 µs. First, 15 pre-bleaching frames were acquired at 2% laser power, subsequently the photobleaching was performed on 3×3 pixels area on one cell pole with 100% laser power for a total of 48 ms and 584 post-bleaching frames were acquired to monitor the fluorescence recovery. We observed that the mobile fraction for all constructs was >0.9. FRAP measurements were analyzed using simFRAP (**Blumenthal et al., 2015**), an ImageJ plugin based on a simulation approach implemented in a fast algorithm, which bypasses the need of using analytical models to interpolate the data. The simFRAP algorithm simulates two-dimensional random walks in each pixel, using the first image acquired after bleaching to define initial and boundary conditions, and it resolves numerically the diffusion equation by iterative simulation. The frame time and pixel size were fixed respectively to 0.018 s and 0.177 µm, and the target cell and the bleached region were defined as ImageJ ROIs (regions of interest). Of note, we used the target cell itself as a reference to compensate for the gradual bleaching during the measurement, as done previously (**Kumar et al., 2010**). This enabled us to achieve the highest possible temporal resolution, by reducing the acquisition area to a single *E. coli* cell. The FRAP derived diffusion coefficient $D_{FRAP}$ was directly obtained as output of the plugin.

## Cellular density measurements

Cell cultures were grown following the same protocol as for the FCS and FRAP measurements. Cultures were harvested at 4000×g for 5 min, and the pellet was resuspended in motility buffer (MB) (10 mM KPO$_4$, 0.1 mM EDTA, 67 mM NaCl, and 0.01% Tween 80). Tween 80 is a surfactant that

prevents cell-surface adhesion (*Nielsen et al., 2016*; *Schwarz-Linek et al., 2016*). Bacterial suspension was adjusted to a high cell density ($OD_{600}$=15) by subsequent centrifugation (4000×$g$, 5 min) and resuspension in a medium containing 20% iodixanol to match the density of MB with that of *E. coli* cell (1.11 g/ml) (*Martínez-Salas et al., 1981*). Each sample was then loaded in the chamber of a previously fabricated poly-di-methylsiloxane (PDMS) microfluidic device. The chamber consists of an inlet connected to an outlet by a straight channel of 50 µm height, 1 mm width, and 1 cm length. The channel was then sealed with grease to prevent fluid flows. After letting the mixtures reach the steady state in the microfluidic device for 40 min, cell sedimentation was visualized by acquiring *z*-stack images of the whole microfluidic channel using the same microscopy setup as for the FCS and FRAP measurements (1px = 0.2 µm in *x* and *y*, 1px = 1 µm in *z*; field of view = 303.64×303.64×70 µm³, 0.35 µs/px exposure). The number of cells in each $Z$ plane was quantified by the connected components labeling algorithm for ImageJ (*Legland et al., 2016*). Each experiment was conducted in three technical replicates. Because the height and the tilt of the microfluidic channels slightly varies from sample to sample, the $Z$ position was binned and the mean of the cell fraction over the bins was calculated.

The vertical density profiles were fitted to the theoretical expectation for diffusing particles in a buoyant fluid, $n\left(z\right) = n_o\exp\left(-\frac{z}{z_o}\right)$, in the range $z = \left[0.25,\ 0.8\right] \times 50$ µm to avoid effects of sample boundaries. The estimated values of the decay lengths $Z_0$ are plotted in *Figure 2—figure supplement 3H*. The fitted decay length is expected to obey $\frac{1}{z_o} = \frac{\rho V g}{k_B T}$, with $\rho$ the difference in density between the cells and the suspending fluid, $V$ the average volume of the cells, $g = 9.81$ m²/s the acceleration of gravity and $k_B T = 4.11$ pN·nm the thermal energy at 25°C. To compute the buoyancy-corrected cell density $\rho = \frac{k_B T}{V g\ z_o}$, the cell volume was estimated assuming the cells are cylinders closed by hemispherical caps, $V = \pi d^3/6 + \left(L - d\right)\pi d^2/4$. For all conditions, the cell diameter $d$ was evaluated on confocal images taken prior to FCS measurement (see *Figure 1—figure supplement 7*, *Figure 2C*, and *Figure 5—figure supplement 1*), and so was the length of cephalexin treated cells ($L$=5.5±0.1 (SEM) µm), cephalexin+A22 treated cells ($L$=5.8±0.1 (SEM) µm), and untreated cells ($L$=2.8±0.2 µm). Cell length for 100 mM NaCl, DMSO, rifampicin, and chloramphenicol treated cells was kept equal to the one of untreated cells, because cephalexin was not used during culture growth for sedimentation assay for these conditions. The estimated cell volumes are plotted in *Figure 2—figure supplement 3I*.

## Brownian dynamics simulations

We performed Brownian dynamics simulations of uncorrelated point particles under confinement. The $N$=50 fluorescent particles performed a random walk with steps taken from a Gaussian distribution of width $\sqrt{2D\Delta t}$, with $D$ the free diffusion coefficient and $\Delta t = 10^{-6}$ $s$ the simulation step. Confinement was imposed by redrawing the random steps that moved out of the confinement volume. Imposing elastic reflections on the walls yielded identical results. Subdiffusive behavior was simulated under reflexive boundary conditions as fractional Brownian motion $r\left(t + \Delta t\right) = r\left(t\right) + \eta\left(t\right)$, where the fractional Gaussian noise $\eta\left(t\right)$ is time correlated, $\left\langle \eta_i\left(t + t_0\right)\eta\left(t_0\right)\right\rangle = \frac{\Gamma_\alpha}{2}\left(\left|t + \Delta t\right|^\alpha + \left|t - \Delta t\right|^\alpha - 2\left|t\right|^\alpha\right)$ for $i$=$x$, $y$, and $z$, leading to the subdiffusive behavior $\left\langle \Delta r^2\left(t\right)\right\rangle = 3\Gamma_\alpha t^\alpha$ in the unconfined case. The correlated noise was produced from uncorrelated Gaussian distributed noise following the Davies and Harte method (*Davies and Harte, 1987*).

The confinement volume was assumed to be a cylinder of diameter $d$ and length ($L$-$d$) closed at both ends by hemispheric caps of diameter $d$, idealizing the shape of *E. coli*. The cell length was fixed to $L$=5 µm. The diameter was varied in the range $d$=[0.7, 1] µm. The collected fluorescence intensity was computed at each time step assuming a Gaussian intensity profile of the laser beam, $I\left(t\right) = \sum_{i=1}^N I_G\left(r_i\left(t\right) - r_0\right)$ with $r_i\left(t\right)$ the position of particle $i$, $r_0$ the center of the confocal volume and $I_G\left(r = \left(x, y, z\right)\right) = \exp\left(-2\left(\frac{x^2 + y^2}{\omega_0^2} + \frac{z^2}{z_0^2}\right)\right)$, with $\omega_0 = 200$ nm and $z_0 = 800$ nm the lateral and axial widths of the confocal volume. The normalized intensity autocorrelation $C\left(dt\right) = \frac{\left\langle I\left(t+dt\right)I\left(t\right)\right\rangle}{\left\langle I\left(t+dt\right)\right\rangle\left\langle I\left(t\right)\right\rangle} - 1$ is computed for logarithmically spaced lag times $dt$, to reflect experimental practices. The center of the confocal volume was chosen in the center of the cell along the *y* and *z* axes and 1 µm away from the edge of the cell along the longitudinal *x* axis of the cell, similarly to experimental conditions. The intensity ACF was finally multiplied by an exponential decay, $\left(1 + 0.1 * \exp-\frac{dt}{\tau_H}\right)/1.1$ with $\tau_H = 25$ µs, to mimic the blinking component due to the protonation-deprotonation process of sfGFP, before fitting with the different models of diffusion. The code used for this simulation is available in GitHub

(https://github.com/croelmiyn/Simulation_FCS_in_Bacteria, copy archived at swh:1:rev:47762b8b-24102b65441a4e2a04ba416a5108b7f0; *Colin, 2022*) and via DOI: 10.5281/zenodo.5940484.

### Validation of fitting by the OU model

We first estimated the relation between the width $\sigma$ of the potential well and the diameter $d$ of the bacteria by fitting the ACF of the Brownian simulations with the OU model, fixing all parameters except $\sigma$ to their ansatz values. The best fit was obtained for $\sigma \simeq d/2$ over the whole range of tested parameters. To mimic the fit procedure of experimental data and evaluate the accuracy of the diffusion coefficient estimation by the OU model (*Figure 2—figure supplement 3*), we then fixed $\sigma = d/2$, and $\omega_0$ and $z_0$ to their ansatz values, since they are measured independently in experiments, whereas the diffusion coefficient, number of particles $N$ in the confocal volume, fraction of triplet excitation and background noise were taken as free parameters.

## Acknowledgements

The authors thank Lotte Søgaard-Andersen, Martin Thanbichler, Knut Drescher, and Andreas Diepold for providing bacterial genomic DNA. The authors thank Silvia Espada Burriel for the assistance with the cellular density measurements and data analysis. This work was supported by the Max Planck Society.

## Additional information

### Funding

| Funder | Grant reference number | Author |
| --- | --- | --- |
| Max-Planck-Gesellschaft | | Nicola Bellotto<br>Jaime Agudo-Canalejo<br>Remy Colin<br>Ramin Golestanian<br>Gabriele Malengo<br>Victor Sourjik |

The funders had no role in study design, data collection and interpretation, or the decision to submit the work for publication.

### Author contributions

Nicola Bellotto, Conceptualization, Data curation, Formal analysis, Investigation, Visualization, Methodology, Writing – original draft, Writing – review and editing; Jaime Agudo-Canalejo, Conceptualization, Resources, Formal analysis, Investigation, Methodology, Writing – original draft, Writing – review and editing; Remy Colin, Conceptualization, Resources, Formal analysis, Investigation, Visualization, Methodology, Writing – original draft, Writing – review and editing; Ramin Golestanian, Conceptualization, Resources, Supervision, Investigation, Methodology, Writing – original draft, Writing – review and editing; Gabriele Malengo, Conceptualization, Resources, Data curation, Formal analysis, Validation, Investigation, Visualization, Methodology, Writing – original draft, Writing – review and editing; Victor Sourjik, Conceptualization, Supervision, Funding acquisition, Writing – original draft, Project administration, Writing – review and editing

### Author ORCIDs

Nicola Bellotto (ID) http://orcid.org/0000-0002-7701-9186
Jaime Agudo-Canalejo (ID) http://orcid.org/0000-0001-9677-6054
Remy Colin (ID) http://orcid.org/0000-0001-9051-8003
Ramin Golestanian (ID) http://orcid.org/0000-0002-3149-4002
Gabriele Malengo (ID) http://orcid.org/0000-0003-3522-8788
Victor Sourjik (ID) http://orcid.org/0000-0003-1053-9192

### Decision letter and Author response

Decision letter https://doi.org/10.7554/eLife.82654.sa1
Author response https://doi.org/10.7554/eLife.82654.sa2

## Additional files

### Supplementary files
• MDAR checklist

### Data availability
All data generated or analysed during this study are included in the manuscript and supporting files. Source data files have been provided for all figures.

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

# Appendix 1

## Lists of primers and plasmids used in this study

**Appendix 1—table 1.** List of primers used in this study.

| Primer name | Sense | Nucleotide sequence | Description |
|---|---|---|---|
| NBp1 | RW | ACCCATGGCACACTCCTTCACTAG | Amplify pTrc99A |
| NBp2 | RW | CTTGGACATGCTACCTCCGCCCCCTTAGTACAACGGTGACGCCGG | Amplify *ubiC* gene of *E. coli* MG1655 and fuse it to linker-*sfgfp* |
| NBp3 | FW | GGGGGCGGAGGTAGCATGTCCAAGGGTGAAGAGCTATTTAC | Amplify pTrc99A |
| NBp4 | FW | GTACTAGTGAAGGAGTGTGCCATGGGTATGTCACACCCCGCGTTAAC | Amplify *ubiC* gene of *E. coli* MG1655 and fuse it to *trc* promoter |
| NBp5 | FW | TTGACAATTAATCATCCGGCTCG | Sequence pTrc99A |
| NBp7 | RW | CTTGGACATGCTACCTCCGCCCCCGTACAACGGTGACGCCGG | Amplify *ubiC* gene from K12 and fuse it to linker-sfGFP |
| NBp8 | FW | GTACTAGTGAAGGAGTGTGCCATGGGTATGCGTATCATTCTGCTTGGCG | Amplify *adk* gene of *E. coli* MG1655 and fuse it to *trc* promoter |
| NBp9 | RW | CTTGGACATGCTACCTCCGCCCCCGCCGAGGATTTTTTCCAGATCAG | Amplify *adk* gene of *E. coli* MG1655 and fuse it to linker-sfGFP |
| NBp10 | FW | GTACTAGTGAAGGAGTGTGCCATGGGTATGTCGCAGAATAATCCGTT | Amplify *mmuM* gene of *E. coli* MG1655 and fuse it to *trc* promoter |
| NBp11 | RW | CTTGGACATGCTACCTCCGCCCCCGCTTCGCGCTTTTAACG | Amplify *mmuM* gene of *E. coli* MG1655 and fuse it to linker-sfGFP |
| NBp12 | FW | GTACTAGTGAAGGAGTGTGCCATGGGTATGGAAAACGCTAAAATGAACTCG | Amplify *dsdA* gene of *E. coli* MG1655 and fuse it to *trc* promoter |
| NBp13 | RW | CTTGGACATGCTACCTCCGCCCCCACGGCCTTTTGCCAGATATTG | Amplify *dsdA* gene of *E. coli* MG1655 and fuse it to linker-sfGFP |
| NBp14 | FW | GTACTAGTGAAGGAGTGTGCCATGGGTATGTCTGTACAGCAAATCGACTGGG | Amplify *hemN* gene from K12 genome and fuse it to *trc* promoter |
| NBp15 | RW | CTTGGACATGCTACCTCCGCCCCCAATCACCCGAGAGAACTGCTGC | Amplify *hemN* gene of *E. coli* MG1655 and fuse it to linker-sfGFP |
| NBp16 | FW | GTACTAGTGAAGGAGTGTGCCATGGGTATGAGTCAAACCATAACCCAGAG | Amplify *glcB* gene of *E. coli* MG1655 and fuse it to *trc* promoter |
| NBp17 | RW | CTTGGACATGCTACCTCCGCCCCCATGACTTTCTTTTTCGCGTAAAC | Amplify *glcB* gene of *E. coli* MG1655 and fuse it to linker-sfGFP |
| NBp18 | RW | GATTTAATCTGTATCAGG | Sequence pTrc99A |
| NBp19 | FW | GTACTAGTGAAGGAGTGTGCCATGGGTATGACAGTGGCGTATATTGC | Amplify *folK* gene of *E. coli* MG1655 and fuse it to *trc* promoter |
| NBp20 | RW | CTTGGACATGCTACCTCCGCCCCCCCCATTTGTTTAATTTGTCAA | Amplify *folK* gene of *E. coli* MG1655 and fuse it to linker-sfGFP |
| NBp21 | FW | GTACTAGTGAAGGAGTGTGCCATGGGTATGGCTATCTCAATCAAGACCCC | Amplify *map* gene of *E. coli* MG1655 and fuse it to *trc* promoter |
| NBp22 | RW | CTTGGACATGCTACCTCCGCCCCCTTCGTCGTGCGAGATTATCG | Amplify *map* gene of *E. coli* MG1655 and fuse it to linker-sfGFP |
| NBp23 | FW | GTACTAGTGAAGGAGTGTGCCATGGGTATGAAACTCTACAATCTGAAAG | Amplify *thrC* gene of *E. coli* MG1655 and fuse it to *trc* promoter |
| NBp24 | RW | CTTGGACATGCTACCTCCGCCCCCCTGATGATTCATCATCAATTTAC | Amplify *thrC* gene of *E. coli* MG1655 and fuse it to linker-sfGFP |
| NBp25 | FW | GTACTAGTGAAGGAGTGTGCCATGGGTATGTCAGCTCAAATCAACAACATCCG | Amplify *prpD* gene of *E. coli* MG1655 and fuse it to *trc* promoter |
| NBp26 | RW | CTTGGACATGCTACCTCCGCCCCCAATGACGTACAGGTCGAGATACTC | Amplify *prpD* gene of *E. coli* MG1655 and fuse it to linker-sfGFP |
| NBp27 | FW | GTACTAGTGAAGGAGTGTGCCATGGGTATGTTAAATGCATGGCACCTGC | Amplify *malZ* gene of *E. coli* MG1655 and fuse it to *trc* promoter |

*Appendix 1—table 1 Continued on next page*

*Appendix 1—table 1 Continued*

| Primer name | Sense | Nucleotide sequence | Description |
|---|---|---|---|
| NBp28 | RW | CTTGGACATGCTACCTCCGCCCCCGTTCATCCATACCGTAGCCGAAATG | Amplify *malZ* gene of *E. coli* MG1655 and fuse it to linker-sfGFP |
| NBp29 | FW | GTACTAGTGAAGGAGTGTGCCATGGGTATGTCTGAACCGCAACGTCTG | Amplify *thrP* gene of *E. coli* MG1655 and fuse it to *trc* promoter |
| NBp30 | RW | CTTGGACATGCTACCTCCGCCCCCTTGCGTTAGCGCCCAGC | Amplify *thrP* gene of *E. coli* MG1655 and fuse it to linker-sfGFP |
| NBp31 | FW | GTACTAGTGAAGGAGTGTGCCATGGGTATGTCTGTAATTAAGATGACCGATC | Amplify *pgk* gene of *E. coli* MG1655 and fuse it to *trc* promoter |
| NBp32 | RW | CTTGGACATGCTACCTCCGCCCCCCTTCTTAGCGCGCTCTTCG | Amplify *pgk* gene of *E. coli* MG1655 and fuse it to linker-sfGFP |
| NBp33 | FW | GTACTAGTGAAGGAGTGTGCCATGGGTATGAGGTATATAGTTGCCTTAACGG | Amplify *coaE* gene of *E. coli* MG1655 and fuse it to *trc* promoter |
| NBp35 | FW | GTACTAGTGAAGGAGTGTGCCATGGGTATGACGGCAATTGCCCC | Amplify *cmk* gene of *E. coli* MG1655 and fuse it to *trc* promoter |
| NBp37 | FW | GTACTAGTGAAGGAGTGTGCCATGGGTATGGATACGTCACTGGCTGAG | Amplify *entC* gene of *E. coli* MG1655 and fuse it to *trc* promoter |
| NBp39 | FW | GTACTAGTGAAGGAGTGTGCCATGGGTATGATTAGCGTAACCCTTAGCC | Amplify *murF* gene of *E. coli* MG1655 and fuse it to *trc* promoter |
| NBp41 | FW | GTACTAGTGAAGGAGTGTGCCATGGGTATGAAAATTACCGTATTGGGATGCG | Amplify *panE* gene of *E. coli* MG1655 and fuse it to *trc* promoter |
| NBp53 | FW | TCCAAGGGTGAAGAGCTATTTACTGGG | Deletion of ATG from *sfgfp* in *dsdA-sfgfp, ubiC-sfgfp, thrC-sfgfp, malZ-sfgfp* * |
| NBp54 | RW | GCTACCTCCGCCCCCACG | Deletion of ATG from *sfgfp* in *dsdA-sfgfp* * |
| NBp55 | FW | TCCAAGGGTGAAGAGCTATTTACTGGGGTTG | Deletion of ATG from *sfgfp* in *adk-sfgfp* * |
| NBp56 | RW | GCTACCTCCGCCCCCGCC | Deletion of ATG from *sfgfp* in *adk-sfgfp* * |
| NBp57 | FW | TCCAAGGGTGAAGAGCTATTTACTGGGG | Deletion of ATG from *sfgfp* in *mmuM-sfgfp* and *folK-sfgfp* * |
| NBp58 | RW | GCTACCTCCGCCCCCGCT | Deletion of ATG from *sfgfp* in *mmuM-sfgfp* * |
| NBp59 | RW | GCTACCTCCGCCCCCGTA | Deletion of ATG from *sfgfp* in *ubiC-sfgfp* * |
| NBp60 | FW | TCCAAGGGTGAAGAGCTATTTACTGG | Deletion of ATG from *sfgfp* in *glcB-sfgfp* * |
| NBp61 | RW | GCTACCTCCGCCCCCATG | Deletion of ATG from *sfgfp* in *glcB-sfgfp* * |
| NBp62 | FW | TCCAAGGGTGAAGAGCTATTTACTG | Deletion of ATG from *sfgfp* in *hemN-sfgfp, map-sfgfp, prpD-sfgfp* * |
| NBp63 | RW | GCTACCTCCGCCCCCAAT | Deletion of ATG from *sfgfp* in *hemN-sfgfp* and *prpD-sfgfp* * |
| NBp64 | RW | GCTACCTCCGCCCCCTTC | Deletion of ATG from *sfgfp* in *map-sfgfp* * |
| NBp65 | RW | GCTACCTCCGCCCCCCTG | Deletion of ATG from *sfgfp* in *thrC-sfgfp* * |
| NBp66 | RW | GCTACCTCCGCCCCCCCA | Deletion of ATG from *sfgfp* in *folK–sfgfp* * |
| NBp67 | RW | GCTACCTCCGCCCCCGTT | Deletion of ATG from *sfgfp* in *malZ-sfgfp* * |
| NBp68 | FW | GGGGGCGGAGGTAGCTCCAAGGGTGAAGAGCTATTTACTG | Amplification of backbone flexible linker-*sfgfp* without ATG |
| NBp81 | RW | GCTCTTCACCCTTGGAGCTACCTCCGCCCCCTGCGAGAGCCAATTTCTGG | Amplify *cmk* gene of *E. coli* MG1655 and fuse it to linker-*sfgfp* |

*Appendix 1—table 1 Continued*

| Primer name | Sense | Nucleotide sequence | Description |
|---|---|---|---|
| NBp82 | RW | GCTCTTCACCCTTGGAGCTACCTCCGCCCCCCGGTTTTTCCTGTGAGACAAAC | Amplify *coaE* gene of *E. coli* MG1655 and fuse it to linker-*sfgfp* |
| NBp83 | RW | GCTCTTCACCCTTGGAGCTACCTCCGCCCCCATGCAATCCAAAAACGTTCAACAT | Amplify *entC* gene of *E. coli* MG1655 and fuse it to linker -*sfgfp* deleted STOP |
| NBp84 | RW | GCTCTTCACCCTTGGAGCTACCTCCGCCCCCACATGTCCCATTCTCCTGTAAAG | Amplify *murF* gene of *E. coli* MG1655 and fuse it to linker-*sfgfp* |
| NBp85 | RW | GCTCTTCACCCTTGGAGCTACCTCCGCCCCCTTGCGTTAGCGCCCAGC | Amplify *thrP* gene of *E. coli* MG1655 and fuse it to linker-*sfgfp* |
| NBp86 | RW | GCTCTTCACCCTTGGAGCTACCTCCGCCCCCCCAGGGGCGAGGCAAAC | Amplify *panE* gene of *E. coli* MG1655 and fuse it to linker-*sfgfp* |
| NBp87 | RW | GCTCTTCACCCTTGGAGCTACCTCCGCCCCCCTTCTTAGCGCGCTCTTCG | Amplify *pgk* gene gene of *E. coli* MG1655 and fuse it to linker-*sfgfp* |
| NBp88 | FW | GTACTAGTGAAGGAGTGTGCCATGGGTATGGGTTTGTTCGATAAACTG | Amplify *crr* gene of *E. coli* MG1655 and fuse it to *trc* promoter |
| NBp89 | RW | TCACCCTTGGAGCTACCTCCGCCCCCCTTCTTGATGCGGATAACC | Amplify *crr* gene of *E. coli* MG1655 and fuse it to linker-*sfgfp* |
| NBp90 | FW | GTACTAGTGAAGGAGTGTGCCATGGGTATGCAAGAGCAATACCGCC | Amplify *leuS* gene of *E. coli* MG1655 and fuse it to *trc* promoter |
| NBp91 | RW | TTCACCCTTGGAGCTACCTCCGCCCCCGCCAACGACCAGATTGAGG | Amplify *leuS* gene of *E. coli* MG1655 and fuse it to linker-*sfgfp* |
| NBp92 | FW | GTACTAGTGAAGGAGTGTGCCATGGGTATGGCACTGCCAATTCTGTTAG | Amplify *rihA* gene of *E. coli* MG1655 and fuse it to *trc* promoter |
| NBp93 | RW | TTCACCCTTGGAGCTACCTCCGCCCCCAGCGTAAAATTTCAGACGATCAG | Amplify *rihA* gene of *E. coli* MG1655 and fuse it to linker-*sfgfp* |
| NBp94 | FW | GTACTAGTGAAGGAGTGTGCCATGGGTATGACCATTAAAAATGTAATTTGCGATATCG | Amplify *nagA* gene of *E. coli* MG1655 and fuse it to *trc* promoter |
| NBp95 | RW | TTCACCCTTGGAGCTACCTCCGCCCCCGATAACGTCGATTTCAGCGACTG | Amplify *nagA* gene of *E. coli* MG1655 and fuse it to linker-*sfgfp* |
| NBp96 | FW | GTACTAGTGAAGGAGTGTGCCATGGGTATGGGTAAAACGAACGACTG | Amplify *clpS* gene of *E. coli* MG1655 and fuse it to *trc* promoter |
| NBp97 | RW | TTCACCCTTGGAGCTACCTCCGCCCCCGGCTTTTTCTAGCGTACACAG | Amplify *clpS* gene of *E. coli* MG1655 and fuse it to linker-*sfgfp* |
| NBp98 | FW | GTACTAGTGAAGGAGTGTGCCATGGGTATGGAATCCCTGACGTTACAACC | Amplify *aroA* gene of *E. coli* MG1655 and fuse it to *trc* promoter |
| NBp99 | RW | TTCACCCTTGGAGCTACCTCCGCCCCCGGCTGCCTGGCTAATCCG | Amplify *aroA* gene of *E. coli* MG1655 and fuse it to linker-*sfgfp* |
| NBp100 | FW | GTACTAGTGAAGGAGTGTGCCATGGGTATGAGTTTTGTGGTCATTATTCCCG | Amplify *kdsB* gene of *E. coli* MG1655 and fuse it to *trc* promoter |
| NBp101 | RW | TTCACCCTTGGAGCTACCTCCGCCCCCGCGCATTTCAGCGCGAAC | Amplify *kdsB* gene of *E. coli* MG1655 and fuse it to linker-*sfgfp* |
| NBp102 | FW | GTACTAGTGAAGGAGTGTGCCATGGGTATGAAATACGATCTCATCATTATTGGCAG | Amplify *solA* gene of *E. coli* MG1655 and fuse it to *trc* promoter |
| NBp103 | RW | TTCACCCTTGGAGCTACCTCCGCCCCCTTGGAAGCGGGAAAGCCTG | Amplify *solA* gene of *E. coli* MG1655 and fuse it to linker-*sfgfp* |
| NBp107 | RW | CACCCTTGGAGCTACCTCCGCCCCCCCCATTTGTTTAATTTGTCAAATGCTC | Amplify *folK* gene of *E. coli* MG1655 and fuse it to linker-*sfgfp* |
| NBp122 | FW | ACTAGTGAAGGAGTGTGCCATGGGTGTGAGCAGCAAAGTGGAACAAC | Amplify *metH* gene of *E. coli* MG1655 and insert it into pTrc99A fused to *sfgfp* |
| NBp123 | RW | CACCCTTGGAGCTACCTCCGCCCCCGTCCGCGTCATACCCCAGATTC | |
| NBp124 | FW | ACTAGTGAAGGAGTGTGCCATGGGTATGTCGTCAACCCTACGAG | Amplify *acnA* gene of *E. coli* MG1655 and insert into pTrc99A fused to *sfgfp* |
| NBp125 | RW | CACCCTTGGAGCTACCTCCGCCCCCCTTCAACATATTACGAATGACATAATGC | |

*Appendix 1—table 1 Continued on next page*

Appendix 1—table 1 Continued

| Primer name | Sense | Nucleotide sequence | Description |
|---|---|---|---|
| NBp126 | FW | ACTAGTGAAGGAGTGTGCCATGGGTATGACAATATTGAATCACACCCTC | Amplify *metE* gene of *E. coli* MG1655 and insert into pTrc99A fused to *sfgfp* |
| NBp127 | RW | CACCCTTGGAGCTACCTCCGCCCCCCCCCCGACGCAAGTTC | |
| NBp177 | FW | ACTAGTGAAGGAGTGTGCCATGGGTATGAGCAGAACGATTTTTTGTAC | Amplify *yggX* of *E. coli* MG1655 and insert into pTrc99A fused to *sfgfp* |
| NBp178 | RW | CACCCTTGGAGCTACCTCCGCCCCCTTTTTTATCTTCCGGCGTATAG | |
| NBp179 | FW | ACTAGTGAAGGAGTGTGCCATGGGTATGAATCTGATCCTGTTCGG | Amplify *adk* gene from *Caulobacter crescentus* and insert into pTrc99A fused to *sfgfp* |
| NBp180 | RW | CACCCTTGGAGCTACCTCCGCCCCCTCCTGCAGCGACG | |
| NBp181 | FW | CAGACCATGTACTAGTGAAGGAGTGTGCCATGGGTATGACCTTCCGCACCCTC | Amplify *pgk* gene from *Caulobacter crescentus* and insert into pTrc99A fused to *sfgfp* |
| NBp182 | RW | CACCCTTGGAGCTACCTCCGCCCCCGGATTCGAGCGCCGC | |
| NBp183 | FW | ACTAGTGAAGGAGTGTGCCATGGGTATGGCCGTCTGTGGACAGC | Amplify *acnA* gene from *Caulobacter crescentus* and insert into pTrc99A fused to *sfgfp* |
| NBp184 | RW | CACCCTTGGAGCTACCTCCGCCCCCGTCGGCCTTGGCCAGG | |
| NBp185 | FW | ACTAGTGAAGGAGTGTGCCATGGGTATGAACTTAGTCTTAATGGGG | Amplify *adk* from *Bacillus subtilis* and insert into pTrc99A fused to *sfgfp* |
| NBp186 | RW | CACCCTTGGAGCTACCTCCGCCCCCTTTTTTTAATCCTCCAAGAAGATCC | |
| NBp187 | FW | ACTAGTGAAGGAGTGTGCCATGGGTATGAATAAAAAAACTCTCAAAGACATCG | Amplify *pgk* from *Bacillus subtilis* and insert into pTrc99A fused to *sfgfp* |
| NBp188 | RW | CACCCTTGGAGCTACCTCCGCCCCCTTTATCGTTCAGTGCAGCTAC | |
| NBp189 | FW | ACTAGTGAAGGAGTGTGCCATGGGTATGGCAAACGAGCAAAAAAC | Amplify *acnA* from *Bacillus subtilis* and insert into pTrc99A fused to *sfgfp* |
| NBp190 | RW | CACCCTTGGAGCTACCTCCGCCCCCGGACTGCTTCATTTTTTCACG | |
| NBp191 | FW | ACTAGTGAAGGAGTGTGCCATGGGTATGAACCTGATCCTGTTGGGG | Amplify *adk* from *Myxococcus xanthus* and insert into pTrc99A fused to *sfgfp* |
| NBp192 | RW | CACCCTTGGAGCTACCTCCGCCCCCGGCCTTGCCCGCAG | |
| NBp193 | FW | ACTAGTGAAGGAGTGTGCCATGGGTATGATCCGTTACATCGATGATCTGC | Amplify *pgk* from *Myxococcus xanthus* and insert into pTrc99A fused to *sfgfp* |
| NBp194 | RW | CACCCTTGGAGCTACCTCCGCCCCCCCGCGTCTCCAGCG | |
| NBp195 | FW | ACTAGTGAAGGAGTGTGCCATGGGTATGACCGACAGTTTCGGC | Amplify *acnA* from *Myxococcus xanthus* and insert into pTrc99A fused to *sfgfp* |
| NBp196 | RW | CACCCTTGGAGCTACCTCCGCCCCCGCCCTTGGCCAGTTG | |
| NBp197 | FW | ACTAGTGAAGGAGTGTGCCATGGGTATGCGCATCATTCTTCTCGG | Amplify *adk* from *Vibrio cholerae* and insert into pTrc99A fused to *sfgfp* |
| NBp198 | RW | CACCCTTGGAGCTACCTCCGCCCCCAGCCAACGCTTTAGCAATGTC | |
| NBp199 | FW | ACTAGTGAAGGAGTGTGCCATGGGTATGTCTGTAATCAAGATGATTGACCTGG | Amplify *pgk* from *Vibrio cholerae* and insert into pTrc99A fused to *sfgfp* |
| NBp200 | RW | CACCCTTGGAGCTACCTCCGCCCCCGCTTTAGCGCGTGCTTC | |
| NBp201 | FW | ACTAGTGAAGGAGTGTGCCATGGGTATGAACAGTCTGTATCGTAAAGC | Amplify *acnA* from *Vibrio cholerae* and insert into pTrc99A fused to *sfgfp* |
| NBp202 | RW | CACCCTTGGAGCTACCTCCGCCCCCCTGCGCCAAAAAGTCTTG | |
| NBp216 | FW | ACTAGTGAAGGAGTGTGCCATGGGTATGCGTATCATTCTGCTGG | amplify *adk* from *Yersinia enterocolitica* and insert into pTrc99A fused to *sfgfp* |
| NBp217 | RW | CACCCTTGGAGCTACCTCCGCCCCCACCGAGAATAGTCGCCAG | |
| NBp218 | FW | CTAGTGAAGGAGTGTGCCATGGGTATGTCTGTAATTAAGATGACCGATCTGG | Amplify *pgk* from *Yersinia enterocolitica* and insert into pTrc99A fused to *sfgfp* |
| NBp219 | RW | CACCCTTGGAGCTACCTCCGCCCCCCTGCTTAGCGCGCTCTTC | |
| NBp220 | FW | ACTAGTGAAGGAGTGTGCCATGGGTATGTCGTTGGATTTGCGGAAAAC | Amplify *acnA* from *Yersinia enterocolitica* and insert into pTrc99A fused to *sfgfp* |
| NBp221 | RW | CACCCTTGGAGCTACCTCCGCCCCCCAACATTTTGCGGATCACATAATGC | |
| NBp227 | FW | GATGGCTGGACGGTAGAAACCGAAGATCGCAGCTTGTCTGCAC | Site-directed mutagenesis of Lys to Glu in *map-sfgfp* |
| NBp228 | RW | TTCCATGGTGCGGATCTCTTTTTCACCCGCGTTGACCATTGG | |
| NBp229 | FW | GATGGCTGGACGGTAGCAACCGCAGATCGCAGCTTGTCTGCAC | Site-directed mutagenesis of Lys to Ala in *map-sfgfp* |
| NBp230 | RW | TGCCATGGTGCGGATCTCTTTTGCACCCGCGTTGACCATTGG | |
| NBp231 | FW | ACTAGTGAAGGAGTGTGCCATGGGTATGGGTAAAATAATTGGTATCG | Amplify *dnaK* from *E. coli* MG1655 and insert into pTrc99A fused to *sfgfp* |
| NBp232 | RW | CACCCTTGGAGCTACCTCCGCCCCCTTTTTTGTCTTTGACTTCTTC | |
| NBp234 | FW | CCAGTCTGCGTTTACCATCCATG | Site-directed mutagenesis of V436F in *dnaK-sfgfp* |
| NBp235 | RW | TTGTCTTCAGCGGTAGAG | |

Appendix 1—table 1 Continued on next page

*Appendix 1—table 1 Continued*

| Primer name | Sense | Nucleotide sequence | Description |
|---|---|---|---|
| NBp240 | FW | TTTTCTTATGATGTAGAACGTGCA ACGCAATTGATGCTCGCTGTTGCGTACCAGGGGAAGGCCATT | |
| NBp241 | RW | GAATTTTTGTAACACGTCAATAA CAAACTCCATCGGAGTGTACGCCGCATTGACTAATATCACTTTATACATAGATGGC | Site-directed mutagenesis of D35A, D36A, H66A in *clpS-sfgfp* |
| Eri121 | FW | CAGTCATAGCCG AATAGCCT | |
| Eri122 | RW | CGGTGCCCTGAA TGAACTGC | Checking insertion of KanR cassette |

*Indicated constructs were erroneously generated omitting deletion of ATG start codon of *sfgfp* gene and thus corrected with site-directed mutagenesis.

**Appendix 1—table 2.** List of plasmids generated for this study.

| Plasmid | Relevant genotype | Reference or source |
|---|---|---|
| pTrc99A | Amp$^r$; expression vector; pBR ori; *trc* promoter, IPTG inducible | *Amann et al., 1988* |
| pCP20 | Amp$^r$, Cam$^r$; *flp* | *Cherepanov and Wackernagel, 1995* |
| pNB1 | Amp$^r$; sfGFP in pTrc99A | This work |
| pNB3 | Amp$^r$; Adk-sfGFP in pTrc99A | This work |
| pNB4 | Amp$^r$; CoaE-sfGFP in pTrc99A | This work |
| pNB5 | Amp$^r$; Cmk-sfGFP in pTrc99A | This work |
| pNB6 | Amp$^r$; Pgk-sfGFP in pTrc99A | This work |
| pNB7 | Amp$^r$; MmuM-sfGFP in pTrc99A | This work |
| pNB8 | Amp$^r$; PrpD-sfGFP in pTrc99A | This work |
| pNB9 | Amp$^r$; DsdA-sfGFP in pTrc99A | This work |
| pNB11 | Amp$^r$; GlcB-sfGFP in pTrc99A | This work |
| pNB13 | Amp$^r$; HemN-sfGFP in pTrc99A | This work |
| pNB14 | Amp$^r$; Map$^{WT}$-sfGFP in pTrc99A | This work |
| pNB15 | Amp$^r$; ThrC-sfGFP in pTrc99A | This work |
| pNB16 | Amp$^r$; MalZ-sfGFP in pTrc99A | This work |
| pNB17 | Amp$^r$; EntC-sfGFP in pTrc99A | This work |
| pNB18 | Amp$^r$; ThpR-sfGFP in pTrc99A | This work |
| pNB19 | Amp$^r$; AroA-sfGFP in pTrc99A | This work |
| pNB20 | Amp$^r$; ClpS$^{WT}$-sfGFP in pTrc99A | This work |
| pNB21 | Amp$^r$; Crr-sfGFP in pTrc99A | This work |
| pNB22 | Amp$^r$; KdsB-sfGFP in pTrc99A | This work |
| pNB23 | Amp$^r$; LeuS-sfGFP in pTrc99A | This work |
| pNB24 | Amp$^r$; MurF-sfGFP in pTrc99A | This work |
| pNB25 | Amp$^r$; NagD-sfGFP in pTrc99A | This work |
| pNB26 | Amp$^r$; RihA-sfGFP in pTrc99A | This work |
| pNB27 | Amp$^r$; SolA-sfGFP in pTrc99A | This work |
| pNB28 | Amp$^r$; UbiC-sfGFP in pTrc99A | This work |
| pNB29 | Amp$^r$; PanE-sfGFP in pTrc99A | This work |
| pNB30 | Amp$^r$; FolK-sfGFP in pTrc99A | This work |
| pNB39 | Amp$^r$; AcnA-sfGFP in pTrc99A | This work |

*Appendix 1—table 2 Continued on next page*

*Appendix 1—table 2 Continued*

| Plasmid | Relevant genotype | Reference or source |
|---|---|---|
| pNB40 | Amp$^r$; MetE-sfGFP in pTrc99A | This work |
| pNB42 | Amp$^r$; MetH-sfGFP in pTrc99A | This work |
| pNB44 | Amp$^r$; YggX-sfGFP in pTrc99A | This work |
| pNB45 | Amp$^r$; Adk$^{C.c.}$-sfGFP in pTrc99A | This work |
| pNB46 | Amp$^r$; Adk$^{V.c.}$-sfGFP in pTrc99A | This work |
| pNB47 | Amp$^r$; Adk$^{M.x.}$-sfGFP in pTrc99A | This work |
| pNB48 | Amp$^r$; AcnA$^{M.x.}$-sfGFP in pTrc99A | This work |
| pNB49 | Amp$^r$; AcnA$^{V.c.}$-sfGFP in pTrc99A | This work |
| pNB51 | Amp$^r$; Adk$^{B.s.}$-sfGFP in pTrc99A | This work |
| pNB52 | Amp$^r$; AcnA$^{B.s.}$-sfGFP in pTrc99A | This work |
| pNB54 | Amp$^r$; Adk$^{Y.e.}$-sfGFP in pTrc99A | This work |
| pNB56 | Amp$^r$; Pgk$^{C.c.}$-sfGFP in pTrc99A | This work |
| pNB58 | Amp$^r$; Pgk$^{V.c.}$-sfGFP in pTrc99A | This work |
| pNB59 | Amp$^r$; Pgk$^{M.x.}$sfGFP in pTrc99A | This work |
| pNB60 | Amp$^r$; AcnA$^{Y.e.}$-sfGFP in pTrc99A | This work |
| pNB61 | Amp$^r$; DnaK$^{WT}$-sfGFP in pTrc99A | This work |
| pNB62 | Amp$^r$; Map$^{K211E\_K218E\_K224E\_K226E}$-sfGFP in pTrc99A | This work |
| pNB63 | Amp$^r$; Map$^{K211A\_K218A\_K224A\_K226A}$-sfGFP in pTrc99A | This work |
| pNB64 | Amp$^r$; DnaK$^{V436F}$-sfGFP in pTrc99A | This work |
| pNB66 | Amp$^r$; ClpS$^{D35A\_D36A\_H66A}$-sfGFP in pTrc99A | This work |

## Appendix 2

### Notes on the acquisition and analysis protocols for FCS measurements in bacterial cells

Due to the limited size of bacterial cells, FCS measurements require precise positioning of the confocal volume in the bacterial cytoplasm and minimization of the photobleaching-induced effects. In order to ensure that, before fitting an ACF, we verified the stability of the lateral (*xy*) positioning of the observation volume by visually analyzing for lateral drifts in confocal images acquired immediately before and after the FCS acquisition. This was done by annotating the *xy* position in the pre-aquisition image and verifying that the positioning did not change in the post-aquisition image after 120 s. Measurements showing *xy* drift were excluded from the analysis (*Appendix 2—figure 1*). Furthermore, the focal stability of the sample was increased by thermal equilibration on the microscope stage before measurements.

Long-term photobleaching due to the progressive decrease of the total number of fluorescent proteins during FCS experiments (*Appendix 2—figure 2*) is unavoidable due to the small volume of *E. coli* cells, and it requires correction to avoid artifacts. We observed that almost identical ACFs were obtained when correcting for the photobleaching using either multi-segment detrending (Jay Unruh, https://research.stowers.org/imagejplugins/index.html, Stowers Institute for Medical Research, USA) or a local averaging approach (*Wachsmuth et al., 2015*; *Appendix 2—figure 3*).

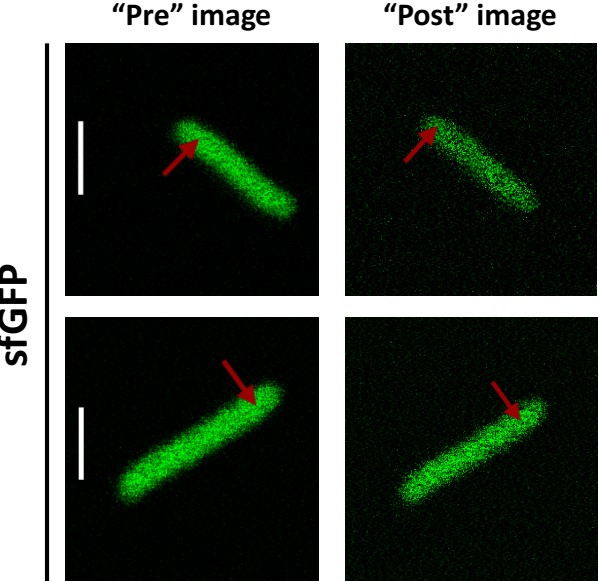

**Appendix 2—figure 1.** Typical examples of presence or absence of lateral focal drift during FCS measurements. Substantial lateral drift could be observed for <10% of experiments (upper images), whereas most measurement showed no perceptible lateral drift (lower images). FCS, fluorescence correlation spectroscopy. Scale bars are 2 µm.

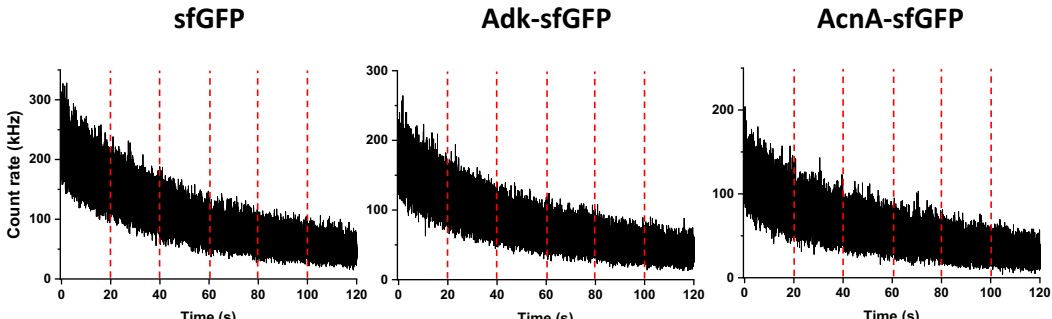

**Appendix 2—figure 2.** Typical traces of fluorescence intensity during FCS measurements. Examples of fluorescence intensity traces for indicated protein fusions. The vertical red dashed lines separate sequential fluorescence intensity acquisitions on the same cell. FCS, fluorescence correlation spectroscopy.

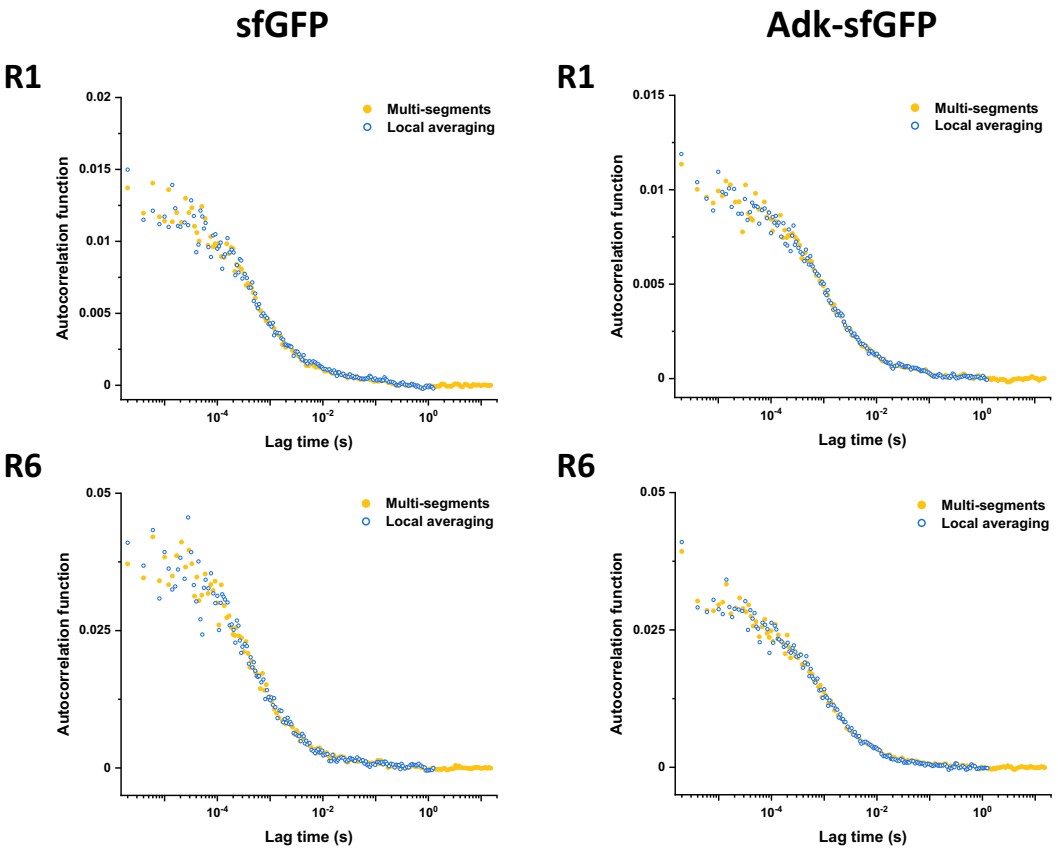

**Appendix 2—figure 3.** Results of detrending with multi-segments and local averaging approaches. Comparison of experimental ACFs corrected using either multi-segments or local averaging approaches (as indicated) for sfGFP and Adk-sfGFP and different data acquisition segments (R1 vs. R6). ACF, autocorrelation function.

We also confirmed that there was no systematic trend in the fitted values of $\tau_D$ and $\alpha$ with the time of the fluorescence trace acquisition (*Appendix 2—figure 4*). An additional process that could potentially affect ACFs is short-term photobleaching of the fluorophore in the confocal volume, also known as cryptic photobleaching, which can artificially accelerate the decrease of the ACF and lead to an underestimation of the protein residence time *Macháň et al., 2016*. This process is different from long-term photobleaching, which is caused by the continuous illumination in the entire illumination light cone. However, the effect of cryptic photobleaching was shown to be typically <5%, even for proteins that diffuse 10–100 times slower and have higher bleaching rates than our constructs (*Macháň et al., 2016*; *Wachsmuth et al., 2015*).

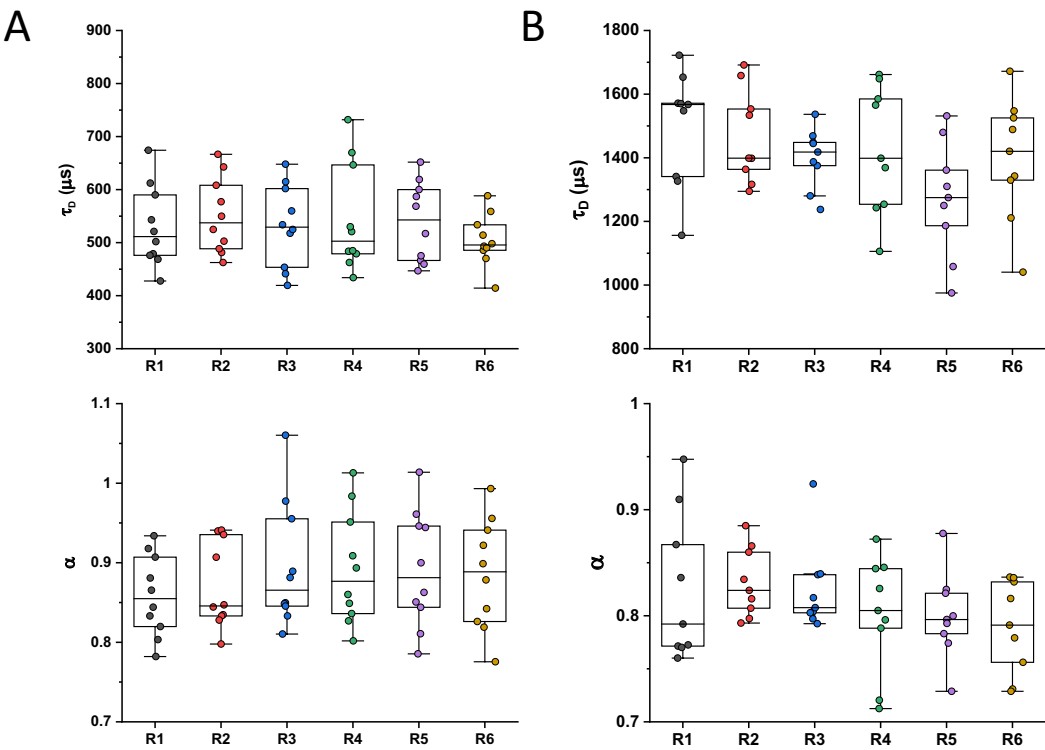

**Appendix 2—figure 4.** Values of $\tau_D$ or $\alpha$ for the six sequential ACFs. Values were determined by fitting the anomalous diffusion model to experimental ACFs for the six sequential time segments per individual cell expressing sfGFP (**A**) or MetH-sfGFP (**B**). ACF, autocorrelation function.

## Appendix 3

## OU model for confinement effect in FCS measurements

We aim to derive the ACF for an FCS experiment in which the fluorescent particles are confined by a (possibly anisotropic) harmonic potential centered at $(x, y, z) = (0, 0, 0)$, that is

$$V(x, y, z) = \frac{k_x x^2}{2} + \frac{k_y y^2}{2} + \frac{k_z z^2}{2}$$

where $k_i$ represents the stiffness of the potential in each dimension, and thus the extent $\sigma_i$ of confinement along that dimension given by $\sigma_i^2 \equiv k_B T / k_i$.

We can treat each dimension independently, using $x$ without loss of generality in what follows. Diffusion in a harmonic potential is described by the OU process. The corresponding Green's function $P(x, t|x_0)$, representing the probability of finding a particle at position $x$ at time $t$ given that it was at position $x_0$ at time $t = 0$, is

$$P(x, t|x_0) = \frac{1}{\sqrt{2\pi\sigma_x^2\left(1 - e^{-\frac{2Dt}{\sigma_x^2}}\right)}} \exp\left[-\frac{1}{2\sigma_x^2}\frac{\left(x - x_0 e^{-\frac{Dt}{\sigma_x^2}}\right)^2}{1 - e^{-\frac{2Dt}{\sigma_x^2}}}\right]$$

at long times, $t \to \infty$, we recover the stationary state given by the Boltzmann distribution corresponding to the harmonic trap

$$P_{st}(x) = \frac{1}{\sqrt{2\pi\sigma_x^2}}\exp\left[-\frac{x^2}{2\sigma_x^2}\right].$$

The ACF $G(t)$ of an FCS measurement is given as the multiple integral over the product of the probability to detect a photon from a molecule at some initial position $x_0$, the probability density that it diffuses from this position to a final position $x$ within time $t$ (given by Green's function), and the probability to detect a photon from a molecule at this final position (**Enderlein et al., 2005**; **Enderlein, 2012**). Note that the probability of detection of a molecule will necessarily be proportional to the intensity of the laser beam, which we can assume Gaussian and also centered at $(x, y, z) = (0, 0, 0)$, with the usual form

$$I(x, y, z) = I_0\exp\left(-\frac{2x^2}{\omega_0^2}\right)\exp\left(-\frac{2y^2}{\omega_0^2}\right)\exp\left(-\frac{2z^2}{S^2\omega_0^2}\right) \equiv I_0 I_x(x)\, I_y(y)\, I_z(z)$$

where $\omega_0$ is the width of the (circular) laser beam along the $x$ and $y$ directions, and $S$ is a dimensionless factor accounting for the anisotropy along the $z$ direction, that is, the axial direction of the beam.

Ignoring constant normalization factors and baselines, the time dependent part of the ACF is then given by $G(t) = G_x(t)\, G_y(t)\, G_z(t)$ with

$$G_x(t) = \int dx \int dx_0 I_x(x)\, P(x, t|x_0)\, I_x(x_0)\, P_{st}(x_0)$$

which can be directly integrated to give, after normalizing so that $G_x(t = 0) = 1$, the expression

$$G_x(t) = \left[1 + 2\frac{\sigma_x^2}{\omega_0^2}\frac{1 - e^{-\frac{2Dt}{\sigma_x^2}}}{1 + \frac{1}{8}\frac{\omega_0^2}{\sigma_x^2}}\right]^{-\frac{1}{2}}. \tag{A2-1}$$

Note that, in the limit of no confinement, $\sigma_x^2 \to \infty$, this equation reduces to the well-known ACF for unconfined diffusion

$$G_x\left(t\right) = \left[1 + \frac{4Dt}{\omega_0^2}\right]^{-\frac{1}{2}} = \left[1 + \frac{t}{\tau_D}\right]^{-\frac{1}{2}}$$

where we have defined the diffusion time $\tau_D \equiv \omega_0^2/\left(4D\right)$. With this definition, the ACF in **Equation A2-1** can be rewritten as

$$G_x\left(t\right) = \left[1 + 2\frac{\sigma_x^2}{\omega_0^2}\frac{1 - e^{-\frac{1}{2}\frac{\omega_0^2}{\sigma_x^2}\frac{t}{\tau_D}}}{1 + \frac{1}{8}\frac{\omega_0^2}{\sigma_x^2}}\right]^{-\frac{1}{2}}.$$

The full three-dimensional ACF is then, in general,

$$G\left(t\right) = \left[1 + 2\frac{\sigma_x^2}{\omega_0^2}\frac{1 - e^{-\frac{1}{2}\frac{\omega_0^2}{\sigma_x^2}\frac{t}{\tau_D}}}{1 + \frac{1}{8}\frac{\omega_0^2}{\sigma_x^2}}\right]^{-\frac{1}{2}} \left[1 + 2\frac{\sigma_y^2}{\omega_0^2}\frac{1 - e^{-\frac{1}{2}\frac{\omega_0^2}{\sigma_y^2}\frac{t}{\tau_D}}}{1 + \frac{1}{8}\frac{\omega_0^2}{\sigma_y^2}}\right]^{-\frac{1}{2}} \left[1 + 2\frac{\sigma_z^2}{S^2\omega_0^2}\frac{1 - e^{-\frac{1}{2}\frac{\omega_0^2}{\sigma_z^2}\frac{t}{\tau_D}}}{1 + \frac{1}{8}\frac{S^2\omega_0^2}{\sigma_z^2}}\right]^{-\frac{1}{2}}.$$

The cylindrical geometry of a bacterium can be approximated by an infinite cylinder along the $y$ direction, so that $\sigma_x = \sigma_z = \sigma$ and $\sigma_y \to \infty$, resulting in the ACF

$$G\left(t\right) = \left[1 + 2\frac{\sigma^2}{\omega_0^2}\frac{1 - e^{-\frac{1}{2}\frac{\omega_0^2}{\sigma^2}\frac{t}{\tau_D}}}{1 + \frac{1}{8}\frac{\omega_0^2}{\sigma^2}}\right]^{-\frac{1}{2}} \left[1 + \frac{t}{\tau_D}\right]^{-\frac{1}{2}} \left[1 + 2\frac{\sigma^2}{S^2\omega_0^2}\frac{1 - e^{-\frac{1}{2}\frac{\omega_0^2}{\sigma^2}\frac{t}{\tau_D}}}{1 + \frac{1}{8}\frac{S^2\omega_0^2}{\sigma^2}}\right]^{-\frac{1}{2}}. \qquad \text{(A2-2)}$$

**Equation A2-2**, with the added baseline and multiplicative correction accounting for particles in the non-fluorescent state, corresponds to **Equation 3** in the main text.

## Appendix 4

### Effective diffusion coefficient of two linked proteins

In previous work, we studied the diffusion of two spherical objects with radii $a_1$ and $a_2$, joined together by a flexible linker (**Agudo-Canalejo and Golestanian, 2020**). In the limit of a rigid linker of length , the effective diffusion coefficient of the composite object goes as:

$$D \simeq \frac{k_B T}{6\pi\eta a_1} \frac{a_1}{(a_1 + a_2)} \left[ 1 + 2 \frac{a_1 a_2}{(a_1 + a_2)(a_1 + a_2 + l)} - \frac{9}{8} \frac{a_1 a_2 (a_1 - a_2)^2}{(a_1 + a_2)^2 (a_1 + a_2 + l)^2} \right] \qquad \text{(A3-1)}$$

plus higher order correction terms of order $O\left( \frac{a_i^3}{(a_1 + a_2 + l)^3} \right)$.

We can then consider what is the effective diffusion coefficient of two proteins that are linked to each other. For that, we first need to connect the molecular mass to the effective radius of the protein. If we identify subunit 1 with GFP, and subunit 2 with the protein attached to it, and we call $M_{GFP}$ the molecular mass of GFP and $M_{tot}$ the total molecular mass (sum of GFP and the protein), we expect relations of the form

$$a_1 = C M_{GFP}^{\beta}$$

$$a_2 = C \left( M_{tot} - M_{GFP} \right)^{\beta}$$

where $C$ is a proportionality constant assumed to be typical for all proteins (which is of order 1, with values reported in the literature of about 0.65 [**Smilgies and Folta-Stogniew, 2015**], when the mass is in kDa and the radius is in nm), and β is the scaling exponent introduced in the main text. For the linker which is made of six amino-acids, we may use the typical conversion factor 0.35 nm/amino-acid to estimate $l \approx 2$ nm.

Plugging these expressions for $a_1$ and $a_2$ into **Equation A3-1** above, one obtains an expression for the diffusion coefficient $D$ as a function of $M_{tot}$ that depends only on three parameters (since the molecular mass of GFP is known): (i) the diffusion coefficient of GFP $\frac{k_B T}{6\pi\eta a_1}$, (ii) the exponent $\beta$, and (iii) the rescaled linker length $l/C$.

# Appendix 5

## Exact *p*-values for all significance analysis

**Appendix 5—table 1.** *Figure 1C*.

| Testing pair | P-value |
|---|---|
| sfGFP vs Adk-sfGFP | 0.000000010 |
| Adk-sfGFP vs AcnA-sfGFP | 0.00000000044 |

**Appendix 5—table 2.** *Figure 1—figure supplement 10A*.

| Testing pair | P-value |
|---|---|
| ClpS$^{WT}$-sfGFP versus ClpS$^{D35A\_D36A\_H66A}$-sfGFP | 0.000094 |
| Map$^{WT}$-sfGFP versus Map$^{Lys \to Ala}$-sfGFP | 0.0000012 |
| Map$^{WT}$-sfGFP versus Map$^{Lys \to Glu}$-sfGFP | 0.000000016 |
| Map$^{Lys \to Ala}$-sfGFP versus Map$^{Lys \to Glu}$-sfGFP | 0.10 |
| DnaK$^{WT}$-sfGFP versus DnaK$^{V436F}$-sfGFP | 0.00023 |

**Appendix 5—table 3.** *Figure 1—figure supplement 10B*.

| Testing pair | P-value |
|---|---|
| ClpS$^{WT}$-sfGFP versus ClpS$^{D35A\_D36A\_H66A}$-sfGFP | 0.000014 |
| Map$^{WT}$-sfGFP versus Map$^{Lys \to Ala}$-sfGFP | 0.0092 |
| Map$^{WT}$-sfGFP versus Map$^{Lys \to Glu}$-sfGFP | 0.065 |
| Map$^{Lys \to Ala}$-sfGFP versus Map$^{Lys \to Glu}$-sfGFP | 0.37 |
| DnaK$^{WT}$-sfGFP versus DnaK$^{V436F}$-sfGFP | 0.00000019 |

**Appendix 5—table 4.** *Figure 2C*.

| Testing pair | P-value |
|---|---|
| Untreated versus A22 treatment | 0.0000000000007 |

**Appendix 5—table 5.** *Figure 2D*.

| Testing pair | P-value |
|---|---|
| Untreated versus A22 treatment | 0.000001 |

**Appendix 5—table 6.** *Figure 2—figure supplement 2*.

| Testing pair | P-value |
|---|---|
| Untreated versus A22 treatment | 0.002 |

**Appendix 5—table 7.** *Figure 2—figure supplement 4A*.

| Testing pair | P-value |
|---|---|
| sfGFP, 1 A.U. versus 0.66 A.U. | 0.00001 |
| DnaK-sfGFP 1 A.U. versus 0.66 A.U. | 0.60 |
| AcnA-sfGFP, 1 A.U. versus 0.66 A.U. | 0.000002 |

**Appendix 5—table 8.** *Figure 2—figure supplement 4B*.

| Testing pair | P-value |
|---|---|
| sfGFP, 1 A.U. versus 0.66 A.U. | 0.24 |

*Appendix 5—table 8 Continued on next page*

*Appendix 5—table 8 Continued*

| Testing pair | P-value |
| --- | --- |
| DnaK-sfGFP 1 A.U. versus 0.66 A.U. | 0.50 |
| AcnA-sfGFP, 1 A.U. versus 0.66 A.U. | 0.002 |

**Appendix 5—table 9.** *Figure 4—figure supplement 1A*.

| Testing pair | P-value |
| --- | --- |
| Adk[E.c.]-sfGFP versus Adk[Y.e.]-sfGFP | 0.17 |
| Adk[E.c.]-sfGFP versus Adk[V.c.]-sfGFP | 0.056 |
| Adk[E.c.]-sfGFP versus Adk[C.c.]-sfGFP | 0.93 |
| Adk[E.c.]-sfGFP versus Adk[M.x.]-sfGFP | 0.21 |
| Adk[E.c.]-sfGFP versus Adk[B.s.]-sfGFP | 0.23 |
| Pgk[E.c.]-sfGFP versus Pgk[V.c.]-sfGFP | 0.75 |
| Pgk[E.c.]-sfGFP versus Pgk[C.c.]-sfGFP | 0.26 |
| Pgk[E.c.]-sfGFP versus Pgk[M.x.]-sfGFP | 0.33 |
| AcnA[E.c.]-sfGFP versus AcnA[Y.e.]-sfGFP | 0.093 |
| AcnA[E.c.]-sfGFP versus AcnA[V.c.]-sfGFP | 0.084 |
| AcnA[E.c.]-sfGFP versus AcnA[M.x.]-sfGFP | 0.0000000023 |
| AcnA[E.c.]-sfGFP versus AcnA[B.s.]-sfGFP | 0.069 |

**Appendix 5—table 10.** *Figure 4—figure supplement 1B*.

| Testing pair | P-value |
| --- | --- |
| Adk[E.c.]-sfGFP versus Adk[Y.e.]-sfGFP | 0.000060 |
| Adk[E.c.]-sfGFP versus Adk[V.c.]-sfGFP | 0.18 |
| Adk[E.c.]-sfGFP versus Adk[C.c.]-sfGFP | 0.042 |
| Adk[E.c.]-sfGFP versus Adk[M.x.]-sfGFP | 0.070 |
| Adk[E.c.]-sfGFP versus Adk[B.s.]-sfGFP | 0.0029 |
| Pgk[E.c.]-sfGFP versus Pgk[V.c.]-sfGFP | 0.11 |
| Pgk[E.c.]-sfGFP versus Pgk[C.c.]-sfGFP | 0.0082 |
| Pgk[E.c.]-sfGFP versus Pgk[M.x.]-sfGFP | 0.0087 |
| AcnA[E.c.]-sfGFP versus AcnA[Y.e.]-sfGFP | 0.035 |
| AcnA[E.c.]-sfGFP versus AcnA[V.c.]-sfGFP | 0.16 |
| AcnA[E.c.]-sfGFP versus AcnA[M.x.]-sfGFP | 0.083 |
| AcnA[E.c.]-sfGFP versus AcnA[B.s.]-sfGFP | 0.00024 |

**Appendix 5—table 11.** *Figure 5—figure supplement 2A*.

| Testing pair | P-value |
| --- | --- |
| sfGFP ionic strength 105 mM versus 305 mM | 0.0000036 |
| Adk-sfGFP ionic strength 105 mM versus 305 mM | 0.000044 |
| AroA-sfGFP ionic strength 105 mM versus 305 mM | 0.035 |
| AcnA-sfGFP ionic strength 105 mM versus 305 mM | 0.0018 |

**Appendix 5—table 12.** *Figure 5—figure supplement 2B*.

| Testing pair | P-value |
|---|---|
| sfGFP 25°C versus 35°C | 0.00015 |
| Adk-sfGFP 25°C versus 35°C | 0.0000025 |
| AcnA-sfGFP 25°C versus 35°C | 0.00077 |

**Appendix 5—table 13.** *Figure 5—figure supplement 2C*.

| Testing pair | P-value |
|---|---|
| sfGFP Untreated versus Chloramphenicol | 0.40 |
| sfGFP DMSO versus Rifampicin | 0.012 |
| Adk-sfGFP Untreated versus Chloramphenicol | 0.12 |
| Adk-sfGFP DMSO versus Rifampicin | 0.00048 |
| Pgk-sfGFP Untreated versus Chloramphenicol | 0.17 |
| Pgk-sfGFP DMSO versus Rifampicin | 0.0000012 |
| AcnA-sfGFP Untreated versus Chloramphenicol | 0.000011 |
| AcnA-sfGFP DMSO versus Rifampicin | 0.0085 |

**Appendix 5—table 14.** *Figure 5—figure supplement 2D*.

| Testing pair | P-value |
|---|---|
| sfGFP M9 salts versus M9 salts, Glu+CA | 0.000023 |
| Adk-sfGFP M9 salts versus M9 salts, Glu+CA | 0.000070 |
| AroA-sfGFP M9 salts versus M9 salts, Glu+CA | 0.00000015 |
| AcnA-sfGFP M9 salts versus M9 salts, Glu+CA | 0.0022 |

**Appendix 5—table 15.** *Figure 5—figure supplement 3A*.

| Testing pair | P-value |
|---|---|
| sfGFP ionic strength 105 mM versus 305 mM | 0.097 |
| Adk-sfGFP ionic strength 105 mM versus 305 mM | 0.54 |
| AroA-sfGFP ionic strength 105 mM versus 305 mM | 0.077 |
| AcnA-sfGFP ionic strength 105 mM versus 305 mM | 0.31 |

**Appendix 5—table 16.** *Figure 5—figure supplement 3B*.

| Testing pair | P-value |
|---|---|
| sfGFP 25°C versus 35°C | 0.12 |
| Adk-sfGFP 25°C versus 35°C | 0.005 |
| AcnA-sfGFP 25°C versus 35°C | 0.26 |

**Appendix 5—table 17.** *Figure 5—figure supplement 3C*.

| Testing pair | P-value |
|---|---|
| sfGFP Untreated versus Chloramphenicol | 0.32 |
| sfGFP DMSO versus Rifampicin | 0.50 |
| Adk-sfGFP Untreated versus Chloramphenicol | 0.53 |
| Adk-sfGFP DMSO versus Rifampicin | 0.32 |

*Appendix 5—table 17 Continued on next page*

*Appendix 5—table 17 Continued*

| Testing pair | P-value |
|---|---|
| Pgk-sfGFP Untreated versus Chloramphenicol | 0.17 |
| Pgk-sfGFP DMSO versus Rifampicin | 0.59 |
| AcnA-sfGFP Untreated versus Chloramphenicol | 0.008 |
| AcnA-sfGFP DMSO versus Rifampicin | 0.42 |

**Appendix 5—table 18.** *Figure 5—figure supplement 3D*.

| Testing pair | P-value |
|---|---|
| sfGFP M9 salts versus M9 salts, Glu+CA | 0.44 |
| Adk-sfGFP M9 salts versus M9 salts, Glu+CA | 0.035 |
| AroA-sfGFP M9 salts versus M9 salts, Glu+CA | 0.69 |
| AcnA-sfGFP M9 salts versus M9 salts, Glu+CA | 0.31 |

**Appendix 5—table 19.** *Figure 5—figure supplement 4A*.

| Testing pair | P-value |
|---|---|
| sfGFP Grown 25°C, measured 25°C versus 35°C | 0.0020 |
| sfGFP Measured 25°C, grown 25°C versus 37°C | 0.060 |
| sfGFP Measured 35°C, grown 25°C versus 37°C | 0.98 |
| sfGFP Grown 37°C, measured 25°C versus 35°C | 0.000040 |

**Appendix 5—table 20.** *Figure 5—figure supplement 4B*.

| Testing pair | P-value |
|---|---|
| sfGFP Grown 25°C, measured 25°C versus 35°C | 0.26 |
| sfGFP Measured 25°C, grown 25°C versus 37°C | 0.45 |
| sfGFP Measured 35°C, grown 25°C versus 37°C | 0.44 |
| sfGFP Grown 37°C, measured 25°C versus 35°C | 0.12 |

**Appendix 5—table 21.** *Figure 5—figure supplement 5A*.

| Testing pair | P-value |
|---|---|
| sfGFP cytoplasm versus nucleoid | 0.20 |
| AcnA-sfGFP cytoplasm versus nucleoid | 0.062 |

**Appendix 5—table 22.** *Figure 5—figure supplement 5B*.

| Testing pair | P-value |
|---|---|
| sfGFP cytoplasm versus nucleoid | 0.09 |
| AcnA-sfGFP cytoplasm versus nucleoid | 0.09 |

**Appendix 5—table 23.** *Figure 5—figure supplement 6A*.

| Testing pair | P-value |
|---|---|
| sfGFP M9 salts versus M9 salts, Cam | 0.82 |
| sfGFP M9 salts versus M9 salts, Glu+CA | 0.000023 |
| sfGFP M9 salts, Cam vs M9 salts, Cam, Glu +CA | 0.019 |
| sfGFP M9 salts, Glu +CA versus M9 salts, Cam, Glu +CA | 0.019 |

*Appendix 5—table 23 Continued on next page*

*Appendix 5—table 23 Continued*

| Testing pair | P-value |
|---|---|
| AcnA-sfGFP M9 salts versus M9 salts, Cam | 0.37 |
| AcnA-sfGFP M9 salts versus M9 salts, Glu+CA | 0.0022 |
| AcnA-sfGFP M9 salts, Cam versus M9 salts, Cam, Glu +CA | 0.0035 |
| AcnA-sfGFP M9 salts, Glu +CA versus M9 salts, Cam, Glu +CA | 0.014 |

**Appendix 5—table 24.** *Figure 5—figure supplement 6B*.

| Testing pair | P-value |
|---|---|
| sfGFP M9 salts versus M9 salts, Cam | 0.33 |
| sfGFP M9 salts versus M9 salts, Glu+CA | 0.44 |
| sfGFP M9 salts, Cam versus M9 salts, Cam, Glu+CA | 0.91 |
| sfGFP M9 salts, Glu+CA versus M9 salts, Cam, Glu+CA | 0.80 |
| AcnA-sfGFP M9 salts versus M9 salts, Cam | 0.099 |
| AcnA-sfGFP M9 salts versus M9 salts, Glu+CA | 0.31 |
| AcnA-sfGFP M9 salts, Cam versus M9 salts, Cam, Glu+CA | 0.0085 |
| AcnA-sfGFP M9 salts, Glu+CA versus M9 salts, Cam, Glu+CA | 0.56 |

**Appendix 5—table 25.** *Figure 5—figure supplement 7A*.

| Testing pair | P-value |
|---|---|
| sfGFP untreated versus DNP treatment, 25°C | 0.52 |
| sfGFP untreated versus DNP treatment, 35°C | 0.66 |
| Adk-sfGFP untreated versus DNP treatment, 25°C | 0.46 |
| Adk-sfGFP untreated versus DNP treatment, 35°C | 0.03 |
| AcnA-sfGFP untreated versus DNP treatment, 25°C | 0.59 |
| AcnA-sfGFP untreated versus DNP treatment, 35°C | 0.98 |

**Appendix 5—table 26.** *Figure 5—figure supplement 7B*.

| Testing pair | P-value |
|---|---|
| sfGFP untreated versus DNP treatment, 25°C | 0.013 |
| sfGFP untreated versus DNP treatment, 35°C | 0.32 |
| Adk-sfGFP untreated versus DNP treatment, 25°C | 0.006 |
| Adk-sfGFP untreated versus DNP treatment, 35°C | 0.25 |
| AcnA-sfGFP untreated versus DNP treatment, 25°C | 0.94 |
| AcnA-sfGFP untreated versus DNP treatment, 35°C | 0.57 |

## Appendix 6

### Numerosity of the constructs and conditions for each experiment

Appendix 6—table 1. *Figure 1* and *Figure 1—figure supplement 5*.

| Construct | Numerosity (*n*) | Construct | Numerosity (*n*) |
|---|---|---|---|
| sfGFP | 52 | EntC-sfGFP | 15 |
| YggX-sfGFP | 8 | AroA-sfGFP | 9 |
| ClpS$^{WT}$-sfGFP | 11 | ThrC-sfGFP | 14 |
| FolK-sfGFP | 8 | MurF-sfGFP | 7 |
| Crr-sfGFP | 14 | DsdA-sfGFP | 14 |
| UbiC-sfGFP | 14 | HemN-sfGFP | 13 |
| CoaE-sfGFP | 11 | PrpD-sfGFP | 12 |
| Adk-sfGFP | 23 | DnaK$^{WT}$-sfGFP | 10 |
| Cmk-sfGFP | 16 | MalZ-sfGFP | 9 |
| KdsB-sfGFP | 22 | GlcB-sfGFP | 16 |
| Map$^{WT}$-sfGFP | 20 | MetE-sfGFP | 8 |
| MmuM-sfGFP | 14 | LeuS-sfGFP | 14 |
| PanE-sfGFP | 18 | AcnA-sfGFP | 19 |
| SolA-sfGFP | 7 | MetH-sfGFP | 9 |
| Pgk-sfGFP | 16 | | |

Appendix 6—table 2. *Figure 1—figure supplement 7*.

| Construct | Numerosity (*n*) |
|---|---|
| sfGFP | Same as *Appendix 6—table 1* |

Appendix 6—table 3. *Figure 1—figure supplement 8*.

| Condition | Numerosity (*n*) |
|---|---|
| Untreated | 5 |
| Cephalexin | 6 |

Appendix 6—table 4. *Figure 1—figure supplement 10*.

| Construct | Numerosity (*n*) |
|---|---|
| ClpS$^{WT}$-sfGFP | Same as *Appendix 6—table 1* |
| ClpS$^{D35A\_D36A\_H66A}$-sfGFP | 10 |
| Map$^{WT}$-sfGFP | Same as *Appendix 6—table 1* |
| Map$^{Lys\rightarrow Glu}$-sfGFP | 10 |
| Map$^{Lys\rightarrow Ala}$-sfGFP | 12 |
| DnaK$^{WT}$-sfGFP | Same as *Appendix 6—table 1* |
| DnaK$^{V436F}$-sfGFP | 10 |

Appendix 6—table 5. *Figure 2C and D* and *Figure 2—figure supplement 2*.

| Condition | Numerosity (*n*) |
|---|---|
| Untreated | Same as *Appendix 6—table 1* |
| A22-treated | 12 |

**Appendix 6—table 6.** *Figure 2—figure supplement 4*.

| Condition | Numerosity (*n*) |
|---|---|
| sfGFP 1 A.U. | Same as *Appendix 6—table 1* |
| sfGFP 0.66 A. U. | 12 |
| DnaK-sfGFP 1 A.U. | Same as *Appendix 6—table 1* |
| DnaK-sfGFP 0.66 A.U, | 10 |
| AcnA-sfGFP 1 A.U. | Same as *Appendix 6—table 1* |
| AcnA-sfGFP 0.66 A. U. | 10 |

**Appendix 6—table 7.** *Figure 2—figure supplement 5*.

| Construct | Numerosity (n) |
|---|---|
| sfGFP | Same as *Appendix 6—table 1* |
| Adk-sfGFP | Same as *Appendix 6—table 1* |
| DnaK-sfGFP | Same as *Appendix 6—table 1* |
| DnaKV436-sfGFP | Same as *Appendix 6—table 4* |
| AcnA-sfGFP | Same as *Appendix 6—table 1* |

**Appendix 6—table 8.** *Figure 3C*.

| Construct | Numerosity (*n*) | Construct | Numerosity (*n*) |
|---|---|---|---|
| sfGFP | 11 | DsdA-sfGFP | 10 |
| YggX-sfGFP | 10 | GlcB-sfGFP | 10 |
| Adk-sfGFP | 16 | AcnA-sfGFP | 10 |
| PanE-sfGFP | 11 | MetH-sfGFP | 15 |

**Appendix 6—table 9.** *Figure 4* and *Figure 4—figure supplement 1*.

| Construct | Numerosity (*n*) | Construct | Numerosity (*n*) |
|---|---|---|---|
| Adk$^{E.c.}$-sfGFP | Same as *Appendix 6—table 1* | Pgk$^{C.c.}$-sfGFP | 5 |
| Adk$^{Y.e.}$-sfGFP | 5 | Pgk$^{M.x.}$-sfGFP | 11 |
| Adk$^{V.c.}$-sfGFP | 10 | AcnA$^{E.c.}$-sfGFP | Same as *Appendix 6—table 1* |
| Adk$^{C.c.}$-sfGFP | 5 | AcnA$^{Y.e.}$-sfGFP | 5 |
| Adk$^{M.x.}$-sfGFP | 11 | AcnA$^{V.c.}$-sfGFP | 10 |
| Adk$^{B.s.}$-sfGFP | 10 | AcnA$^{M.x.}$-sfGFP | 10 |
| Pgk$^{E.c.}$-sfGFP | Same as *Appendix 6—table 1* | AcnA$^{B.s.}$-sfGFP | 10 |
| Pgk$^{V.c.}$-sfGFP | 10 | | |

**Appendix 6—table 10.** *Figure 5A*, *Figure 5—figure supplement 1A, B*, *Figure 5—figure supplement 2*, and *Figure 5—figure supplement 3A*.

| Construct and condition | Numerosity (*n*) | Construct and condition | Numerosity (*n*) |
|---|---|---|---|
| sfGFP, 105 mM | Same as *Appendix 6—table 1* | AroA-sfGFP, 105 mM | Same as *Appendix 6—table 1* |
| sfGFP, 305 mM | 11 | AroA-sfGFP, 305 mM | 12 |
| Adk-sfGFP, 105 mM | Same as *Appendix 6—table 1* | AcnA-sfGFP, 105 mM | Same as *Appendix 6—table 1* |
| Adk-sfGFP, 305 mM | 11 | AcnA-sfGFP, 305 mM | 6 |

**Appendix 6—table 11.** *Figure 5B*, *Figure 5—figure supplement 1C, D*, *Figure 5—figure supplement 2*, and *Figure 5—figure supplement 3B*.

| Construct and condition | Numerosity (*n*) | Construct and condition | Numerosity (*n*) |
|---|---|---|---|
| sfGFP, 25°C | Same as *Appendix 6—table 1* | AcnA-sfGFP, 25°C | Same as *Appendix 6—table 1* |
| sfGFP, 35°C | 14 | AcnA-sfGFP, 35°C | 18 |
| Adk-sfGFP, 25°C | Same as *Appendix 6—table 1* | | |
| Adk-sfGFP, 35°C | 21 | | |

**Appendix 6—table 12.** *Figure 5C*, *Figure 5—figure supplement 1E, F*, *Figure 5—figure supplement 2C*, and *Figure 5—figure supplement 3C*.

| Construct and condition | Numerosity (*n*) | Construct and condition | Numerosity (*n*) |
|---|---|---|---|
| sfGFP, untreated | Same as *Appendix 6—table 1* | Pgk-sfGFP, untreated | Same as *Appendix 6—table 1* |
| sfGFP, chloramphenicol | 10 | Pgk-sfGFP, chloramphenicol | 10 |
| sfGFP, DMSO | 15 | Pgk-sfGFP, DMSO | 10 |
| sfGFP, rifampicin | 15 | Pgk-sfGFP, rifampicin | 10 |
| Adk-sfGFP, untreated | Same as *Appendix 6—table 1* | AcnA-sfGFP, untreated | Same as *Appendix 6—table 1* |
| Adk-sfGFP, chloramphenicol | 10 | AcnA-sfGFP, chloramphenicol | 10 |
| Adk-sfGFP, DMSO | 10 | AcnA-sfGFP, DMSO | 10 |
| Adk-sfGFP, rifampicin | 10 | AcnA-sfGFP, rifampicin | 10 |

**Appendix 6—table 13.** *Figure 5D*, *Figure 5—figure supplement 1G, H*, *Figure 5—figure supplement 2D*, and *Figure 5—figure supplement 3D*.

| Construct and condition | Numerosity (*n*) | Construct and condition | Numerosity (*n*) |
|---|---|---|---|
| sfGFP, M9 salts | 10 | AroA-sfGFP, M9 salts | 11 |
| sfGFP, M9 salts, Glu+CA | 15 | AroA-sfGFP, M9 salts, Glu+CA | 11 |
| Adk-sfGFP, M9 salts | 11 | AcnA-sfGFP, M9 salts | 10 |
| Adk-sfGFP, M9 salts, Glu+CA | 12 | AcnA-sfGFP, M9 salts, Glu+CA | 10 |

**Appendix 6—table 14.** *Figure 5—figure supplement 4*.

| Construct and condition | Numerosity (*n*) |
|---|---|
| sfGFP, grown 25°C, measured 25°C | 10 |
| sfGFP, grown 25°C, measured 35°C | 10 |
| sfGFP, grown 37°C, measured 25°C | Same as *Appendix 6—table 1* |
| sfGFP, grown 37°C, measured 35°C | Same as *Appendix 6—table 11* |

**Appendix 6—table 15.** *Figure 5—figure supplement 5*.

| Construct and condition | Numerosity (*n*) |
|---|---|
| sfGFP, cytoplasm | 10 |
| sfGFP, nucleoid | 10 |

*Appendix 6—table 15 Continued on next page*

*Appendix 6—table 15 Continued*

| Construct and condition | Numerosity (*n*) |
|---|---|
| AcnA-sfGFP, cytoplasm | 10 |
| AcnA-sfGFP, nucleoid | 10 |

**Appendix 6—table 16.** *Figure 5—figure supplement 6*.

| Construct and condition | Numerosity (*n*) | Construct and condition | Numerosity (*n*) |
|---|---|---|---|
| sfGFP, M9 salts | Same as *Appendix 6—table 12* | AcnA-sfGFP, M9 salts | Same as *Appendix 6—table 12* |
| sfGFP, M9 salts, Cam | 13 | AcnA-sfGFP, M9 salts, Cam | 8 |
| sfGFP, M9 salts, Glu+CA | Same as *Appendix 6—table 12* | AcnA-sfGFP, M9 salts, Glu+CA | Same as *Appendix 6—table 12* |
| sfGFP, M9 salts, Cam, Glu+CA | 13 | AcnA-sfGFP, M9 salts, Cam, Glu+CA | 10 |

**Appendix 6—table 17.** *Figure 5—figure supplement 7*.

| Construct and condition | Numerosity (*n*) |
|---|---|
| sfGFP, untreated, 25°C | Same as *Appendix 6—table 1* |
| sfGFP, 2 mM DNP, 25°C | 5 |
| sfGFP, untreated, 35°C | Same as *Appendix 6—table 10* |
| sfGFP, 2 mM DNP, 35°C | 6 |
| Adk-sfGFP, untreated, 25°C | Same as *Appendix 6—table 1* |
| Adk-sfGFP, 2 mM DNP, 25°C | 5 |
| Adk-sfGFP, untreated, 35°C | Same as *Appendix 6—table 10* |
| Adk-sfGFP, 2 mM DNP, 35°C | 6 |
| AcnA-sfGFP, untreated, 25°C | Same as *Appendix 6—table 1* |
| AcnA-sfGFP, 2 mM DNP, 25°C | 5 |
| AcnA-sfGFP, untreated, 35°C | Same as *Appendix 6—table 10* |
| AcnA-sfGFP, 2 mM DNP, 35°C | 6 |

