## [Editor Report]

The work of Bellotto et al. provides a comprehensive and compelling study of the diffusion of proteins in the cytoplasm of the bacterium *Escherichia coli*, using multiple measurement methods, notably Fluorescence Correlation Spectroscopy. It is found that fast diffusing proteins roughly follow the Stokes-Einstein relation, while proteins that strongly interact with the cytoplasm manifest subdiffusion. This study will be a valuable resource for scientists seeking to understand the temporal dynamics of proteins within cells.

---

## [Decision Letter]

**Decision letter after peer review:**

[Editors’ note: the authors submitted for reconsideration following the decision after peer review. What follows is the decision letter after the first round of review.]

Thank you for submitting the paper "Dependence of diffusion in *Escherichia coli* cytoplasm on protein size, environmental conditions and cell growth" for consideration by *eLife*. Your article has been reviewed by 2 peer reviewers, and the evaluation has been overseen by a Reviewing Editor and a Senior Editor. The following individual involved in review of your submission has agreed to reveal their identity: Conrad W Mullineaux (Reviewer #1).

Comments to the Authors:

We are sorry to say that, after consultation with the reviewers, we have decided that this work will not be considered further for publication by *eLife*.

As you will see in the reviews, both reviewers have appreciated the technical rigor of your work and the characterization of diffusion within *E. coli* well beyond previous work and with superior methodology. However, both were concerned whether major conceptual advances were obtained based on these results. In light of this, we regretfully cannot accept the paper in its current form. If you believe that you can restructure the paper and interpret your data in a manner that will enable you to draw novel conclusions and obtain a fundamental advance, we will be glad to reconsider this decision.

*Reviewer #1 (Recommendations for the authors):*

A very thorough study of protein diffusion in the *E. coli* cytoplasm, looking at multiple proteins (including mutants in which specific interactions are disabled), multiple conditions and two different methods to measure diffusion. I don't think it leads to any major conceptual advances: basically the results confirm what was already inferred from the previous studies cited: smaller proteins roughly follow the Stokes-Einstein relation for the size dependence of the diffusion coefficient, larger proteins show subdiffusion, interactions with other cell components slow everything down, different measurement methods give comparable but not identical answers. There is merit in having such a comprehensive set of measurements all in one place: this will be a valuable reference point for anyone who wants to explore the physical nature of the cytoplasm further, or who wants to factor cytoplasmic protein diffusion into a model for some dynamic process in *E. coli*.

The paper is well-written and well-presented, and I cannot find any technical flaws with the parts that I am able to judge. The weakness is maybe a lack of novelty: this provides strong confirmation of things that were already inferred on the basis of less complete data, but I missed any major conceptual advances in the understanding of the dynamics of the cytoplasm. If there is anything I missed, please highlight it better!

*Reviewer #2 (Recommendations for the authors):*

Belotto and co-workers performed a systematic experimental study of the intracellular mobility of 28 (+3 discarded on the way) cytoplasmic proteins in *E. coli*, using fluorescence correlation spectroscopy (FCS), complemented by some computational/ modeling. This technique is underexplored in the body of work looking at intracellular diffusion in bacteria, and allows them to probe very short time scales compared to commonly used techniques such as single-particle tracking (SPT).

The manuscript is not really focused on a main finding, but a main finding may be that the data cannot falsify a Brownian diffusion model when confinement is accounted for (see below). Other interesting findings consider the temperature and growth-rate dependency, and the agreement of FCS with FRAP (recovery after photobleaching) data.

The work is well written and provides a useful and precise set of measurements to the community. Its main advantages are it being systematic (28 is a large set of proteins for this kind of study), the novelty of FCS in this context, and the use of physical models in support of the data. However, we have some major concerns about the main results and conclusions:

0) In the way it is written, the manuscript does not identify a central question, and the findings could be better connected to the current debate.

1) In our view the central question could become the fact that the authors argue that conventional diffusion seems to be supported (or at least cannot be ruled out) for these data, while previous (SPT) data have supported mild but clear subdiffusion for (larger) cytoplasmic particles, including protein complexes. However, we have some possibly important concerns about this analysis, and in any case we think it needs more experimental and data-analysis/modeling controls (see below).

2) The other main results (dependency on growth, temperature, etc.) are interesting, but most of these things have been quantified by SPT, and a more careful comparison appears to be needed. Additionally, these results also need controls on cell size and density/crowding levels (see below).

With these important revisions, we believe the work could make a nice addition to the current debate.

We try to detail our impressions in the comments below.

1) Our main concern is that the controls/support of the claim of conventional diffusion may be insufficient. If well supported, my impression is that this could become (either way) a central result of the study.

The authors clearly show using modeling that their data cannot falsify Brownian diffusion. However, it is not clear to us that they can falsify fBM or fLe subdiffusion. A22 treatment provides an interesting control but PMID: 34341116 (see Figure 3) clearly shows that this treatment affects dry-mass density (QPI measurements are actually a proxy of macromolecular density, hence crowding).

Previous studies have clearly supported the idea that density (crowding) affects cytoplasmic diffusion (see e.g. PMID: 33083729), hence it seems to us that we do not know whether the observed changes in FCS may come from the density changes rather than the confinement. For example the previously observed anomalous diffusion could be due to larger size of (non-interacting) proteins or protein complexes, or to the presence of the chromosome, or the current data would just not allow to falsify either scenario, etc.

Also, the lack of a clear Stokes-Einstein relation even using fairly complex models of diffusion makes us think of a possible complex underlying dynamics (due to disorder or viscoelasticity, or both) or more in general other possible (but possibly interesting) physical scenarios.

Below we try to propose some controls and analyses.

The authors do not mention that (larger) protein particles like GFP muNS were also be reported to be subdiffusive by SPT. For example in PMID: 30374466 SPT was performed at 0.1s resolution and the MSDs of cytoplasmic particles do not show any sign of saturation.

One possible control would be to use velocity-velocity correlation functions (by SPT, PMID: 22713559, we do not know whether there is be a FCS analog of this). As far as we know this kind of analysis has not been published on cytoplasmic particles.

Movements on very small lags should not be affected by confinement. Since FCS allows to probe very small lags, the authors may try to examine how robust their results on confinement are if they limit the range of lags to the smaller values. For example repeating the analyses of Figure 2 as a function of an upper bound in the lag time.

Experimentally, the A22 control is not satisfactory unless the dry mass density is controlled for in some way. L forms may be obtained with several protocols, but once again density has to be measured and accounted for. Possibly FCS of GFP muNS particles can be of some use.

Side note: is A22 now in place of cephalexin or in addition to it? This may be important as there have been claims (Lobritz 2015, PNAS) of β-lactams increasing cell respiration rates (and thus change metabolic rates, and thus alter cytoplasmic metabolic stirring?). A control of diffusion in untreated cells VS cells treated with Cepha or Cepha+A22 is needed here.

One interesting control on width could use the cell-to-cell variability within a population, to check whether there is some effect.

2) The controls on cell size and density should apply also to the other main results.

Nutrient changes should keep the crowding levels (dry mass density, PMID: 4600702) constant but vary a lot cell geometry and width PMID: 13611202 (and thus are entangled with the control on cell width of the previous part of the study).

The other perturbations affect crowding (and some also cell geometry), and SPT results suggest that crowding levels recapitulate many (though not all) of the observed variations in mobility (see e.g. PMID: 33083729).

Regarding temperature effects, it would be interesting to compare with the results in

PMID: 22517744, which (using SPT) argues in favor of active (nonthermal) motion.

Regarding this point ATP depletion might also be an interesting control. Cell metabolism and. "stirring" is presumably pretty different at 25 or 35C.

Osmotic shocks (p19): besides checking cell size and density, it was not clear at what point before measurements was salt added. Were these cells shocked and allowed to recover?

Comparing measurements at the pole with measurements at midcell could also provide further insight (also maybe to claim a role for the chromosome).

[Editors’ note: further revisions were suggested prior to acceptance, as described below.]

Thank you for resubmitting your work entitled "Dependence of diffusion in *Escherichia coli* cytoplasm on protein size, environmental conditions and cell growth" for further consideration by *eLife*. Your revised article has been evaluated by Naama Barkai (Senior Editor) and a Reviewing Editor.

The manuscript has been improved but there are some remaining issues that need to be addressed, as outlined below:

Both reviewers acknowledged that the revised paper is significantly improved, and would be adequate for *eLife*.

However, please address the two comments by reviewer #2, regarding the consistency of the data with subdiffusive behavior, and the interpretation of the density measurements.

*Reviewer #1 (Recommendations for the authors):*

The authors have made a strong response to reviewers' comments on the first submission. I think the novelty and significance of the findings are now much clearer.

*Reviewer #2 (Recommendations for the authors):*

I have shared again the revisions with the same close experimental collaborator. We are happy about the changes but we still have two main outstanding issues that seem possibly important, and we would like the authors to address.

1) We appreciate that diffusion is the most parsimonious scenario, but there is a different (important) question. If the data were derived from subdiffusive particles, would the technique reveal it and to what quantitative extent the data must deviate from diffusion in order to be detected?

Probably several indications that the authors have could be used to support the authors' conclusions. For example, Figure 2 supplement 3 and the plot on time cutoffs provided in the reply seem in line with their interpretation.

Could the authors show with the technique used in Figure 2 supplement 3 that DnaK-sfGFP behaves differently?

Additionally, probably the authors can strengthen this point with additional arguments, e.g. by analyzing simulated data from subdiffusive particles and investigating the limitations of the technique in detecting this "ground truth".

In brief, we ask the authors not to lean automatically on the most parsimonious scenario, but to gather the existing evidence/arguments in the direction of rejecting subdiffusion, and address the point in a focused discussion in the text. Also extend the arguments whenever possible (also based on previous recommendations).

2) We are grateful that the authors provided extra measurements connected to the problem of density change, but we are not entirely convinced and/or we do not fully understand the results.

Looking at figure 2 supplement 2 it seems that cephalexin and A22 have quite some effect on density.

The authors quote a 0.1% but it is not clear where this number comes from.

Possibly from a quoted literature value of 1.1 g/ml = 1000 Kg/m^3 (but the source should be cited, and the estimate explained), but probably they did not measure directly density (?). Also note that in the Oldewurtel et al. paper the mean value seems closer to 0.35 g/ml (and in Figure 3 of the same paper density perturbations from A22 seem non-negligible) .

Additionally, looking at the plots in Figure 1 supplement 8 there seems to be a visible difference in 1/tauD: the quoted P-value is 0.08, which is not so large considering that there are so few points.

Going back to the density measurements in Figure 2 – Supplement 2, the slope between the two plots is clearly different. It also seems difficult to fit an exponential in the unperturbed case, so maybe the channel is too small to achieve good sensitivity in this case.

If one has to judge visually the differences in z0 between perturbed and unperturbed case they could be in the range of a factor of 10-100 (in the treated cases z0 seems of the order of the channel size, in the untreated case it is much larger).

Hence at fixed volume, δ rho would also be different by a factor of 10-100. Instead, it's only a factor of 2, which means that volume changes by a factor of 5-50. Already a factor of five seems quite large.

In brief, we would ask the authors to clarify their measurements of density and mobility (show the fits, quote the volume measurements, describe the estimates, possibly perform more measurements etc.)

---

## [Author Response]

[Editors’ note: the authors resubmitted a revised version of the paper for consideration. What follows is the authors’ response to the first round of review.]

Reviewer #1 (Recommendations for the authors):A very thorough study of protein diffusion in the *E. coli* cytoplasm, looking at multiple proteins (including mutants in which specific interactions are disabled), multiple conditions and two different methods to measure diffusion. I don't think it leads to any major conceptual advances: basically the results confirm what was already inferred from the previous studies cited: smaller proteins roughly follow the Stokes-Einstein relation for the size dependence of the diffusion coefficient, larger proteins show subdiffusion, interactions with other cell components slow everything down, different measurement methods give comparable but not identical answers.

We might have been too conservative in the discussion of our results and have not sufficiently emphasized the novelty of our findings. We do believe that – besides being the most comprehensive study of cytoplasmic protein diffusion in bacteria, as acknowledged by both referees – our work allows us to draw a number of conceptually important and fundamental conclusions:

First and foremost, the apparent “simplicity” of protein diffusion demonstrated by our study is by no means trivial, and it does resolve an important discussion in the field. We are well aware that this Reviewer has previously concluded that the Stokes-Einstein relation is valid for bacterial cytoplasmic proteins, based on measurements for a small set of proteins. However, most other studies (including those that appeared after publication of this Referee’s work) concluded that the size dependence of protein diffusion is significantly steeper. Our work could resolve this contradiction, by showing that while the apparent size dependence is indeed steeper than predicted by the Stokes-Einstein relation, it could be reconciled with this relation if the peculiar “dumbbell shape” of the utilized GFP fusion proteins is taken into account by the model. Moreover, we would argue that drawing a reliable conclusion about the exact size dependence of protein diffusion based on a small set of proteins is simply difficult to impossible, given large variability between mobilities of individual proteins of similar mass seen in Figure 1D of our manuscript. In this context, systematic measurements for a large set of proteins were essential to determine the upper limit on protein mobility at a particular molecular mass, which is given by free diffusion.

Similarly non-trivial is the (apparently simple) conclusion that cytoplasmic protein mobility is normal rather than subdiffusive. Here again, only systematic measurements for a large number of proteins enabled us to clearly distinguish between the general trend of α values and protein-specific deviations from this trend (see Figure 1E). Additionally, modeling and simulations were required to demonstrate that the observed deviation of α from the normal diffusive behavior could be explained for most proteins by the confined geometry of a cell, rather than by real subdiffusion. To correct the statement of this reviewer, we actually did not observe dependence of α (i.e. of subdiffusion) on protein size in the measured size range. Pronounced subdiffusion for individual proteins only emerged as a consequence of extensive interactions with other proteins, which might be intuitive and theoretically predicted but again not trivial, and to the best of our knowledge it has not been experimentally shown for cytoplasmic proteins in bacteria yet.

Moreover, and this part was obviously not sufficiently emphasized in the previous version of our manuscript, our results demonstrate that all tested physiological perturbations that affect protein diffusion in bacterial cytoplasm (including osmolarity, temperature, antibiotic treatment and active cell growth) do not change the Stokes-Einstein size dependence or normality of protein diffusion but rather affect all tested proteins in a proportional manner. Thus, the effects of all these different perturbations could be explained by changes in the cytoplasmic viscosity (at least within the tested size range of individual proteins). We believe that this conclusion is of high conceptual importance for understanding physical properties of bacterial cytoplasm under different conditions experienced by bacteria. In order to better emphasize these latter findings, we have modified Figure 4 (now Figure 5) to illustrate that changes in protein mobility under different perturbations are proportional for proteins of different size.

Finally, we demonstrate that proteins from other, even very distant, bacteria, show very similar diffusion times to their *E. coli* counterparts, and only slight subdiffusion, meaning there is little if any organismal specificity of protein diffusion among bacteria. This is another important and novel conclusion that is now shown in the main text (new Figure 4).

There is merit in having such a comprehensive set of measurements all in one place: this will be a valuable reference point for anyone who wants to explore the physical nature of the cytoplasm further, or who wants to factor cytoplasmic protein diffusion into a model for some dynamic process in *E. coli*.

We completely agree on this point. We believe that having such a comprehensive data set is indeed absolutely important for quantitative understanding of the physical properties of bacterial cytoplasm and its changes upon different physicochemical perturbation, and (as argued above) only the acquisition of such large set of data enabled us to reliably distinguish the general trend from individual protein-specific effects. We now rephrased several of our conclusions and modified figures to better highlight the importance of our findings, including characterization of protein mobility across different physicochemical perturbation and the effect of cell growth.

The paper is well-written and well-presented, and I cannot find any technical flaws with the parts that I am able to judge.

We thank the reviewer for acknowledging the quality of our work from the technical point of view.

The weakness is maybe a lack of novelty: this provides strong confirmation of things that were already inferred on the basis of less complete data, but I missed any major conceptual advances in the understanding of the dynamics of the cytoplasm. If there is anything I missed, please highlight it better!

We apologize for not sufficiently emphasizing the novelty and significance of our finding in the previous version of the manuscript. From our perspective, the main value of the manuscript is in providing a comprehensive data set (see the comment above), combined with modeling, to quantitatively characterize physical properties of bacterial cytoplasm under different conditions. Although, unsurprisingly, some of our results are consistent with previous findings, our systematic approach helped to resolve several debated questions, and it provided a consistent and surprisingly simple description of the cytoplasmic protein diffusion (see discussion above), which we believe itself represents a major conceptual advance. Besides, we also report several novel and physiologically important findings, such as reduction of the cytoplasmic viscosity in growing bacteria. As suggested by the reviewer, we now rephrased the text to better highlight the novelty and conceptual advance provided by our findings.

Reviewer #2 (Recommendations for the authors):Belotto and co-workers performed a systematic experimental study of the intracellular mobility of 28 (+3 discarded on the way) cytoplasmic proteins in *E. coli*, using fluorescence correlation spectroscopy (FCS), complemented by some computational/ modeling. This technique is underexplored in the body of work looking at intracellular diffusion in bacteria, and allows them to probe very short time scales compared to commonly used techniques such as single-particle tracking (SPT).The manuscript is not really focused on a main finding, but a main finding may be that the data cannot falsify a Brownian diffusion model when confinement is accounted for (see below). Other interesting findings consider the temperature and growth-rate dependency, and the agreement of FCS with FRAP (recovery after photobleaching) data.The work is well written and provides a useful and precise set of measurements to the community. Its main advantages are it being systematic (28 is a large set of proteins for this kind of study), the novelty of FCS in this context, and the use of physical models in support of the data. However, we have some major concerns about the main results and conclusions:0) In the way it is written, the manuscript does not identify a central question, and the findings could be better connected to the current debate.

We thank the reviewer for the overall positive assessment of the quality of our manuscript, and for acknowledging the systematic nature of our work and the novelty of using FCS to probe protein diffusion in bacteria on short time scales. Indeed, providing a systematic and possibly comprehensive view of protein diffusion and its dependence on physicochemical perturbations in bacterial cells was the main aim of our study. This systematic and comprehensive approach has made it difficult to focus the discussion on a single specific question or finding, but we acknowledge that our most important conclusions could have been better highlighted (as also mentioned in our response to the Reviewer #1) and discussed more extensively in the context of the current debate. We have now addressed these issues by modifying the text, including the Abstract, as well as figures.

Related to the comment on the novelty of using FCS in bacteria, we also now mention in the text that the methodology developed in our manuscript is generally applicable to study protein diffusion in a confined space by FCS, both in bacteria and in other cellular systems.

1) In our view the central question could become the fact that the authors argue that conventional diffusion seems to be supported (or at least cannot be ruled out) for these data, while previous (SPT) data have supported mild but clear subdiffusion for (larger) cytoplasmic particles, including protein complexes. However, we have some possibly important concerns about this analysis, and in any case we think it needs more experimental and data-analysis/modeling controls (see below).

We absolutely agree with the Reviewer that one of our main findings is the consistency of our results for most proteins with the normal diffusion (and identifying protein-protein interactions as a cause of subdiffusion for other proteins), which is in contrast to the observations for larger cytoplasmic particles. We also thank the Reviewer for raising several important points that we now address in the revised version of the manuscript.

2) The other main results (dependency on growth, temperature, etc.) are interesting, but most of these things have been quantified by SPT, and a more careful comparison appears to be needed. Additionally, these results also need controls on cell size and density/crowding levels (see below).

We again agree with the Reviewer that quantifying the dependency of mobility of differently-sized proteins on physicochemical perturbations of bacterial cytoplasm is another main point of our work. Although for some of these perturbations the effects on mobility of large cytoplasmic particles or nucleoid loci have been previously quantified by SPT (and in some cases also measured for free GFP), there was no systematic investigation of mobility of individual differently-size proteins under these different physiological perturbations. We have now modified Figure 5 (former Figure 4) to better highlight the size dependence of protein mobility under these different conditions. Therefore, our results are novel and relevant and not redundant with what was previously investigated by SPT in a different range of molecular mass. Nevertheless, we have now expanded the comparison of our results with the previous SPT work. We have also performed quantification of cells size and density as suggested by the Reviewer.

With these important revisions, we believe the work could make a nice addition to the current debate.

We thank the Reviewer for this positive feedback.

We try to detail our impressions in the comments below.1) Our main concern is that the controls/support of the claim of conventional diffusion may be insufficient. If well supported, my impression is that this could become (either way) a central result of the study.The authors clearly show using modeling that their data cannot falsify Brownian diffusion. However, it is not clear to us that they can falsify fBM or fLe subdiffusion.

If we understand this question correctly, the Reviewer is asking about falsification of specific models that can describe subdiffusion. Since most analyzed proteins show little if any deviation from normal diffusion, we do not believe that falsifying these models would be necessary. To elaborate further on this point, fractional Langevin Equation (fLe) or fractional Brownian Motion (fBM) approaches are phenomenological models in non-equilibrium statistical physics that are built to study situations where non-equilibrium processes that are intrinsically based on fractal statistics govern the dynamics of the system. To be parsimonious in the interpretation of our data, we would hypothesize the existence or relevance of such processes either when there is a priori a physical mechanistic reason to include them, or if it is impossible to describe the observations using equilibrium statistical physics. In the current case, we assert that if the dumbbell structure of tagged proteins and the role of the confining wall is taken into consideration, then equilibrium statistical physics suffices and can quantitatively account for the systematic large-scale observations, and therefore considering these more complex models appears unnecessary.

A22 treatment provides an interesting control but PMID: 34341116 (see Figure 3) clearly shows that this treatment affects dry-mass density (QPI measurements are actually a proxy of macromolecular density, hence crowding).Previous studies have clearly supported the idea that density (crowding) affects cytoplasmic diffusion (see e.g. PMID: 33083729), hence it seems to us that we do not know whether the observed changes in FCS may come from the density changes rather than the confinement. For example the previously observed anomalous diffusion could be due to larger size of (non-interacting) proteins or protein complexes, or to the presence of the chromosome, or the current data would just not allow to falsify either scenario, etc.

This is indeed a valid point, and we thank the Reviewer for raising it. We performed measurements of the cellular density after treatment with cephalexin, and cephalexin plus A22 (presented in new Figure 2 —figure supplement 2). While we observed that treatment with cephalexin slightly reduced cellular density, the effect was minor (<0.1%) and additional treatment with A22 had even less (and not significant) impact. Thus, we believe that the effect of treatment with A22 is primarily due to changes in the cell width and not to the cytoplasmic crowding. But we nevertheless discuss this possibility citing the paper mentioned by the Reviewer and now interpret our A22 experiment more cautiously. This does not change, however, our simulation-based conclusion that cell confinement can account for the weak apparent anomaly of diffusion. Moreover, we added the FCS measurements using a smaller confocal volume (less optimal for regular experiments due to worse signal to noise ratio) to probe diffusion of proteins further away from the cell boundary (new Figure 2 —figure supplement 3). Consistent with our hypothesis, these measurements yielded significantly higher values of α.

As for the second point raised by the Reviewer, we do not question the validity of the previously measured subdiffusion of large protein particles or try to elucidate its causes, since our work analyses mobility of smaller proteins. What we clearly see in our work is that proteins showing most pronounced subdiffusion are the ones known to interact extensively with other proteins or with large multiprotein complexes, and disrupting these known interactions makes their diffusion more normal. Besides these specific interactions, recent simulation-based work (PMID: 31036655, cited in our revised manuscript) shows that larger proteins are more likely to engage in weak non-specific interactions, which might lead to slower subdiffusive mobility. But within the measured range of protein sizes this effect appears to be moderate.

Also, the lack of a clear Stokes-Einstein relation even using fairly complex models of diffusion makes us think of a possible complex underlying dynamics (due to disorder or viscoelasticity, or both) or more in general other possible (but possibly interesting) physical scenarios.

We would like to emphasize that the deviation from the Stokes-Einstein relation in our data is only moderate and primarily observed for few largest proteins in our data set, or for proteins that are known to be interacting with multiple other proteins or with large multiprotein complexes. It is indeed plausible that the diffusion of the largest or strongly interacting proteins might exhibit more complex dynamics that is not captured by the current model of two linked proteins. Indeed, recent simulations (PMID: 31036655 mentioned above) suggest that larger proteins may have stronger propensity to be engaged in non-specific protein-protein interactions. We extended our discussion to include these possibilities.

Below we try to propose some controls and analyses.The authors do not mention that (larger) protein particles like GFP muNS were also be reported to be subdiffusive by SPT. For example in PMID: 30374466 SPT was performed at 0.1s resolution and the MSDs of cytoplasmic particles do not show any sign of saturation.

We apologize for not sufficiently covering this previous work on larger protein particles. We did mention in the introduction that mobility of larger nucleoprotein particles was shown to be subdiffusive, but we agree that the work mentioned by the Reviewer should be cited, too, which is now corrected.

One possible control would be to use velocity-velocity correlation functions (by SPT, PMID: 22713559, we do not know whether there is be a FCS analog of this). As far as we know this kind of analysis has not been published on cytoplasmic particles.

As discussed above, we see little evidence of subdiffusion, except for largest and strongly interacting proteins, and we thus believe that such analysis that enables to distinguish between different models of subdiffusion would not be necessary (and to our understanding such analysis is not possible using FCS).

Movements on very small lags should not be affected by confinement. Since FCS allows to probe very small lags, the authors may try to examine how robust their results on confinement are if they limit the range of lags to the smaller values. For example repeating the analyses of Figure 2 as a function of an upper bound in the lag time.

We thank the Reviewer for this suggestion. Reducing the analysis to shorter lag times indeed significantly decreases the anomaly of diffusion (i.e. making α closer to 1), as shown in Author response image 1 for several constructs. In contrast, for DnaK-sfGFP that is truly subdiffusive due to interactions with other proteins, α remains much lower even at shorter times, as expected.

**Author response image 1. sa2fig1:** 

Since such truncation of the data to include only shorter times is not common for the analysis of FCS data, we were rather reluctant to include this analysis in the manuscript itself (but we could do if the Reviewer feels that it is essential). Instead, we now addressed the same question in a different way, having performed FCS measurements for the smaller confocal volume (shown in new Figure 2 —figure supplement 3). Although this reduction of the volume decreases intensity of the fluorescence and thus makes the measurements more difficult, we do observe that they yield a significantly increased value of α, consistent with our expectations.

Experimentally, the A22 control is not satisfactory unless the dry mass density is controlled for in some way. L forms may be obtained with several protocols, but once again density has to be measured and accounted for. Possibly FCS of GFP muNS particles can be of some use.

As mentioned before, we now included measurements of density for cells treated with cephalexin and A22. The reduction in cell density observed under our conditions is very minor and not significant compared to the effect of cephalexin alone. But we nevertheless discuss possible effects of A22 on cytoplasmic density as suggested by the Reviewer, since we agree that it is an important point.

Side note: is A22 now in place of cephalexin or in addition to it? This may be important as there have been claims (Lobritz 2015, PNAS) of β-lactams increasing cell respiration rates (and thus change metabolic rates, and thus alter cytoplasmic metabolic stirring?). A control of diffusion in untreated cells VS cells treated with Cepha or Cepha+A22 is needed here.

We apologize for the unclarity. A22 was in addition to cephalexin, we made this clearer in the text. In addition, we performed an experiment with cells with or without cephalexin treatment, which showed that, for cells with similar length, the protein mobility and anomaly of diffusion are comparable in these two conditions, suggesting that cephalexin treatment itself does not have measurable effect on protein mobility (although it does slightly (by ~ 0.1%) reduce cell density, as our newly added control measurements show).

One interesting control on width could use the cell-to-cell variability within a population, to check whether there is some effect.

We made and now present such analysis for the entire population of cells expressing sfGFP (new Figure 1 —figure supplement 7). There appear to be a weak trend for α to increase with cell width, as might be expected, but due to the limited natural variation in cell width and moderate sample size this effect is not significant within our data set.

2) The controls on cell size and density should apply also to the other main results.Nutrient changes should keep the crowding levels (dry mass density, PMID: 4600702) constant but vary a lot cell geometry and width PMID: 13611202 (and thus are entangled with the control on cell width of the previous part of the study).

Since in our experiments we were typically choosing cells of similar length, we expectedly see no differences in the range of cell lengths upon different treatment, and consequently also no trend in protein mobility or α with cell length (new Figure 1 —figure supplement 7). In contrast, cell width is indeed affected by some treatments, possibly correlating with the cell density changes and in some cases with protein mobility (new Figure 2 —figure supplement 2 and Figure 5 —figure supplement 1). We thank the Reviewer for these suggestions.

The other perturbations affect crowding (and some also cell geometry), and SPT results suggest that crowding levels recapitulate many (though not all) of the observed variations in mobility (see e.g. PMID: 33083729).

Again, we thank the Reviewer for this suggestion. Indeed, in some (though not all) cases, the observed changes in protein mobility, cell width and cell density correlate with each other. This is now discussed in the manuscript.

Regarding temperature effects, it would be interesting to compare with the results inPMID: 22517744, which (using SPT) argues in favor of active (nonthermal) motion.Regarding this point ATP depletion might also be an interesting control. Cell metabolism and. "stirring" is presumably pretty different at 25 or 35C.

We previously performed the treatment with DNP for sfGFP at 25°C, but we did not observe any effect of the ATP depletion by DNP treatment on protein mobility and therefore did not include these data in the previous version of the manuscript. We have now expanded this analysis to two other constructs and an additional temperature, as suggested by the Reviewer. Except for one case (Adk-sfGFP at 35°C), we still did not observe any significant impact of DNP treatment, suggesting that at least under our conditions and within measured protein size, the non-thermal mixing does not seem to occur in *E. coli* cytoplasm. We now specifically discuss this in the manuscript.

As a sidenote, the exception of Adk is actually quite interesting, since this enzyme uses ATP as a substrate, and depletion of ATP might either change its conformation or activity at higher temperature, thereby affecting mobility. But we aim to study this effect in detail in our subsequent work.

Osmotic shocks (p19): besides checking cell size and density, it was not clear at what point before measurements was salt added. Were these cells shocked and allowed to recover?

Again, thank you for this suggestion. We checked the cell size and density, and the density exhibited expected increase at high osmolarity. Cells were indeed allowed to adapt to a final ionic strength of 305 mM and the experiments were performed in agarose pads prepared in the same buffer. FCS was measured only in cells that do not show plasmolysis.

Comparing measurements at the pole with measurements at midcell could also provide further insight (also maybe to claim a role for the chromosome).

We now included experiments where diffusion of sfGFP and a larger construct was measured within the nucleoid and in the cytoplasm of cells where the nucleoid was compacted by the chloramphenicol treatment and stained with Sytox Orange dye (new Figure 5 —figure supplement 5). We observed only minor difference between protein mobility at the two positions of the cell.

[Editors’ note: further revisions were suggested prior to acceptance, as described below.]

Reviewer #2 (Recommendations for the authors):I have shared again the revisions with the same close experimental collaborator. We are happy about the changes but we still have two main outstanding issues that seem possibly important, and we would like the authors to address.

We thank the Reviewer for the positive feedback on this revised version of our manuscript and for raising the remaining points that still required clarification.

1) We appreciate that diffusion is the most parsimonious scenario, but there is a different (important) question. If the data were derived from subdiffusive particles, would the technique reveal it and to what quantitative extent the data must deviate from diffusion in order to be detected?Probably several indications that the authors have could be used to support the authors' conclusions. For example, Figure 2 supplement 3 and the plot on time cutoffs provided in the reply seem in line with their interpretation.

As suggested by the Reviewer, the plot of the time-dependence of the anomalous diffusion exponent has now been added as Figure 2 —figure supplement 5, and these results are discussed in the same paragraph as the measurements at smaller pinhole size.

Could the authors show with the technique used in Figure 2 supplement 3 that DnaK-sfGFP behaves differently?

Results from the measurement of DnaK-sfGFP diffusion with smaller pinhole size (0.66 Airy units) have now been added to Figure 2 —figure supplement 4 (former Figure 2 —figure supplement 3). As expected for truly subdiffusive behavior, the value of anomalous diffusion exponent for DnaK-sfGFP did not increase with smaller pinhole size, in contrast to other tested proteins.

Additionally, probably the authors can strengthen this point with additional arguments, e.g. by analyzing simulated data from subdiffusive particles and investigating the limitations of the technique in detecting this "ground truth".

As suggested by the Reviewer, we have performed additional simulations of FCS measurements using a model of subdiffusion, fractional Brownian motion, for fluorescent proteins under confinement. According to these simulations, in a cell of the experimentally observed average diameter *d* = 0.85 µm, we find that the anomalous diffusion exponent *α* extracted by fitting the simulated autocorrelation function is approximately an affine function of the “ground truth” (ansatz) coefficient. We estimate that the range of *α* observed in our experiments for most protein constructs (*α*_measured_ = 0.82 – 0.90) corresponds to the unconfined subdiffusion exponent in the range [0.95-1], hence very close to Brownian. These results are now shown as Figure 2 —figure supplement 1 and discussed in the text.

In brief, we ask the authors not to lean automatically on the most parsimonious scenario, but to gather the existing evidence/arguments in the direction of rejecting subdiffusion, and address the point in a focused discussion in the text. Also extend the arguments whenever possible (also based on previous recommendations).

We believe that the additional experiments and simulations suggested by the Reviewer provided further evidence for our conclusions. We also amended the discussion of this point, as suggested by the Reviewer.

2) We are grateful that the authors provided extra measurements connected to the problem of density change, but we are not entirely convinced and/or we do not fully understand the results.Looking at figure 2 supplement 2 it seems that cephalexin and A22 have quite some effect on density/The authors quote a 0.1% but it is not clear where this number comes from.Possibly from a quoted literature value of 1.1 g/ml = 1000 Kg/m^3 (but the source should be cited, and the estimate explained), but probably they did not measure directly density (?). Also note that in the Oldewurtel et al. paper the mean value seems closer to 0.35 g/ml(and in Figure 3 of the same paper density perturbations from A22 seem non-negligible)

As explained in the Materials and methods, we resuspend the cells in a mix of 20% iodixanol in buffer, which has been adjusted to have the same density as the one of the wild-type, untreated cells, ~ 1.11 g/ml (see ref. (Martínez-Salas, Martín, and Vicente 1981)). The reference was cited in the corresponding section of the Materials and methods and is now reported also in the Results section. Note that the value given in Oldewurtel is the dry mass density (ρdry), *i.e.* corrected for the water content of the cell (Φw), hence the difference with our value, which do account for the water content (ρ=ρdry+Φwρw). Since the average densities of proteins, DNA and water are constant, the two variables are an affine function of each other, since they depend on the volume fraction of proteins and DNA ΦPD=1−Φw. Note also that iodixanol is a standard density-matching agent for biological samples, and does not penetrate in the cells. Since the suspending medium density is matched to the density of untreated cells (within a ~ 0.5 g/l error), we are able to precisely measure small density differences for the treated cells by measuring tilts in the cell sedimentation profile.

Additionally, looking at the plots in Figure 1 supplement 8 there seems to be a visible difference in 1/tauD: the quoted P-value is 0.08, which is not so large considering that there are so few points.

We now rephrased our conclusions more cautiously, explicitly citing the *P*-value when referring to the results of Figure 1 —figure supplement 8. Such (slightly) higher mobility might indeed be consistent with the (slightly) lower density of the cephalexin-treated cells (Figure 2 —figure supplement 3).

Going back to the density measurements in Figure 2 – Supplement 2, the slope between the two plots is clearly different. It also seems difficult to fit an exponential in the unperturbed case, so maybe the channel is too small to achieve good sensitivity in this case.If one has to judge visually the differences in z0 between perturbed and unperturbed case they could be in the range of a factor of 10-100 (in the treated cases z0 seems of the order of the channel size, in the untreated case it is much larger).Hence at fixed volume, δ rho would also be different by a factor of 10-100. Instead, it's only a factor of 2, which means that volume changes by a factor of 5-50. Already a factor of five seems quite large.In brief, we would ask the authors to clarify their measurements of density and mobility (show the fits, quote the volume measurements, describe the estimates, possibly perform more measurements etc.)

We have now added to Figure 2 —figure supplement 3 (former Figure 2 —figure supplement 2) the curves of the exponential fits n(z)=n0exp⁡(−zz0). Note that the fit has only two free parameters, n0 and a=1z0, and therefore converges unambiguously even when the profile is fairly flat, as is the case for untreated cells. The parameter a=1z0 is simply close to 0 in this case. Note also that the Boltzmann cell density distribution for thermal particles under gravity is an exponential that decays to 0 at infinity, and not to some baseline, including when the suspension is sandwiched between two walls.

We now plot the inverse decay lengths 1z0 for all conditions in Figure 2 —figure supplement 3H. As we can see, there is a factor of ~2 between the inverse decay lengths for the cells treated with cephalexin and the cells treated with both cephalexin and A22. This factor of ~2 can actually be fully accounted for by the difference in volume of the cells between the two treatments. We now plot the estimated volumes in Figure 2 – supplementary figure 3I. Since 1z0, as explained in Materials and methods, the estimated cell volumetric mass density, relative to the 20% iodixanol in buffer suspending fluid, is therefore very similar between cells treated with cephalexin and cells treated with cephalexin and A22, and slightly lower than the volumetric mass density of untreated cells.